# Byzantine-Tolerant Methods for Distributed Variational Inequalities

**Nazarii Tupitsa**
MBZUAI, MIPT

**Abdulla Jasem Almansoori**
MBZUAI

**Yanlin Wu**
MBZUAI

**Martin Takáč**
MBZUAI

**Karthik Nandakumar**
MBZUAI

**Samuel Horváth**
MBZUAI

**Eduard Gorbunov**[*]
MBZUAI

## Abstract

Robustness to Byzantine attacks is a necessity for various distributed training scenarios. When the training reduces to the process of solving a minimization problem, Byzantine robustness is relatively well-understood. However, other problem formulations, such as min-max problems or, more generally, variational inequalities, arise in many modern machine learning and, in particular, distributed learning tasks. These problems significantly differ from the standard minimization ones and, therefore, require separate consideration. Nevertheless, only one work [Adibi et al., 2022] addresses this important question in the context of Byzantine robustness. Our work makes a further step in this direction by providing several (provably) Byzantine-robust methods for distributed variational inequality, thoroughly studying their theoretical convergence, removing the limitations of the previous work, and providing numerical comparisons supporting the theoretical findings.

## 1 Introduction

Modern machine learning tasks require to train large models with billions of parameters on huge datasets to achieve reasonable quality. Training of such models is usually done in a distributed manner since otherwise it can take a prohibitively long time [Li, 2020]. Despite the attractiveness of distributed training, it is associated with multiple difficulties not existing in standard training.

In this work, we focus on one particular aspect of distributed learning – *Byzantine tolerance/robustness* – the robustness of distributed methods to the presence of *Byzantine workers*[2], i.e., such workers that can send incorrect information (maliciously or due to some computation errors/faults) and are assumed to be omniscient. For example, this situation can appear in collaborative training, when several participants (companies, universities, individuals) that do not necessarily know each other train some model together [Kijsipongse et al., 2018, Diskin et al., 2021] or when the devices used in training are faulty [Ryabinin et al., 2021]. When the training reduces to the distributed *minimization* problem, the question of Byzantine robustness is studied relatively well both in theory and practice [Karimireddy et al., 2022, Lyu et al., 2020].

However, there are a lot of problems that cannot be reduced to minimization, e.g., adversarial training [Goodfellow et al., 2015, Madry et al., 2018], generative adversarial networks (GANs) [Goodfellow et al., 2014], hierarchical reinforcement learning [Wayne and Abbott, 2014, Vezhnevets et al., 2017], adversarial examples games [Bose et al., 2020], and other problems arising in game theory, control theory, and differential equations [Facchinei and Pang, 2003]. Such problems lead to min-max

---

[*]Corresponding author: `eduard.gorbunov@mbzuai.ac.ae`

[2]The term "Byzantine workers" is a standard term for the field [Lamport et al., 1982, Lyu et al., 2020]. We do not aim to offend any group of people but rather use common terminology.

or, more generally, variational inequality (VI) problems [Gidel et al., 2018] that have significant differences from minimization ones and require special consideration [Harker and Pang, 1990, Ryu and Yin, 2022]. Such problems can also be huge scale, meaning that, in some cases, one has to solve them distributedly. Therefore, similarly to the case of minimization, the necessity in Byzantine-robust methods for distributed VIs arises.

The only existing work addressing this problem is [Adibi et al., 2022], where the authors propose the first Byzantine-tolerant distributed method for min-max and VI problems called Robust Distributed Extragradient (RDEG). However, several interesting directions such as application of $(\delta, c)$-robust aggregation rules, client momentum [Karimireddy et al., 2021], and checks of computations [Gorbunov et al., 2022b] studied for minimization problems are left unexplored in the case of VIs. Moreover, [Adibi et al., 2022] prove the convergence to the solution's neighborhood that can be reduced only via increasing the batchsize and rely on the assumption that the number of workers is sufficiently large and the fraction of Byzantine workers is smaller than $1/16$, which is much smaller than for SOTA results in minimization case. *Our work closes these gaps in the literature and resolves the limitations of the results from [Adibi et al., 2022].*

## 1.1 Setting

To make the further presentation precise, we need to introduce the problem and assumptions we make. We consider the distributed unconstrained variational inequality (non-linear equation) problem[3]:

$$\text{find } \boldsymbol{x}^* \in \mathbb{R}^d \text{ such that } F(\boldsymbol{x}^*) = 0, \text{ where } F(\boldsymbol{x}) := \frac{1}{G} \sum_{i \in \mathcal{G}} F_i(\boldsymbol{x}), \tag{1}$$

where $\mathcal{G}$ denotes the set of regular/good workers and operators $F_i$ have an expectation form $F_i(\boldsymbol{x}) := \mathbb{E}_{\boldsymbol{\xi}_i}[\boldsymbol{g}_i(\boldsymbol{x}; \boldsymbol{\xi}_i)]$. We assume that $n$ workers connected with a server take part in the learning/optimization process and $[n] = \mathcal{G} \sqcup \mathcal{B}$, where $\mathcal{B}$ is the set of *Byzantine workers* – the subset $\mathcal{B}$ of workers $[n]$ that can deviate from the prescribed protocol (send incorrect information, e.g., arbitrary vectors instead of stochastic estimators) either intentionally or not and are *omniscient*[4], i.e., Byzantine workers can know the results of computations on regular workers and the aggregation rule used by the server. The number of Byzantine workers $B = |\mathcal{B}|$ is assumed to satisfy $B \leq \delta n$, where $\delta < 1/2$ (otherwise Byzantines form a majority and the problem becomes impossible to solve). The number of regular workers is denoted as $G = |\mathcal{G}|$.

**Assumptions.** Here, we formulate the assumptions related to the stochasticity and properties of operators $\{F_i\}_{i \in \mathcal{G}}$.

**Assumption 1.** *For all $i \in \mathcal{G}$ the stochastic estimator $\boldsymbol{g}_i(\boldsymbol{x}, \boldsymbol{\xi}_i)$ is an unbiased estimator of $F_i(\boldsymbol{x})$ with bounded variance, i.e., $\mathbb{E}_{\boldsymbol{\xi}_i}[\boldsymbol{g}_i(\boldsymbol{x}, \boldsymbol{\xi}_i)] = F_i(\boldsymbol{x})$ and for some $\sigma \geq 0$*

$$\mathbb{E}_{\boldsymbol{\xi}_i}\left[\|\boldsymbol{g}_i(\boldsymbol{x}, \boldsymbol{\xi}_i) - F_i(\boldsymbol{x})\|^2\right] \leq \sigma^2. \tag{2}$$

The above assumption is known as the bounded variance assumption. It is classical for the analysis of stochastic optimization methods [Nemirovski et al., 2009, Juditsky et al., 2011] and is used in the majority of existing works on Byzantine robustness with theoretical convergence guarantees.

Further, we assume that the data heterogeneity across the workers is bounded.

**Assumption 2.** *There exists $\zeta \geq 0$ such that for all $\boldsymbol{x} \in \mathbb{R}^d$*

$$\frac{1}{G} \sum_{i \in \mathcal{G}} \|F_i(\boldsymbol{x}) - F(\boldsymbol{x})\|^2 \leq \zeta^2. \tag{3}$$

Condition (3) is a standard notion of data heterogeneity in Byzantine-robust distributed optimization [Wu et al., 2020, Zhu and Ling, 2021, Karimireddy et al., 2022, Gorbunov et al., 2023a]. It is worth

---

[3]We assume that the problem (1) has a unique solution $\boldsymbol{x}^*$. This assumption can be relaxed, but for simplicity of exposition, we enforce it.

[4]This assumption gives Byzantine workers a lot of power and rarely holds in practice. Nevertheless, if the algorithm is robust to such workers, then it is provably robust to literally any type of workers deviating from the protocol.

mentioning that without any kind of bound on the heterogeneity of $\{F_i\}_{i \in \mathcal{G}}$, it is impossible to tolerate Byzantine workers. In addition, homogeneous case ($\zeta = 0$) is also very important and arises in collaborative learning, see [Kijsipongse et al., 2018, Diskin et al., 2021].

Finally, we formulate here several assumptions on operator $F$. Each particular result in this work relies only on a subset of listed assumptions.

**Assumption 3.** *Operator $F : \mathbb{R}^d \to \mathbb{R}^d$ is L-Lipschitz, i.e.,*

$$\|F(\boldsymbol{x}) - F(\boldsymbol{y})\| \leq L\|\boldsymbol{x} - \boldsymbol{y}\|, \quad \forall \, \boldsymbol{x}, \boldsymbol{y} \in \mathbb{R}^d. \tag{Lip}$$

**Assumption 4.** *Operator $F : \mathbb{R}^d \to \mathbb{R}^d$ is $\mu$-quasi strongly monotone, i.e., for $\mu \geq 0$*

$$\langle F(\boldsymbol{x}), \boldsymbol{x} - \boldsymbol{x}^* \rangle \geq \mu\|\boldsymbol{x} - \boldsymbol{x}^*\|^2, \quad \forall \, \boldsymbol{x} \in \mathbb{R}^d. \tag{QSM}$$

**Assumption 5.** *Operator $F : \mathbb{R}^d \to \mathbb{R}^d$ is monotone, i.e.,*

$$\langle F(\boldsymbol{x}) - F(\boldsymbol{y}), \boldsymbol{x} - \boldsymbol{y} \rangle \geq 0, \quad \forall \, \boldsymbol{x}, \boldsymbol{y} \in \mathbb{R}^d. \tag{Mon}$$

**Assumption 6.** *Operator $F : \mathbb{R}^d \to \mathbb{R}^d$ is $\ell$-star-cocoercive, i.e., for $\ell \geq 0$*

$$\langle F(\boldsymbol{x}), \boldsymbol{x} - \boldsymbol{x}^* \rangle \geq \frac{1}{\ell}\|F(\boldsymbol{x})\|^2, \quad \forall \, \boldsymbol{x} \in \mathbb{R}^d. \tag{SC}$$

Assumptions 3 and 5 are quite standard for the literature on VIs. Assumptions 4 and 6 can be seen as structured non-monotonicity assumptions. Indeed, there exist examples of non-monotone (and even non-Lipschitz) operators such that Assumptions 4 and 6 holds [Loizou et al., 2021]. However, Assumptions 3 and 5 imply neither (QSM) nor (SC). It is worth mentioning that Assumption 4 is also known under different names, i.e., strong stability [Mertikopoulos and Zhou, 2019] and strong coherent [Song et al., 2020] conditions.

**Robust aggregation.** We use the formalism proposed by Karimireddy et al. [2021, 2022].

**Definition 1.1** (($\delta, c$)-RAGG [Karimireddy et al., 2021, 2022]). *Let there exist a subset $\mathcal{G}$ of random vectors $\{\boldsymbol{y}_1, \ldots, \boldsymbol{y}_n\}$ such that $G \geq (1 - \delta)n$ for some $\delta < 1/2$ and $\mathbb{E}\|\boldsymbol{y}_i - \boldsymbol{y}_j\|^2 \leq \rho^2$ for any fixed pair $i, j \in \mathcal{G}$ and some $\rho \geq 0$. Then, $\widehat{\boldsymbol{y}} = \mathrm{RAGG}(\boldsymbol{y}_1, \ldots, \boldsymbol{y}_n)$ is called ($\delta, c$)-robust aggregator if for some constant $c \geq 0$*

$$\mathbb{E}\left[\|\widehat{\boldsymbol{y}} - \overline{\boldsymbol{y}}\|^2\right] \leq c\delta\rho^2, \tag{4}$$

*where $\overline{\boldsymbol{y}} = \frac{1}{G}\sum_{i \in \mathcal{G}} y_i$. Further, if the value of $\rho$ is not used to compute $\widehat{\boldsymbol{y}}$, then $\widehat{\boldsymbol{y}}$ is called agnostic ($\delta, c$)-robust aggregator and denoted as $\widehat{\boldsymbol{y}} = \mathrm{ARAGG}(\boldsymbol{y}_1, \ldots, \boldsymbol{y}_n)$.*

The above definition is tight in the sense that for any estimate $\widehat{\boldsymbol{y}}$ the best bound one can guarantee is $\mathbb{E}\left[\|\widehat{\boldsymbol{y}} - \overline{\boldsymbol{y}}\|^2\right] = \Omega(\delta\rho^2)$ [Karimireddy et al., 2021]. Moreover, there are several examples of ($\delta, c$)-robust aggregation rules that work well in practice; see Appendix B.

Another important concept for Byzantine-robust learning is the notion of permutation inveriance.

**Definition 1.2** (Permutation invariant algorithm). *Define the set of stochastic gradients computed by each of the $n$ workers at some round $t$ to be $[\tilde{\boldsymbol{g}}_{1,t}, \ldots, \tilde{\boldsymbol{g}}_{n,t}]$. For a good worker $i \in \mathcal{G}$, these represent the true stochastic gradients whereas for a bad worker $j \in \mathcal{B}$, these represent arbitrary vectors. The output of any optimization algorithm $\mathrm{ALG}$ is a function of these gradients. A permutation-invariant algorithm is one which for any set of permutations over $t$ rounds $\{\pi_1, \ldots, \pi_t\}$, its output remains unchanged if we permute the gradients.*

$$\mathrm{ALG}\begin{pmatrix} [\tilde{\boldsymbol{g}}_{1,1}, ..., \tilde{\boldsymbol{g}}_{n,1}], \\ ... \\ [\tilde{\boldsymbol{g}}_{1,t}, ..., \tilde{\boldsymbol{g}}_{n,t}] \end{pmatrix} = \mathrm{ALG}\begin{pmatrix} [\tilde{\boldsymbol{g}}_{\pi_1(1),1}, ..., \tilde{\boldsymbol{g}}_{\pi_1(n),1}], \\ ... \\ [\tilde{\boldsymbol{g}}_{\pi_t(1),t}, ..., \tilde{\boldsymbol{g}}_{\pi_t(n),t}] \end{pmatrix}$$

As Karimireddy et al. [2021] prove, any permutation-invariant algorithm fails to converge to any predefined accuracy of the solution (under Assumption 1) even if all regular workers have the same operators/functions, i.e., even when $\zeta = 0$.

Table 1: Summary of known and new complexity results for Byzantine-robust methods for distributed variational inequalities. Column "Setup" indicates the varying assumptions. By the complexity, we mean the number of stochastic oracle calls needed for a method to guarantee that Metric $\leq \varepsilon$ (for RDEG $\mathbf{P}\{\text{Metric} \leq \varepsilon\} \geq 1 - \delta_{\mathsf{RDEG}}$, $\delta_{\mathsf{RDEG}} \in (0,1]$) and "Metric" is taken from the corresponding column. For simplicity, we omit numerical and logarithmic factors in the complexity bounds. Column "BS" indicates the minimal batch-size used for achieving the corresponding complexity. Notation: $c, \delta$ are robust aggregator parameters; $\alpha$ = momentum parameter; $\beta$ = ratio of inner and outer stepsize in SEG-like methods; $n$ = total numbers of peers; $m$ = number of checking peers; $G$ = number of peers following the protocol; $R$ = any upper bound on $\|x^0 - x^*\|$; $\mu$ = quasi-strong monotonicity parameter; $\ell$ = star-cocoercivity parameter; $L$ = Lipschitzness parameter; $\sigma^2$ = bound on the variance. The definition $x^T$ can vary; see corresponding theorems for the exact formulas.

| Setup | Method | Citation | Metric | Complexity | BS |
|---|---|---|---|---|---|
| SC, QSM | SGDA-RA | Cor. 1 | $\mathbb{E}[\|x^T - x^*\|^2]$ | $\frac{\ell}{\mu} + \frac{1}{c\delta n}$ | $\frac{c\delta\sigma^2}{\mu^2\varepsilon}$ |
| | M-SGDA-RA | Cor. 4 | | $\frac{\ell}{\mu\alpha^2} + \frac{1}{c\delta\alpha n}$ | $\frac{c\delta\sigma^2}{\alpha^2\mu^2\varepsilon}$ |
| | SGDA-CC | Cor. 6 | | $\frac{\ell}{\mu} + \frac{\sigma^2}{\mu^2 n\varepsilon} + \frac{\sigma^2 n^2}{\mu^2 m\varepsilon} + \frac{\sigma^2 n^2}{\mu^2 m\sqrt{\varepsilon}}$ | 1 |
| | R-SGDA-CC | Cor. 8 | | $\frac{\ell}{\mu} + \frac{\sigma^2}{n\mu\varepsilon} + \frac{n^2\sigma}{m\sqrt{\mu\varepsilon}}$ | 1 |
| Lip, QSM | SEG-RA | Cor. 3 | $\mathbb{E}[\|x^T - x^*\|^2]$ | $\frac{L}{\beta\mu} + \frac{1}{\beta c\delta G} + \frac{1}{\beta}$ | $\frac{c\delta\sigma^2}{\beta\mu^2\varepsilon}$ |
| | SEG-CC | Cor. 9 | | $\frac{L}{\mu} + \frac{1}{\beta} + \frac{\sigma^2}{\beta^2\mu^2 n\varepsilon} + \frac{\sigma^2 n^2}{\beta^2\mu^2 m\varepsilon} + \frac{\sigma^2 n^2}{\beta^2\mu^2 m\sqrt{\varepsilon}}$ | 1 |
| | R-SEG-CC | Cor. 11 | | $\frac{L}{\mu} + \frac{\sigma^2}{n\mu\varepsilon} + \frac{n^2\sigma}{m\sqrt{\mu\varepsilon}}$ | 1 |
| Lip, QSM | RDEG | Adibi et al. [2022][(1)] | $\|x^T - x^*\|^2$ | $\frac{L}{\mu}$ | $\frac{\sigma^2\mu^2 R^2}{L^4\varepsilon^2}$ |

[(1)] consider only homogeneous case ($\zeta = 0$) .

## 1.2 Our Contributions

Now we are ready to describe the main contributions of this work.

• **Methods with provably robust aggregation.** We propose new methods called Stochastic Gradient Descent-Ascent and Stochastic Extragradient with Robust Aggregation (SGDA-RA and SEG-RA) – variants of popular SGDA [Dem'yanov and Pevnyi, 1972, Nemirovski et al., 2009] and SEG [Korpelevich, 1976, Juditsky et al., 2011]. We prove that SGDA-RA and SEG-RA work with any $(\delta, c)$-robust aggregation rule and converge to the desired accuracy *if the batchsize is large enough*. In the experiments, we observe that SGDA-RA and SEG-RA outperform RDEG in several cases.

• **Client momentum.** As the next step, we add client momentum to SGDA-RA and propose Momentum SGDA-RA (M-SGDA-RA). As it is shown by [Karimireddy et al., 2021, 2022], client momentum helps to break the permutation invariance of the method and ensures convergence to any predefined accuracy with any batchsize for ***non-convex** minimization problems*. In the case of star-cocoercive quasi-strongly monotone VIs, we prove the convergence to the neighborhood of the solution; the size of the neighborhood can be reduced via increasing batchsize only – similarly to the results for RDEG, SGDA-RA, and SEG-RA. We discuss this limitation in detail and point out the non-triviality of this issue. Nevertheless, we show in the experiments that client momentum does help to achieve better accuracy of the solution.

• **Methods with random checks of computations.** Finally, for homogeneous data case ($\zeta = 0$), we propose a version of SGDA and SEG with random checks of computations (SGDA-CC, SEG-CC and their restarted versions – R-SGDA-CC and R-SEG-CC). We prove that the proposed methods converge *to any accuracy of the solution without any assumptions on the batchsize*. This is the first result of this type on Byzantine robustness for distributed VIs. Moreover, when the target accuracy of the solution is small enough, the obtained convergence rates for R-SGDA-CC and R-SEG-CC are not worse than the ones for distributed SGDA and SEG derived in the case of $\delta = 0$ (no Byzantine workers); see the comparison of the convergence rates in Table 1. In the numerical experiments, we consistently observe the superiority of the methods with checks of computations to the previously proposed methods.

## 1.3 Related Work

**Byzantine-robust methods for minimization problems.** Classical distributed methods like Parallel SGD [Zinkevich et al., 2010] cannot tolerate even one Byzantine worker. The most evident vulnerability of such methods is an aggregation rule (averaging). Therefore, many works focus on designing and application of different aggregation rules to Parallel SGD-like methods [Blanchard et al., 2017, Yin et al., 2018, Damaskinos et al., 2019, Guerraoui et al., 2018, Pillutla et al., 2022]. However, this is not sufficient for Byzantine robustness: there exist particular attacks [Baruch et al., 2019, Xie et al., 2019] that can bypass popular defenses. [Karimireddy et al., 2021] formalize the definition of robust aggregation (see Definition 1.1), show that many standard aggregation rules are non-robust according to that definition, and prove that any permutation-invariant algorithm with a fixed batchsize can converge only to the ball around the solution with algorithm-independent radius. Therefore, more in-depth algorithmic changes are required that also explain why RDEG, SGDA-RA, and SEG-RA are not converging to any accuracy without increasing batchsize.

One possible way to resolve this issue is to use client momentum [Karimireddy et al., 2021, 2022] that breaks permutation-invariance and allows for convergence to any accuracy. It is also worth mentioning a recent approach by [Allouah et al., 2023], who propose an alternative definition of robust aggregation to the one considered in this paper, though to achieve the convergence to any accuracy in the homogeneous case [Allouah et al., 2023] apply client momentum like in [Karimireddy et al., 2021, 2022]. Another line of work achieves Byzantine robustness through the variance reduction mechanism [Wu et al., 2020, Zhu and Ling, 2021, Gorbunov et al., 2023a]. Finally, for the homogeneous data case, one can apply validation test [Alistarh et al., 2018, Allen-Zhu et al., 2021] or checks of computations [Gorbunov et al., 2022b]. For the summary of other advances, we refer to [Lyu et al., 2020].

**Methods for min-max and variational inequalities problems.** As mentioned before, min-max/variational inequalities (VIs) problems have noticeable differences with standard minimization. In particular, it becomes evident from the differences in the algorithms' behavior. For example, a direct analog of Gradient Descent for min-max/VIs – Gradient Descent-Ascent (GDA) [Krasnosel'skii, 1955, Mann, 1953, Dem'yanov and Pevnyi, 1972, Browder, 1966] – fails to converge for a simple bilinear game. Although GDA converges for a different class of problems (cocoercive/star-cocoercive ones) and its version with alternating steps works well in practice and even provably converges locally [Zhang et al., 2022], many works focus on Extragradient (EG) type methods [Korpelevich, 1976, Popov, 1980] due to their provable convergence for monotone Lipschitz problems and beyond [Tran-Dinh, 2023]. Stochastic versions of GDA and EG (SGDA and SEG) are studied relatively well, e.g., see [Hsieh et al., 2020, Loizou et al., 2021, Mishchenko et al., 2020, Pethick et al., 2023] for the recent advances.

**On the results from [Adibi et al., 2022].** In the context of Byzantine robustness for distributed min-max/VIs, the only existing work is [Adibi et al., 2022]. The authors propose a method called Robust Distributed Extragradient (RDEG) – a distributed version of EG that uses a univariate trimmed-mean estimator from [Lugosi and Mendelson, 2021] for aggregation. This estimator satisfies a similar property to (4) that is shown for $\delta < 1/16$ and large enough $n$ (see the discussion in Appendix B). In contrast, the known $(\delta, c)$-robust aggregation rules allow larger $\delta$, and do not require large $n$. Despite these evident theoretical benefits, such aggregation rules were not considered in prior works on Byzantine robustness for distributed variational inequalities/min-max problems.

## 2 Main Results

In this section, we describe three approaches proposed in this work and formulate our main results.

### 2.1 Methods with Robust Aggregation

We start with the Stochastic Gradient Descent-Accent with $(\delta, c)$-robust aggregation (SGDA-RA):

$$\boldsymbol{x}^{t+1} = \boldsymbol{x}^t - \gamma \mathrm{RAGG}(\boldsymbol{g}_1^t, \ldots, \boldsymbol{g}_n^t), \quad \text{where } \boldsymbol{g}_i^t = \boldsymbol{g}_i(\boldsymbol{x}^t, \boldsymbol{\xi}_i^t) \ \forall i \in \mathcal{G} \quad \text{and} \quad \boldsymbol{g}_i^t = * \ \forall i \in \mathcal{B},$$

where $\{\boldsymbol{g}_i^t\}_{i \in \mathcal{G}}$ are sampled independently. The main result for SGDA-RA is given below.

**Theorem 1.** *Let Assumptions 1, 2, 4 and 6 hold. Then after $T$ iterations* SGDA-RA *(Algorithm 1) with $(\delta, c)$-RAGG and $\gamma \leq \frac{1}{2\ell}$ outputs $\boldsymbol{x}^T$ such that*

$$\mathbb{E}\|\boldsymbol{x}^T - \boldsymbol{x}^*\|^2 \leq \left(1 - \frac{\gamma\mu}{2}\right)^T \|\boldsymbol{x}^0 - \boldsymbol{x}^*\|^2 + \frac{2\gamma\sigma^2}{\mu G} + \frac{2\gamma c\delta(24\sigma^2 + 12\zeta^2)}{\mu} + \frac{c\delta(24\sigma^2 + 12\zeta^2)}{\mu^2}.$$

The first two terms in the derived upper bound are standard for the results on SGDA under Assumptions 1, 4, and 6, e.g., see [Beznosikov et al., 2023]. The third and the fourth terms come from the presence of Byzantine workers and robust aggregation since the existing $(\delta, c)$-robust aggregation rules explicitly depend on $\delta$. The fourth term cannot be reduced without increasing batchsize even when $\zeta = 0$ (homogeneous data case). This is expected since SGDA-RA is permutation invariant. When $\sigma = 0$ (regular workers compute full operators), then SGDA-RA converges linearly to the ball centered at the solution with radius $\mathcal{O}(\sqrt{c\delta}\zeta/\mu)$ that matches the lower bound from [Karimireddy et al., 2022]. In contrast, the known results for RDEG are derived for homogeneous data case ($\zeta = 0$). The proof of Theorem 1 is deferred to Appendix D.1.

Using a similar approach we also propose a version of Stochastic Extragradient method with $(\delta, c)$-robust aggregation called SEG-RA:

$$\widetilde{\boldsymbol{x}}^t \ \ = \boldsymbol{x}^t - \gamma_1 \text{RAGG}(\boldsymbol{g}_{\boldsymbol{\xi}_1}^t, \ldots, \boldsymbol{g}_{\boldsymbol{\xi}_n}^t), \ \ \text{where } \boldsymbol{g}_{\boldsymbol{\xi}_i}^t = \boldsymbol{g}_i(\boldsymbol{x}^t, \boldsymbol{\xi}_i^t), \ \ \forall \, i \in \mathcal{G} \ \text{ and } \ \boldsymbol{g}_{\boldsymbol{\xi}_i}^t = * \ \forall \, i \in \mathcal{B},$$

$$\boldsymbol{x}^{t+1} = \boldsymbol{x}^t - \gamma_2 \text{RAGG}(\boldsymbol{g}_{\boldsymbol{\eta}_1}^t, \ldots, \boldsymbol{g}_{\boldsymbol{\eta}_n}^t), \ \ \text{where } \boldsymbol{g}_{\boldsymbol{\eta}_i}^t = \boldsymbol{g}_i(\widetilde{\boldsymbol{x}}^t, \boldsymbol{\eta}_i^t), \ \ \forall \, i \in \mathcal{G} \ \text{ and } \ \boldsymbol{g}_{\boldsymbol{\eta}_i}^t = * \ \forall \, i \in \mathcal{B},$$

where $\{\boldsymbol{g}_{\boldsymbol{\eta}_i}^t\}_{i\in\mathcal{G}}$ and $\{\boldsymbol{g}_{\boldsymbol{\eta}_i}^t\}_{i\in\mathcal{G}}$ are sampled independently. Our main convergence result for SEG-RA is presented in the following theorem; see Appendix D.2 for the proof.

**Theorem 2.** *Let Assumptions[5] 1, 2, 3 and 4 hold. Then after $T$ iterations* SEG-RA *(Algorithm 2) with $(\delta, c)$-RAGG, $\gamma_1 \leq \frac{1}{2\mu + 2L}$ and $\beta = \gamma_2/\gamma_1 \leq 1/4$ outputs $\boldsymbol{x}^T$ such that*

$$\mathbb{E}\|\boldsymbol{x}^T - \boldsymbol{x}^*\|^2 \leq \ \left(1 - \frac{\mu\beta\gamma_1}{4}\right)^T \|\boldsymbol{x}^0 - \boldsymbol{x}^*\|^2 + \frac{8\gamma_1\sigma^2}{\mu\beta G} + 8c\delta(24\sigma^2 + 12\zeta^2)\left(\frac{\gamma_1}{\beta\mu} + \frac{2}{\mu^2}\right).$$

Similar to the case of SGDA-RA, the bound for SEG-RA has the term that cannot be reduced without increasing batchsize even in the homogeneous data case. RDEG, which is also a modification of SEG, has the same linearly convergent term, but SEG-RA has a better dependence on the batchsize, needed to obtain the convergence to any predefined accuracy, that is $\mathcal{O}(\varepsilon^{-1})$ versus $\mathcal{O}(\varepsilon^{-2})$ for RDEG; see Cor. 3.

In heterogeneous case when $\sigma = 0$, SEG-RA also converges linearly to the ball centered at the solution with radius $\mathcal{O}(\sqrt{c\delta}\zeta/\mu)$ that matches the lower bound.

## 2.2  Client Momentum

Next, we focus on the version of SGDA-RA that utilizes worker momentum $\boldsymbol{m}_i^t$, i.e.,

$$\boldsymbol{x}^{t+1} = \boldsymbol{x}^t - \gamma \text{RAGG}(\boldsymbol{m}_1^t, \ldots, \boldsymbol{m}_n^t), \ \ \text{with } \boldsymbol{m}_i^t = (1 - \alpha)\boldsymbol{m}_i^{t-1} + \alpha\boldsymbol{g}_i^t,$$

where $\boldsymbol{g}_i^t = \boldsymbol{g}_i(\boldsymbol{x}^t, \boldsymbol{\xi}_i^t)$, $\forall i \in \mathcal{G}$ and $\boldsymbol{g}_i^t = *$ $\forall \, i \in \mathcal{B}$ and $\{\boldsymbol{g}_{\boldsymbol{\xi}_i}^t\}_{i\in\mathcal{G}}$ are sampled independently. Our main convergence result for this version called M-SGDA-RA is summarized in the following theorem.

**Theorem 3.** *Let Assumptions 1, 2, 4, and 6 hold. Then after $T$ iterations* M-SGDA-RA *(Algorithm 3) with $(\delta, c)$-RAGG outputs $\overline{\boldsymbol{x}}^T$ such that*

$$\mathbb{E}\left[\|\overline{\boldsymbol{x}}^T - \boldsymbol{x}^*\|^2\right] \leq \frac{2\|\boldsymbol{x}^0 - \boldsymbol{x}^*\|^2}{\mu\gamma\alpha W_T} + \frac{8\gamma c\delta(24\sigma^2 + 12\zeta^2)}{\mu\alpha^2} + \frac{6\gamma\sigma^2}{\mu\alpha^2 G} + \frac{4c\delta(24\sigma^2 + 12\zeta^2)}{\mu^2\alpha^2}.$$

*where $\overline{\boldsymbol{x}}^T = \frac{1}{W_T}\sum_{t=0}^T w_t \widehat{\boldsymbol{x}}^t$, $\quad \widehat{\boldsymbol{x}}^t = \frac{\alpha}{1-(1-\alpha)^{t+1}}\sum_{j=0}^t (1-\alpha)^{t-j}\boldsymbol{x}^j, \quad w_t = \left(1 - \frac{\mu\gamma\alpha}{2}\right)^{-t-1}$, and $W_T = \sum_{t=0}^T w_t$.*

---

[5]SGDA-based and SEG-based methods are typically analyzed under different assumptions. Although (SC) follows from (Lip) and (QSM) with $\ell = L^2/\mu$, some operators may satisfy (SC) with significantly smaller $\ell$. Next, when $\mu = 0$, SGDA is not guaranteed to converge [Gidel et al., 2018], while SEG does

Despite the fact that M-SGDA-RA is the first algorithm (for VIs) non-invariant to permutations, it also requires large batches to achieve convergence to any accuracy. Even in the context of minimization, which is much easier than VI, the known SOTA analysis of Momentum-SGD relies **in the convex case** on the unbiasedness of the estimator that is not available due to a robust aggregation. Nevertheless, we prove[6] the convergence to the ball centered at the solution with radius $\mathcal{O}(\sqrt{c\delta}(\zeta+\sigma)/\alpha\mu)$; see Appendix D.3. Moreover, we show that M-SGDA-RA outperforms in the experiments other methods that require large batches.

## 2.3 Random Checks of Computations

We start with the Stochastic Gradient Descent-Accent with Checks of Computations (SGDA-CC). At each iteration of SGDA-CC, the server selects $m$ workers (uniformly at random) and requests them to check the computations of other $m$ workers from the previous iteration. Let $V_t$ be the set of workers that verify/check computations, $A_t$ are active workers at iteration $t$, and $V_t \cap A_t = \varnothing$. Then, the update of SGDA-CC can be written as

$$\boldsymbol{x}^{t+1} = \boldsymbol{x}^t - \gamma\overline{\boldsymbol{g}}^t, \ \ \text{if} \ \ \overline{\boldsymbol{g}}^t = \frac{1}{|A_t|}\sum_{i\in A_t}\boldsymbol{g}_i(\boldsymbol{x}^t,\boldsymbol{\xi}_i^t) \ \ \text{is accepted,}$$

where $\{\boldsymbol{g}_i(\boldsymbol{x}^t,\boldsymbol{\xi}_i^t)\}_{i\in\mathcal{G}}$ are sampled independently.

The acceptance (of the update) event occurs when the condition $\left\|\overline{\boldsymbol{g}}^t - \boldsymbol{g}_i(\boldsymbol{x}^t,\boldsymbol{\xi}_i^t)\right\| \le C\sigma$ holds for the majority of workers. If $\overline{\boldsymbol{g}}^t$ is rejected, then all workers re-sample $\boldsymbol{g}_i(\boldsymbol{x}^t,\boldsymbol{\xi}_i^t)$ until acceptance is achieved. The rejection probability is bounded, as per [Gorbunov et al., 2022b], and can be adjusted by choosing a constant $C = \mathcal{O}(1)$. We assume that the server knows the seeds for generating randomness on workers, and thus, verification of computations is possible. Following each aggregation of $\boldsymbol{g}_i(\boldsymbol{x}^t,\boldsymbol{\xi}_i^t)_{i\in\mathcal{G}}$, the server selects uniformly at random $2m$ workers: $m$ workers check the computations at the previous step of the other $m$ workers. For instance, at the $(t+1)$-th iteration, the server asks a checking peer $i$ to compute $\boldsymbol{g}_j(\boldsymbol{x}^t,\boldsymbol{\xi}_j^t)$, where $j$ is a peer being checked. This is possible if all seeds are broadcasted at the start of the training. Workers assigned to checking do not participate in the training while they check and do not contribute to $\overline{\boldsymbol{g}}^t$. Therefore, each Byzantine peer is checked at each iteration with a probability of $\sim m/n$ by some good worker (see the proof of Theorem 4). If the results are mismatched, then both the checking and checked peers are removed from training.

This design ensures that every such mismatch, whether it is caused by honest or Byzantine peers, eliminates at least one Byzantine peer and at most one honest peer (see details in Appendix E.1). It's worth noting that we assume any information is accessible to Byzantines except when each of them will be checked. As such, Byzantine peers can only reduce their relative numbers, which leads us to the main result for SGDA-CC, which is presented below.

**Theorem 4.** *Let Assumptions 1, 4 and 6 hold. Then after $T$ iterations* SGDA-CC *(Algorithm 5) with $\gamma \le \frac{1}{2\ell}$ outputs $\boldsymbol{x}^T$ such that*

$$\mathbb{E}\left\|\boldsymbol{x}^{T+1} - \boldsymbol{x}^*\right\|^2 \le \left(1 - \frac{\gamma\mu}{2}\right)^{T+1}\left\|\boldsymbol{x}^0 - \boldsymbol{x}^*\right\|^2 + \frac{4\gamma\sigma^2}{\mu(n-2B-m)} + \frac{2q\sigma^2 nB}{m}\left(\frac{\gamma}{\mu}+\gamma^2\right),$$

*where $q = 2C^2 + 12 + \frac{12}{n-2B-m}$; $q = \mathcal{O}(1)$ since $C = \mathcal{O}(1)$.*

The above theorem (see Appendix E.1 for the proof) provides the first result that does not require large batchsizes to converge to any predefined accuracy. The first and the second terms in the convergence bound correspond to the SOTA results for SGDA [Loizou et al., 2021]. Similarly to the vanilla SGDA, the convergence can be obtained by decreasing stepsize, however, such an approach does not benefit from collaboration, since the dominating term $\frac{\gamma\sigma^2 nB}{\mu m}$ (coming from the presence of Byzantine workers) is not inversely dependent on $n$. Moreover, the result is even worse than for single node SGDA in terms of dependence on $n$.

---

[6]In contrast to Theorems 1-2, the result from Theorem 3 is given for the averaged iterate. We consider the averaged iterate to make the analysis simpler. We believe that one can extend the analysis to the last iterate as well, but we do not do it since we expect that the same problem (the need for large batches) will remain in the last-iterate analysis.

To overcome this issue we consider the restart technique for SGDA-CC and propose the next algorithm called R-SGDA-CC. This method consists of $r$ stages. On the $t$-th stage R-SGDA-CC runs SGDA-CC with $\gamma_t$ for $K_t$ iterations from the starting point $\widehat{\boldsymbol{x}}^t$, which is the output from the previous stage, and defines the obtained point as $\widehat{\boldsymbol{x}}^{t+1}$ (see details in Appendix E.2). The main result for R-SGDA-CC is given below.

**Theorem 5.** *Let Assumptions 1, 4 and 6 hold. Then, after* $r = \left\lceil \log_2 \frac{R^2}{\varepsilon} \right\rceil - 1$ *restarts* R-SGDA-CC *(Algorithm 6) with* $\gamma_t = \min\left\{ \frac{1}{2\ell}, \sqrt{\frac{(n-2B-m)R^2}{6\sigma^2 2^t K_t}}, \sqrt{\frac{m^2 R^2}{72q\sigma^2 2^t B^2 n^2}} \right\}$ *and*
$K_t = \left\lceil \max\left\{ \frac{8\ell}{\mu}, \frac{96\sigma^2 2^t}{(n-2B-m)\mu^2 R^2}, \frac{34n\sigma B \sqrt{q2^t}}{m\mu R} \right\} \right\rceil$, *where* $R \geq \|\boldsymbol{x}^0 - \boldsymbol{x}^*\|$, *outputs* $\widehat{\boldsymbol{x}}^r$ *such that*
$\mathbb{E}\|\widehat{\boldsymbol{x}}^r - \boldsymbol{x}^*\|^2 \leq \varepsilon$. *Moreover, the total number of executed iterations of* SGDA-CC *is*

$$\sum_{t=1}^{r} K_t = \mathcal{O}\left( \frac{\ell}{\mu} \log \frac{\mu R_0^2}{\varepsilon} + \frac{\sigma^2}{(n-2B-m)\mu\varepsilon} + \frac{nB\sigma}{m\sqrt{\mu\varepsilon}} \right). \tag{5}$$

The above result implies that R-SGDA-CC also converges to any accuracy without large batch-sizes (see Appendix E.2 for details). However, as the accuracy tends to zero, the dominant term $\frac{\sigma^2}{(n-2B-m)\mu\varepsilon}$ inversely depends on the number of workers. This makes R-SGDA-CC benefit from collaboration, as the algorithm becomes more efficient with an increasing number of workers. Moreover, when $B$ and $m$ are small the derived complexity result for R-SGDA-CC matches the one for parallel SGDA [Loizou et al., 2021], which is obtained for the case of no Byzantine workers.

Next, we present a modification of Stochastic Extragradient with Checks of Computations (SEG-CC):

$$\widetilde{\boldsymbol{x}}^t = \boldsymbol{x}^t - \gamma_1 \overline{\boldsymbol{g}}_{\boldsymbol{\xi}}^t, \qquad \text{if } \overline{\boldsymbol{g}}_{\boldsymbol{\xi}}^t = \frac{1}{|A_t|} \sum_{i \in A_t} \boldsymbol{g}_i(\boldsymbol{x}^t, \boldsymbol{\xi}_i^t) \text{ is accepted,}$$

$$\boldsymbol{x}^{t+1} = \boldsymbol{x}^t - \gamma_2 \overline{\boldsymbol{g}}_{\boldsymbol{\eta}}^t, \qquad \text{if } \overline{\boldsymbol{g}}_{\boldsymbol{\eta}}^t = \frac{1}{|A_t|} \sum_{i \in A_t} \boldsymbol{g}_i(\widetilde{\boldsymbol{x}}^t, \boldsymbol{\eta}_i^t) \text{ is accepted,}$$

where $\{\boldsymbol{g}_i(\boldsymbol{x}^t, \boldsymbol{\xi}_i^t)\}_{i \in \mathcal{G}}$ and $\{\boldsymbol{g}_i(\widetilde{\boldsymbol{x}}^t, \boldsymbol{\eta}_i^t)\}_{i \in \mathcal{G}}$ are sampled independently. The events of acceptance $\overline{\boldsymbol{g}}_{\boldsymbol{\eta}}^t$ (or $\overline{\boldsymbol{g}}_{\boldsymbol{\xi}}^t$) happens if

$$\left\| \overline{\boldsymbol{g}}^t - \boldsymbol{g}_i(\boldsymbol{x}^t, \boldsymbol{\xi}_i^t) \right\| \leq C\sigma \quad \left( \text{or } \left\| \overline{\boldsymbol{g}}_{\boldsymbol{\eta}}^t - \boldsymbol{g}_i(\widetilde{\boldsymbol{x}}^t, \boldsymbol{\eta}_i^t) \right\| \leq C\sigma \right)$$

holds for the majority of workers. An iteration of SEG-CC actually represents two subsequent iteration of SGDA-CC, so we refer to the beginning of the section for more details. Our main convergence results for SEG-CC are summarized in the following theorem; see Appendix E.3 for the proof.

**Theorem 6.** *Let Assumptions 1, 3 and 4 hold. Then after $T$ iterations* SEG-CC *(Algorithm 7) with* $\gamma_1 \leq \frac{1}{2\mu+2L}$ *and* $\beta = \gamma_2/\gamma_1 \leq 1/4$ *outputs* $\boldsymbol{x}^T$ *such that*

$$\mathbb{E}\|\boldsymbol{x}^T - \boldsymbol{x}^*\|^2 \leq \left(1 - \frac{\mu\beta\gamma_1}{4}\right)^T \|\boldsymbol{x}^0 - \boldsymbol{x}^*\|^2 + 2\sigma^2 \left( \frac{4\gamma_1}{\beta\mu^2(n-2B-m)} + \frac{\gamma_1 qnB}{m} \right),$$

*where* $q = 2C^2 + 12 + \frac{12}{n-2B-m}$; $q = \mathcal{O}(1)$ *since* $C = \mathcal{O}(1)$.

Similarly to SGDA-CC, SEG-CC does not require large batchsizes to converge to any predefined accuracy and does not benefit of collaboration, though the first two terms correspond to the SOTA convergence results for SEG under bounded variance assumption [Juditsky et al., 2011]. The last term appears due to the presence of the Byzantine workers. The restart technique can also be applied; see Appendix E.4 for the proof.

**Theorem 7.** *Let Assumptions 1, 3, 4 hold. Then, after* $r = \left\lceil \log_2 \frac{R^2}{\varepsilon} \right\rceil - 1$ *restarts* R-SEG-CC *(Algotithm 8) with* $\gamma_{1_t} = \min\left\{ \frac{1}{2L}, \sqrt{\frac{(G-B-m)R^2}{16\sigma^2 2^t K_t}}, \sqrt{\frac{mR^2}{8q\sigma^2 2^t Bn}} \right\}$, $\gamma_{2_t} = \min\left\{ \frac{1}{4L}, \sqrt{\frac{m^2 R^2}{64q\sigma^2 2^t B^2 n^2}}, \sqrt{\frac{(G-B-m)R^2}{64\sigma^2 K_t}} \right\}$ *and* $K_t = \left\lceil \max\left\{ \frac{8L}{\mu}, \frac{16n\sigma B\sqrt{q2^t}}{m\mu R}, \frac{256\sigma^2 2^t}{(G-B-m)\mu^2 R^2} \right\} \right\rceil$, *where* $R \geq \|\boldsymbol{x}^0 - \boldsymbol{x}^*\|$ *outputs* $\widehat{\boldsymbol{x}}^r$ *such that* $\mathbb{E}\|\widehat{\boldsymbol{x}}^r - \boldsymbol{x}^*\|^2 \leq \varepsilon$. *Moreover, the total number of executed iterations of* SEG-CC *is*

$$\sum_{t=1}^{r} K_t = \mathcal{O}\left( \frac{\ell}{\mu} \log \frac{\mu R_0^2}{\varepsilon} + \frac{\sigma^2}{(n-2B-m)\mu\varepsilon} + \frac{nB\sigma}{m\sqrt{\mu\varepsilon}} \right). \tag{6}$$

The above result states that R-SEG-CC also converges to any accuracy without large batchsizes; see Appendix E.4. But with accuracy tending to zero ($\varepsilon \to 0$) the dominating term $\frac{\sigma^2}{(n-2B-m)\mu\varepsilon}$ inversely depends on the number of workers, hence R-SEG-CC benefits from collaboration. Moreover, when $B$ and $m$ are small the derived complexity result for R-SEG-CC matches the one for parallel/mini-batched SEG [Juditsky et al., 2011], which is obtained for the case of no Byzantine workers.

## 3 Numerical Experiments

**Quadratic game.**  To illustrate our theoretical results, we conduct numerical experiments on a quadratic game

$$\min_y \max_z \frac{1}{s} \sum_{i=1}^{s} \frac{1}{2} y^\top \mathbf{A}_{1,i} y + y^\top \mathbf{A}_{2,i} z - \frac{1}{2} z^\top \mathbf{A}_{3,i} z + b_{1,i}^\top y - b_{2,i}^\top z.$$

The above problem can be re-formulated as a special case of (1) with $F$ defined as follows:

$$F(\boldsymbol{x}) = \frac{1}{s} \sum_{i=1}^{s} \mathbf{A}_i \boldsymbol{x} + b_i, \quad \text{where } \boldsymbol{x} = (y^\top, z^\top)^\top, \ b_i = (b_{1,i}^\top, b_{2,i}^\top)^\top, \tag{7}$$

with symmetric matrices $\mathbf{A}_{j,i}$ s.t. $\mu \mathbf{I} \preccurlyeq \mathbf{A}_{j,i} \preccurlyeq \ell \mathbf{I}$, $\mathbf{A}_i \in \mathbb{R}^{d \times d}$ and $b_i \in \mathbb{R}^d$; see Appendix F for the detailed description.

We set $\ell = 100$, $\mu = 0.1$, $s = 1000$ and $d = 50$. Only one peer checked the computations on each iteration ($m = 1$). We used RFA (geometric median) with bucketing as an aggregator since it showed the best performance. For approximating the median we used Weiszfeld's method with 10 iterations and parameter $\nu = 0.1$ [Pillutla et al., 2022]. RDEG [Adibi et al., 2022] provably works only if $n \geq 100$, so here we provide experiments with $n = 150$, $B = 20$, $\gamma = 2e-5$. We set the parameter $\alpha = 0.1$ for M-SGDA-RA, and the following parameters for RDEG: $\alpha_{\text{RDEG}} = 0.06, \delta_{\text{RDEG}} = 0.9$ and theoretical value of $\epsilon$; see Appendix F for more experiments. We tested the algorithms under the following attacks: bit flipping (BF), random noise (RN), inner product manipulation (IPM) Xie et al. [2019] and "a little is enough" (ALIE) Baruch et al. [2019].

**Robust Neural Networks training.**  Let $f(u; x, y)$ be the loss function of a neural network with parameters $u \in \mathbb{R}^d$ given input $x \in \mathbb{R}^m$ and label $y$. For example, in our experiments, we let $f$ be the cross entropy loss, and $\{(x_i, y_i)\}_1^N$ is the MNIST dataset. Now consider the following objective:

$$\min_{u \in \mathbb{R}^d} \max_{v \in \mathbb{R}^m} \frac{1}{N} \sum_{i=1}^{N} f(u; x_i + v, y_i) + \frac{\lambda_1}{2} \|u\|_2^2 - \frac{\lambda_2}{2} \|v\|_2^2. \tag{8}$$

This min-max objective adds an extra adversarial noise variable to the input data such that it maximizes the loss, so the neural network should become robust to such noise as it minimizes the loss. We can reformulate this objective as a variational inequality with

$$\boldsymbol{x} = \begin{pmatrix} u \\ v \end{pmatrix}, \quad F_i(\boldsymbol{x}) = \begin{pmatrix} \nabla_u f(u; x_i + v, y_i) + \lambda_1 u \\ -\nabla_v f(u; x_i + v, y_i) + \lambda_2 v \end{pmatrix}, \quad F(\boldsymbol{x}) = \frac{1}{N} \sum_{i=1}^{N} F_i(\boldsymbol{x}). \tag{9}$$

We let $n = 20$, $B = 4$, $\lambda_1 = 0$, and $\lambda_2 = 100$. We fix the learning rate to 0.01 and use a batch size of 32. We run the algorithm for 50 epochs and average our results across 3 runs. We test the algorithms under the following attacks: i) bit flipping (BF), ii) label flipping (LF), iii) inner product manipulation (IPM) Xie et al. [2019], and iv) a little is enough (ALIE) Baruch et al. [2019]. We compare our algorithm SGDA-CC against the following algorithms: i) SGDA-RA, ii) M-SGDA-RA, and iii) RDEG Adibi et al. [2022]. We use RFA with bucket size 2 as the robust aggregator. The results are shown in Figure 2. Specifically, we show the validation error on MNIST after each epoch. We can see that SGDA-CC performs the best, followed closely by M-SGDA-RA.

## 4 Conclusion

This paper proposes several new algorithms for Byzantine-robust distributed variational inequalities and provides rigorous theoretical convergence analysis for the developed methods. In particular, we

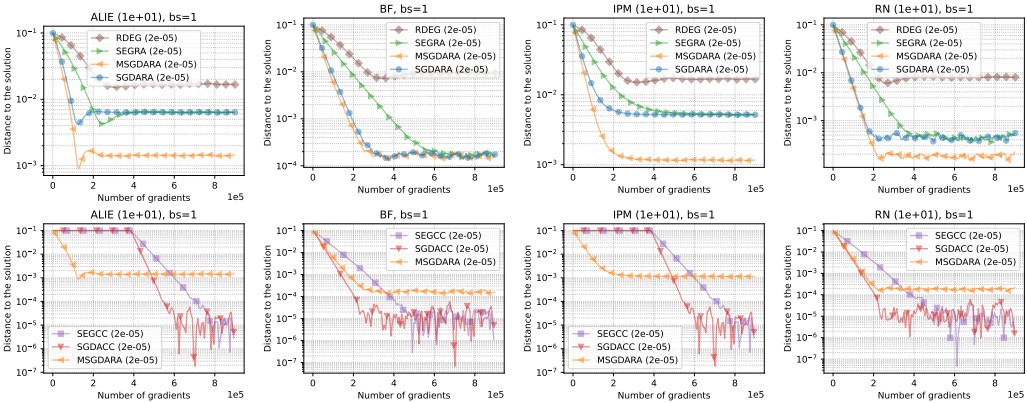

Figure 1: Error plots for quadratic games experiments under different Byzantine attacks. The first row shows the outperformance of M-SGDA-RA over methods without checks of computations. The second row illustrates advantages of SGDA-CC and SEG-CC.

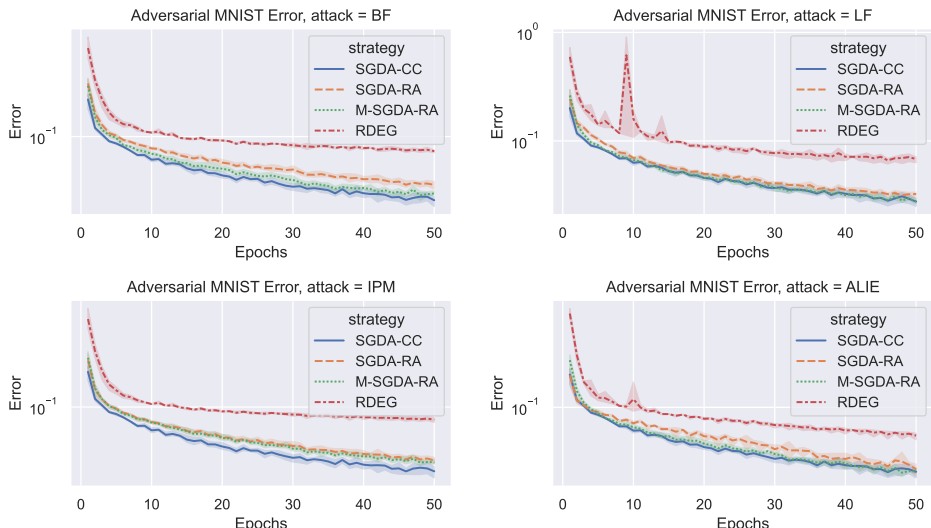

Figure 2: Error plots for the robust neural network experiment on MNIST under different byzantine attacks (BF, LF, IPM, and ALIE). Each algorithm is shown with a consistent choice of color and style across plots, as indicated in the legends.

propose the first methods in this setting that provably converge to any predefined accuracy in the case of homogeneous data. We believe this is an important step towards building a strong theory of Byzantine robustness in the case of distributed VIs.

However, our work has several limitations. First of all, one can consider different/more general assumptions about operators [Beznosikov et al., 2023, Gorbunov et al., 2022a, 2023b] in the analysis of the proposed methods. Next, as we mention in the discussion after Theorem 3, our result for M-SGDA-RA requires large batchsizes, and it remains unclear to us whether this requirement can be removed. Finally, the only results that do not require large batchsizes are derived using the checks of computations that create (although small) computation overhead. Obtaining similar results without checks of computations remains an open problem. Addressing these limitations is a prominent direction for future research.

## Acknowledgments and Disclosure of Funding

This work of N. Tupitsa was supported by a grant for research centers in the field of artificial intelligence, provided by the Analytical Center for the Government of the Russian Federation in accordance with the subsidy agreement (agreement identifier 000000D730321P5Q0002) and the agreement with the Moscow Institute of Physics and Technology dated November 1, 2021 No. 70-2021-00138.

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

# Contents

# A  Examples of $(\delta, c)$-Robust Aggregation Rules

This section is about how to construct an aggregator satisfying 1.1.

## A.1  Aggregators

This subsection examines various aggregators that lack robustness. It means that new attacks can be easily designed to exploit the aggregation scheme, causing its failure. We analyze three commonly employed defenses that are representative.

**Krum.** For $i \neq j$, let $i \to j$ denote that $\boldsymbol{x}_j$ belongs to the $n - q - 2$ closest vectors to $\boldsymbol{x}_i$. Then,

$$\text{KRUM}(\boldsymbol{x}_1, \ldots, \boldsymbol{x}_n) := \operatorname*{argmin}_i \sum_{i \to j} \|\boldsymbol{x}_i - \boldsymbol{x}_j\|^2 .$$

Krum is computationally expensive, requiring $\mathcal{O}(n^2)$ work by the server Blanchard et al. [2017].
**CM.** Coordinate-wise median computes for the $k$-th coordinate:

$$[\text{CM}(\boldsymbol{x}_1, \ldots, \boldsymbol{x}_n)]_k := \text{median}([\boldsymbol{x}_1]_k, \ldots, [\boldsymbol{x}_n]_k) = \operatorname*{argmin}_i \sum_{j=1}^n |[\boldsymbol{x}_i]_k - [\boldsymbol{x}_j]_k| .$$

Coordinate-wise median is fast to implement requiring only $\mathcal{O}(n)$ time Chen et al. [2017].

**RFA.** Robust federated averaging (RFA) computes the geometric median

$$\text{RFA}(\boldsymbol{x}_1, \ldots, \boldsymbol{x}_n) := \operatorname*{argmin}_{\boldsymbol{v}} \sum_{i=1}^n \|\boldsymbol{v} - \boldsymbol{x}_i\|_2 .$$

Although there is no closed form solution for the geometric median, an approximation technique presented by Pillutla et al. [2022] involves performing several iterations of the smoothed Weiszfeld algorithm, with each iteration requiring a computation of complexity $\mathcal{O}(n)$.

## A.2  Bucketing algorithm

We use the process of *s-bucketing*, propose by [Yang and Li, 2021, Karimireddy et al., 2022] to randomly divide $n$ inputs, $\boldsymbol{x}_1$ to $\boldsymbol{x}_n$, into $\lceil n/s \rceil$ buckets, each containing no more than $s$ elements. After averaging the contents of each bucket to create $\boldsymbol{y}_1, \ldots, \boldsymbol{y}_{\lceil n/s \rceil}$, we input them into the aggregator AGGR. The Bucketing Algorithm outlines the procedure. Our approach's main feature is that the resulting set of averaged $\boldsymbol{y}_1, \ldots, \boldsymbol{y}_{\lceil n/s \rceil}$ are more homogeneous (with lower variance) than the original inputs.

---

**Algorithm** Bucketing Algorithm

1: **input** $\{\boldsymbol{x}_1, \ldots, \boldsymbol{x}_n\}$, $s \in \mathbb{N}$, aggregation rule AGGR
2: pick random permutation $\pi$ of $[n]$
3: compute $\boldsymbol{y}_i \leftarrow \frac{1}{s} \sum_{k=(i-1) \cdot s+1}^{\min(n, i \cdot s)} \boldsymbol{x}_{\pi(k)}$ for $i = \{1, \ldots, \lceil n/s \rceil\}$
4: **output** $\widehat{\boldsymbol{x}} \leftarrow \text{AGGR}(\boldsymbol{y}_1, \ldots, \boldsymbol{y}_{\lceil n/s \rceil})$           // aggregate after bucketing

---

## A.3  Robust Aggregation examples

Next we recall the result from [Karimireddy et al., 2022], that shows that aggregators which we saw, can be made to satisfy 1.1 by combining with bucketing.

**Theorem 8.** *Suppose we are given $n$ inputs $\{\boldsymbol{x}_1, \ldots, \boldsymbol{x}_n\}$ such that $\mathbb{E}\|\boldsymbol{x}_i - \boldsymbol{x}_j\|^2 \leq \rho^2$ for any fixed pair $i, j \in \mathcal{G}$ and some $\rho \geq 0$ for some $\delta \leq \delta_{\max}$, with $\delta_{\max}$ to be defined. Then, running Bucketing Algorithm with $s = \lfloor \frac{\delta_{\max}}{\delta} \rfloor$ yields the following:*

- *Krum:* $\mathbb{E}\|\text{KRUM} \circ \text{BUCKETING}(\boldsymbol{x}_1, \ldots, \boldsymbol{x}_n) - \bar{\boldsymbol{x}}\|^2 \leq \mathcal{O}(\delta\rho^2)$ *with* $\delta_{\max} < \frac{1}{4}$.

- *Geometric median:* $\mathbb{E}\|\text{RFA} \circ \text{BUCKETING}(\boldsymbol{x}_1, \ldots, \boldsymbol{x}_n) - \bar{\boldsymbol{x}}\|^2 \leq \mathcal{O}(\delta\rho^2)$ *with* $\delta_{\max} < \frac{1}{2}$.

- *Coordinate-wise median:* $\mathbb{E}\|\mathrm{CM} \circ \mathrm{BUCKETING}(\boldsymbol{x}_1, \ldots, \boldsymbol{x}_n) - \bar{\boldsymbol{x}}\|^2 \leq \mathcal{O}(d\delta\rho^2)$ *with* $\delta_{\max} < \frac{1}{2}$.

Note that all these methods satisfy the notion of an *agnostic* Byzantine robust aggregator (Definition 1.1).

# B  Further Details on RDEG

Originally RDEG was proposed for min-max problems and represents a variation of SEG with Univariate Trimmed-Mean Estimator aggregation rule. For convenience we give here RDEG pseudo-code we used in experiments.

In this section we use the notation $\pi \in (0,1)$ for a confidence level.

---

**Robust Distributed Extra-Gradient** (RDEG)

---

**Input:** $\text{TRIM}_{\epsilon,\alpha,\delta}$, $\gamma$
1: **for** $t = 1, \ldots$ **do**
2:      **for** worker $i \in [n]$ **in parallel**
3:          $\boldsymbol{g}^t_{\boldsymbol{\xi}_i} \leftarrow \boldsymbol{g}_i(\boldsymbol{x}^t, \boldsymbol{\xi}_i)$
4:          **send** $\boldsymbol{g}^t_{\boldsymbol{\xi}_i}$ if $i \in \mathcal{G}$, else **send** $*$ if Byzantine
5:      $\widehat{\boldsymbol{g}}_{\boldsymbol{\xi}^t}(\boldsymbol{x}^t) = \text{TRIM}_{\epsilon,\alpha,\delta}\,(\boldsymbol{g}^t_{\boldsymbol{\xi}_1}, \ldots, \boldsymbol{g}^t_{\boldsymbol{\xi}_n})$
6:      $\widetilde{\boldsymbol{x}}^t \leftarrow \boldsymbol{x}^t - \gamma_1 \widehat{\boldsymbol{g}}_{\boldsymbol{\xi}^t}(\boldsymbol{x}^t)$.
7:      **for** worker $i \in [n]$ **in parallel**
8:          $\boldsymbol{g}^t_{\boldsymbol{\eta}_i} \leftarrow \boldsymbol{g}_i(\widetilde{\boldsymbol{x}}^t, \boldsymbol{\eta}_i)$
9:          **send** $\boldsymbol{g}^t_{\boldsymbol{\eta}_i}$ if $i \in \mathcal{G}$, else **send** $*$ if Byzantine
10:     $\widehat{\boldsymbol{g}}_{\boldsymbol{\eta}^t}(\widetilde{\boldsymbol{x}}^t) = \text{TRIM}_{\epsilon,\alpha,\delta}\,(\boldsymbol{g}^t_{\boldsymbol{\eta}_1}, \ldots, \boldsymbol{g}^t_{\boldsymbol{\eta}_n})$
11:     $\boldsymbol{x}^{t+1} \leftarrow \boldsymbol{x}^t - \gamma_2 \widehat{\boldsymbol{g}}_{\boldsymbol{\eta}^t}(\widetilde{\boldsymbol{x}}^t)$.

---

**Performance of Univariate Trimmed-Mean Estimator.** The TRIM operator takes as input $n$ vectors, and applies coordinatewisely the univariate trimmed mean estimator from Lugosi and Mendelson [2021], described bellow here as Univariate Trimmed-Mean Estimator Algorithm.

---

**Univariate Trimmed-Mean Estimator Algorithm** Lugosi and Mendelson [2021]

---

**Input:** Corrupted data set $Z_1, \ldots, Z_{n/2}, \widetilde{Z}_1, \ldots \widetilde{Z}_{n/2}$, corruption fraction $\delta$, and confidence level $\pi$.
1: Set $\epsilon = 8\delta + 24\frac{\log(4/\pi)}{n}$.
2: Let $Z_1^* \leq Z_2^* \leq \cdots \leq Z_{n/2}^*$ represent a non-decreasing arrangement of $\{Z_i\}_{i \in [n/2]}$. Compute quantiles: $\gamma = Z_{\epsilon n/2}^*$ and $\beta = Z_{(1-\epsilon)n/2}^*$.
3: Compute robust mean estimate $\widehat{\mu}_Z$ as follows:

$$\widehat{\mu}_Z = \frac{2}{n} \sum_{i=1}^{n/2} \phi_{\gamma,\beta}(\widetilde{Z}_i); \phi_{\gamma,\beta}(x) = \begin{cases} \beta & x > \beta \\ x & x \in [\gamma, \beta] \\ \gamma & x < \gamma \end{cases}$$

---

The following result on the performance of Univariate Trimmed-Mean Estimator plays a key role in the analysis of RDEG.

**Theorem.** *[Adibi et al., 2022, Theorem 1] Consider the trimmed mean estimator. Suppose $\delta \in [0, 1/16)$, and let $\pi \in (0,1)$ be such that $\pi \geq 4e^{-n/2}$. Then, there exists an universal constant $c$, such that with probability at least $1 - \pi$,*

$$|\widehat{\mu}_Z - \mu_Z| \leq c\sigma_Z \left( \sqrt{\delta} + \sqrt{\frac{\log(1/\pi)}{n}} \right).$$

Using the latter componentwise result the authors states that

$$\|\widehat{\boldsymbol{g}}_{\boldsymbol{\xi}^t}(\boldsymbol{x}^t) - F(\boldsymbol{x}^t)\| \leq c\sigma \left( \sqrt{\delta} + \sqrt{\frac{\log(1/\pi)}{n}} \right).$$

In fact this result is very similar to the Definition 1.1. The main difference is that for Univariate Trimmed-Mean Estimator we have a bound with some probability. The other difference that using

the following representation of the result with $\rho^2 = c^2 \sigma^2$

$$\|\widehat{\boldsymbol{g}}_{\boldsymbol{\xi}^t}\left(\boldsymbol{x}^t\right) - F(\boldsymbol{x}^t)\|^2 \leq \delta\rho^2 + \frac{\rho^2 \log(1/\pi)}{n}, \quad \text{w.p. } 1 - \pi$$

Univariate Trimmed-Mean Estimator has the additional term inversely depending on the number of workers.

Moreover, the result requires $\delta \in [0, 1/16)$ in contrast to the aggregators we used, that work for wider range of corruption level $\delta \in [0, 1/5]$.

**Performance guarantees for** RDEG. The authors of [Adibi et al., 2022] consider only homogeneous case.

**Theorem.** *[Adibi et al., 2022, Theorem 3] Suppose Assumptions 3 and 4 hold in conjunction with the assumptions on $\delta$ and $n$: the fraction $\delta$ of corrupted devices satisfies $\delta \in [0, 1/16)$, and the number of agents $n$ is sufficiently large: $n \geq 48 \log(16dT^2)$. Then, with $\pi = 1/(4dT^2)$ and step-size $\eta \leq 1/(4L)$,* RDEG *guarantees the following with probability at least $1 - 1/T$:*

$$\|\boldsymbol{x}^* - \boldsymbol{x}^{T+1}\|^2 \leq 2e^{-\frac{T}{4\kappa}} R^2 + \frac{8c\sigma R\kappa}{L} \left( \sqrt{\delta} + \sqrt{\frac{\log(4dT^2)}{n}} \right), \tag{10}$$

*where $\kappa = \mu/L$.*

The result implies that RDEG benefits of collaboration only when the corruption level is small. In fact, the term $\frac{\log(4dT^2)}{n} \leq \frac{\log(4dT^2)}{48 \log(16dT^2)} \leq 1/48$, so the corruption level should be less than $1/48$ to make RDEG significantly benefit of collaboration in contrast to our SEG-CC that requires corruption level only less than $1/5$. Moreover, in case of larger corruption level, RDEG converges to a ball centered at the solution with radius $\widetilde{\mathcal{O}}\left( \sqrt{\frac{\sqrt{\delta}\sigma R\kappa}{L}} \right)$ in contrast to our methods SGDA-RA, SEG-RA and M-SGDA-RA converge to a ball centered at the solution with radius $\widetilde{\mathcal{O}}\left( \sqrt{\frac{c\delta\sigma^2}{\mu^2}} \right)$, that has a better dependence on $\sigma$. It is crucial with increasing batchsize ($b$ = batchsize), since $\sigma^2$ depends on a batchsize as $\frac{1}{b}$.

# C  Auxilary results

## C.1  Basic Inequalities

For all $a, b \in \mathbb{R}^n$ and $\lambda > 0, q \in (0, 1]$

$$|\langle a, b \rangle| \leq \frac{\|a\|_2^2}{2\lambda} + \frac{\lambda \|b\|_2^2}{2}, \tag{11}$$

$$\|a + b\|_2^2 \leq 2\|a\|_2^2 + 2\|b\|_2^2, \tag{12}$$

$$\|a + b\|^2 \leq (1 + \lambda)\|a\|^2 + \left(1 + \frac{1}{\lambda}\right)\|b\|^2, \tag{13}$$

$$\langle a, b \rangle = \frac{1}{2}\left(\|a + b\|_2^2 - \|a\|_2^2 - \|b\|_2^2\right), \tag{14}$$

$$\langle a, b \rangle = \frac{1}{2}\left(-\|a - b\|_2^2 + \|a\|_2^2 + \|b\|_2^2\right), \tag{15}$$

$$\left\|\sum_{i=1}^n a_i\right\|^2 \leq n \sum_{i=1}^n \|a_i\|^2, \tag{16}$$

$$\|a + b\|^2 \geq \frac{1}{2}\|a\|^2 - \|b\|^2, \tag{17}$$

$$\left(1 - \frac{q}{2}\right)^{-1} \leq 1 + q, \tag{18}$$

$$\left(1 + \frac{q}{2}\right)(1 - q) \leq 1 - \frac{q}{2}. \tag{19}$$

## C.2  Usefull Lemmas

We write $\boldsymbol{g}_i^t$ or simply $\boldsymbol{g}_i$ instead of $\boldsymbol{g}_i(\boldsymbol{x}^t, \boldsymbol{\xi}_i^t)$ when there is no ambiguity.

**Lemma C.1.** *Suppose that the operator $F$ is given in the form* (1) *and Assumptions* 1 *and* 6 *hold. Then*

$$\mathbb{E}_{\boldsymbol{\xi}}\|\overline{\boldsymbol{g}}(\boldsymbol{x}, \boldsymbol{\xi})\|^2 \leq \ell\left\langle F(\boldsymbol{x}), \boldsymbol{x} - \boldsymbol{x}^*\right\rangle + \frac{\sigma^2}{G},$$

*where $\mathbb{E}_{\boldsymbol{\xi}} := \Pi_i \mathbb{E}_{\boldsymbol{\xi}_i}$ and $\overline{\boldsymbol{g}}(\boldsymbol{x}, \boldsymbol{\xi}) = \frac{1}{G}\sum_{i \in \mathcal{G}} \boldsymbol{g}_i(\boldsymbol{x}; \boldsymbol{\xi}_i)$.*

*Proof of Lemma C.1.* First of one can decomposeda squared norm of a difference and obtain

$$\mathbb{E}_{\boldsymbol{\xi}}\|\overline{\boldsymbol{g}}(\boldsymbol{x}, \boldsymbol{\xi}) - F(\boldsymbol{x})\|^2 = \mathbb{E}_{\boldsymbol{\xi}}\|\overline{\boldsymbol{g}}(\boldsymbol{x}, \boldsymbol{\xi})\|^2 - 2\langle \mathbb{E}_{\boldsymbol{\xi}}\overline{\boldsymbol{g}}(\boldsymbol{x}, \boldsymbol{\xi}), F(\boldsymbol{x})\rangle + \|F(\boldsymbol{x})\|^2.$$

Since $\overline{\boldsymbol{g}}(\boldsymbol{x}, \boldsymbol{\xi}) = \frac{1}{G}\sum_{i \in \mathcal{G}}\boldsymbol{g}_i(\boldsymbol{x}; \boldsymbol{\xi}_i)$, by the definition (1) of $F$ and by Assumption 1 one has

$$\mathbb{E}_{\boldsymbol{\xi}}\overline{\boldsymbol{g}}(\boldsymbol{x}, \boldsymbol{\xi}) = \frac{1}{G}\sum_{i \in \mathcal{G}}\mathbb{E}_{\boldsymbol{\xi}_i}\boldsymbol{g}_i(\boldsymbol{x}, \boldsymbol{\xi}_i) = \frac{1}{G}\sum_{i \in \mathcal{G}}F_i(\boldsymbol{x}) = F(x),$$

and consequently

$$\mathbb{E}_{\boldsymbol{\xi}}\|\overline{\boldsymbol{g}}(\boldsymbol{x}, \boldsymbol{\xi})\|^2 = \mathbb{E}_{\boldsymbol{\xi}}\|\overline{\boldsymbol{g}}(\boldsymbol{x}, \boldsymbol{\xi}) + F(\boldsymbol{x})\|^2 - \|F(\boldsymbol{x})\|^2. \tag{20}$$

One can bound $\mathbb{E}_{\boldsymbol{\xi}}\|\overline{\boldsymbol{g}}(\boldsymbol{x}, \boldsymbol{\xi}) - F(\boldsymbol{x})\|^2$ as

$$\mathbb{E}_{\boldsymbol{\xi}}\|\overline{\boldsymbol{g}}(\boldsymbol{x}, \boldsymbol{\xi}) - F(\boldsymbol{x})\|^2 = \mathbb{E}_{\boldsymbol{\xi}}\left\|\frac{1}{G}\sum_{i \in \mathcal{G}}(\boldsymbol{g}_i(\boldsymbol{x}; \boldsymbol{\xi}_i) - F_i(\boldsymbol{x}))\right\|^2$$

$$\stackrel{\text{independence of } \boldsymbol{\xi}_i}{=} \frac{1}{G^2}\sum_{i \in \mathcal{G}}\mathbb{E}_{\boldsymbol{\xi}_i}\|\boldsymbol{g}_i(\boldsymbol{x}; \boldsymbol{\xi}_i) - F_i(\boldsymbol{x})\|^2 \leq \frac{\sigma^2}{G},$$

where the last inequality of the above chain follows from (SC). The above chain together with (20) and (SC) implies the statement of the theorem. $\qquad\square$

**Lemma C.2.** *Let $K > 0$ be a positive integer and $\eta_1, \eta_2, \ldots, \eta_K$ be random vectors such that $\mathbb{E}_k[\eta_k] \overset{def}{=} \mathbb{E}[\eta_k \mid \eta_1, \ldots, \eta_{k-1}] = 0$ for $k = 2, \ldots, K$. Then*

$$\mathbb{E}\left[\left\|\sum_{k=1}^{K} \eta_k\right\|^2\right] = \sum_{k=1}^{K} \mathbb{E}[\|\eta_k\|^2]. \tag{21}$$

*Proof.* We start with the following derivation:

$$
\begin{aligned}
\mathbb{E}\left[\left\|\sum_{k=1}^{K} \eta_k\right\|^2\right] &= \mathbb{E}[\|\eta_K\|^2] + 2\mathbb{E}\left[\left\langle \eta_K, \sum_{k=1}^{K-1} \eta_k \right\rangle\right] + \mathbb{E}\left[\left\|\sum_{k=1}^{K-1} \eta_k\right\|^2\right] \\
&= \mathbb{E}[\|\eta_K\|^2] + 2\mathbb{E}\left[\mathbb{E}_K\left[\left\langle \eta_K, \sum_{k=1}^{K-1} \eta_k \right\rangle\right]\right] + \mathbb{E}\left[\left\|\sum_{k=1}^{K-1} \eta_k\right\|^2\right] \\
&= \mathbb{E}[\|\eta_K\|^2] + 2\mathbb{E}\left[\left\langle \mathbb{E}_K[\eta_K], \sum_{k=1}^{K-1} \eta_k \right\rangle\right] + \mathbb{E}\left[\left\|\sum_{k=1}^{K-1} \eta_k\right\|^2\right] \\
&= \mathbb{E}[\|\eta_K\|^2] + \mathbb{E}\left[\left\|\sum_{k=1}^{K-1} \eta_k\right\|^2\right].
\end{aligned}
$$

Applying similar steps to $\mathbb{E}\left[\left\|\sum_{k=1}^{K-1} \eta_k\right\|^2\right], \mathbb{E}\left[\left\|\sum_{k=1}^{K-2} \eta_k\right\|^2\right], \ldots, \mathbb{E}\left[\left\|\sum_{k=1}^{2} \eta_k\right\|^2\right]$, we get the result. $\square$

**Lemma C.3.** *Suppose*

$$r_K \leq r_0(1 - a\gamma)^K + \frac{c_1\gamma}{b} + \frac{c_0}{b} \tag{22}$$

*holds for $\gamma \leq \gamma_0$. Then the choise of*

$$b \geq \frac{3c_0}{\varepsilon}$$

*and*

$$\gamma \leq \min\left(\gamma_0, \frac{c_0}{c_1}\right)$$

*implies that $r_K \leq \varepsilon$ for*

$$K \geq \frac{1}{a} \max\left(\frac{c_1}{c_0}, \frac{1}{\gamma_0}\right) \ln \frac{3r_0}{\varepsilon}$$

*Proof.* Since $b \geq \frac{3c_0}{\varepsilon}$ then $\frac{c_0}{b} \leq \frac{\varepsilon}{3}$ and $\frac{c_1\gamma}{b} \leq \frac{c_1\gamma\varepsilon}{3c_0}$. The choise of $\gamma \leq \min\left(\gamma_0, \frac{c_0}{c_1}\right)$ implies that $\frac{c_1\gamma}{b} \leq \frac{\varepsilon}{3}$.

The choice of $K \geq \frac{1}{a} \max\left(\frac{c_1}{c_0}, \frac{1}{\gamma_0}\right) \ln \frac{3r_0}{\varepsilon}$ implies that $r_0(1 - a\gamma)^K \leq \frac{\varepsilon}{3}$ and finishes the proof. $\square$

**Lemma C.4** (see also Lemma 2 from Stich [2019] and Lemma D.2 from Gorbunov et al. [2020])**.** *Let $\{r_k\}_{k \geq 0}$ satisfy*

$$r_K \leq r_0(1 - a\gamma)^{K+1} + c_1\gamma + c_2\gamma^2 \tag{23}$$

*for all $K \geq 0$ with some constants $a, c_2 > 0, c_1 \geq 0, \gamma \leq \gamma_0$.*

*Then for*

$$\gamma = \min\left\{\gamma_0, \frac{\ln\left(\max\{2, \min\{a r_0 K / c_1, a^2 r_0 K^2 / c_2\}\}\right)}{a(K+1)}\right\} \tag{24}$$

*we have that*

$$r_K = \widetilde{\mathcal{O}}\left(r_0 \exp\left(-a\gamma_0(K+1)\right) + \frac{c_1}{aK} + \frac{c_2}{a^2 K^2}\right).$$

*Moreover $r_K \leq \varepsilon$ after*

$$K = \widetilde{\mathcal{O}}\left(\frac{1}{a\gamma_0} + \frac{c_1}{a\varepsilon} + \frac{c_2}{a^2\sqrt{\varepsilon}}\right)$$

*iterations.*

*Proof.* We have

$$r_K \leq r_0(1 - a\gamma)^{K+1} + c_1\gamma + c_2\gamma^2 \leq r_0 \exp\left(-a\gamma(K+1)\right) + c_1\gamma + c_2\gamma^2. \tag{25}$$

Next we consider two possible situations.

1. If $\gamma_0 \geq \frac{\ln\left(\max\{2,\min\{ar_0K/c_1, a^2r_0K^2/c_2\}\}\right)}{a(K+1)}$ then we choose $\gamma = \frac{\ln\left(\max\{2,\min\{ar_0K/c_1, a^2r_0K^2/c_2\}\}\right)}{a(K+1)}$ and get that

$$
\begin{aligned}
r_K &\overset{(25)}{\leq} r_0 \exp\left(-a\gamma(K+1)\right) + c_1\gamma + c_2\gamma^2 \\
&= \widetilde{\mathcal{O}}\left(r_0 \exp\left(-\frac{\ln\left(\max\{2,\min\{ar_0K/c_1, a^2r_0K^2/c_2\}\}\right)}{a(K+1)}a(K+1)\right)\right) \\
&\quad + \widetilde{\mathcal{O}}\left(\frac{c_1}{aK} + \frac{c_2}{a^2K^2}\right) \\
&= \widetilde{\mathcal{O}}\left(r_0 \exp\left(-\ln\left(\max\left\{2,\min\left\{\frac{ar_0K}{c_1}, \frac{a^2r_0K^2}{c_2}\right\}\right\}\right)\right)\right) \\
&\quad + \widetilde{\mathcal{O}}\left(\frac{c_1}{aK} + \frac{c_2}{a^2K^2}\right) \\
&= \widetilde{\mathcal{O}}\left(\frac{c_1}{aK} + \frac{c_2}{a^2K^2}\right).
\end{aligned}
$$

2. If $\gamma_0 \leq \frac{\ln\left(\max\{2,\min\{aK/c_1, a^2r_0K^2/c_2\}\}\right)}{a(K+1)}$ then we choose $\gamma = \gamma_0$ which implies that

$$
\begin{aligned}
r_K &\overset{(25)}{\leq} r_0 \exp\left(-a\gamma_0(K+1)\right) + c_1\gamma_0 + c_2\gamma_0^2 \\
&= \widetilde{\mathcal{O}}\left(r_0 \exp\left(-a\gamma_0(K+1)\right) + \frac{c_1}{aK} + \frac{c_2}{a^2K^2}\right).
\end{aligned}
$$

Combining the obtained bounds we get the result. $\qquad\square$

# D Methods that use robust aggregators

First of we provide the result of Karimireddy et al. [2022] that describes error of RAGG, where $\overline{m}^t = \alpha \overline{g}^t + (1 - \alpha)\overline{m}^{t-1}$.

**Lemma D.1** (Aggregation error Karimireddy et al. [2022])**.** *Given that* RAGG *satisfies 1.1 holds, the error between the ideal average momentum $\overline{m}^t$ and the output of the robust aggregation rule $m^t$ for any $t \geq 1$ can be bounded as*

$$\mathbb{E}\|m^t - \overline{m}^t\|^2 \leq c\delta(\rho^t)^2,$$

*where we define for $t \geq 1$*

$$(\rho^t)^2 := 4(6\alpha\sigma^2 + 3\zeta^2) + 4(6\sigma^2 - 3\zeta^2)(1 - \alpha)^{t+1}.$$

*For $t = 0$ we can simplify the bound as $(\rho^0)^2 := 24\sigma^2 + 12\zeta^2$.*

*Moreover, one can state a uniform bound for $(\rho^t)^2$*

$$(\rho^t)^2 \leq \rho^2 = 24\sigma^2 + 12\zeta^2. \tag{26}$$

## D.1 Proofs for SGDA-RA

---
**Algorithm 1** SGDA-RA

---
**Input:** RAGG, $\gamma$
 1: **for** $t = 0, ...$ **do**
 2:     **for** worker $i \in [n]$ **in parallel**
 3:         $g_i^t \leftarrow g_i(x^t, \xi_i)$
 4:         **send** $g_i^t$ if $i \in \mathcal{G}$, else **send** $*$ if Byzantine
 5:     $\widehat{g}^t = \text{RAGG}(g_1^t, \ldots, g_n^t)$ and $x^{t+1} \leftarrow x^t - \gamma\widehat{g}^t$.   // update params using robust aggregate

---

### D.1.1 Quasi-Strongly Monotone Case

**Theorem** (Theorem 1 duplicate)**.** *Let Assumptions 1, 2, 4 and 6 hold. Then after $T$ iterations* SGDA-RA *(Algorithm 1) with $(\delta, c)$-RAGG and $\gamma \leq \frac{1}{2\ell}$ outputs $x^T$ such that*

$$\mathbb{E}\|x^T - x^*\|^2 \leq \left(1 - \frac{\gamma\mu}{2}\right)^T \|x^0 - x^*\|^2 + \frac{2\gamma\sigma^2}{\mu G} + \frac{2\gamma c\delta\rho^2}{\mu} + \frac{c\delta\rho^2}{\mu^2},$$

*where $\rho^2 = 24\sigma^2 + 12\zeta^2$ by Lemma D.1 with $\alpha = 1$.*

*Proof of Theorem 1.* We start the proof with

$$\|x^{t+1} - x^*\|^2 = \|x^t - x^* - \gamma\widehat{g}^t\|^2 = \|x^t - x^*\|^2 - 2\gamma\langle\widehat{g}^t, x^t - x^*\rangle + \gamma^2\|\widehat{g}^t\|^2.$$

Since $\widehat{g}^t = \widehat{g}^t - F^t + F^t$ one has

$$\|x^{t+1} - x^*\|^2 = \|x^t - x^*\|^2 - 2\gamma\langle\widehat{g}^t - \overline{g}^t, x^t - x^*\rangle - 2\gamma\langle\overline{g}^t, x^t - x^*\rangle + \gamma^2\|\widehat{g}^t\|^2.$$

Applying (11) for $\langle\widehat{g}^t - \overline{g}^t, x^t - x^*\rangle$ with $\lambda = \frac{\gamma\mu}{2}$ and (12) for $\|\widehat{g}^t\|^2 = \|\widehat{g}^t - \overline{g}^t + \overline{g}^t\|^2$ we derive

$$\begin{aligned}\|x^{t+1} - x^*\|^2 &\leq \left(1 + \frac{\gamma\mu}{2}\right)\|x^t - x^*\|^2 - 2\gamma\langle\overline{g}^t, x^t - x^*\rangle \\ &\quad + \frac{2\gamma}{\mu}\|\widehat{g}^t - \overline{g}^t\|^2 + 2\gamma^2\|\widehat{g}^t - \overline{g}^t\|^2 + 2\gamma^2\|\overline{g}^t\|^2.\end{aligned}$$

Next by taking an expectation $\mathbb{E}_{\boldsymbol{\xi}}$ of both sides of the above inequality and rearranging terms obtain

$$\begin{aligned}\mathbb{E}_{\boldsymbol{\xi}}\|x^{t+1} - x^*\|^2 &\leq \left(1 + \frac{\gamma\mu}{2}\right)\|x^t - x^*\|^2 - 2\gamma\langle F(x^t), x^t - x^*\rangle \\ &\quad + \frac{2\gamma}{\mu}\mathbb{E}_{\boldsymbol{\xi}}\|\widehat{g}^t - \overline{g}^t\|^2 + 2\gamma^2\mathbb{E}_{\boldsymbol{\xi}}\|\widehat{g}^t - \overline{g}^t\|^2 + 2\gamma^2\mathbb{E}_{\boldsymbol{\xi}}\|\overline{g}^t\|^2.\end{aligned}$$

Next we use Lemmas C.1 and D.1 to derive

$$
\begin{aligned}
\mathbb{E}_{\boldsymbol{\xi}}\big\|\boldsymbol{x}^{t+1}-\boldsymbol{x}^*\big\|^2 \;\leq\; & \left(1+\frac{\gamma\mu}{2}\right)\big\|\boldsymbol{x}^t-\boldsymbol{x}^*\big\|^2 + \left(2\gamma^2\ell-2\gamma\right)\langle F(\boldsymbol{x}^t),\boldsymbol{x}^t-\boldsymbol{x}^*\rangle \\
& +\frac{2\gamma^2\sigma^2}{G}+2c\delta\rho^2\left(\frac{\gamma}{\mu}+\gamma^2\right),
\end{aligned}
$$

that together with the choice of $\gamma \leq \frac{1}{2\ell}$ and Assumption (QSM) allows to obtain

$$
\mathbb{E}_{\boldsymbol{\xi}}\big\|\boldsymbol{x}^{t+1}-\boldsymbol{x}^*\big\|^2 \;\leq\; \left(1-\frac{\gamma\mu}{2}\right)\big\|\boldsymbol{x}^t-\boldsymbol{x}^*\big\|^2 + \frac{2\gamma^2\sigma^2}{G}+2c\delta\rho^2\left(\frac{\gamma}{\mu}+\gamma^2\right).
$$

Next we take full expectation of both sides and obtain

$$
\mathbb{E}\big\|\boldsymbol{x}^{t+1}-\boldsymbol{x}^*\big\|^2 \leq \left(1-\frac{\gamma\mu}{2}\right)\mathbb{E}\big\|\boldsymbol{x}^t-\boldsymbol{x}^*\big\|^2 + \frac{2\gamma^2\sigma^2}{G}+2c\delta\rho^2\left(\frac{\gamma}{\mu}+\gamma^2\right).
$$

The latter implies

$$
\mathbb{E}\big\|\boldsymbol{x}^T-\boldsymbol{x}^*\big\|^2 \leq \left(1-\frac{\gamma\mu}{2}\right)^T\big\|\boldsymbol{x}^0-\boldsymbol{x}^*\big\|^2 + \frac{4\gamma\sigma^2}{\mu G}+\frac{4\gamma c\delta\rho^2}{\mu}+\frac{4c\delta\rho^2}{\mu^2},
$$

where $\rho$ is bounded by Lemma D.1 with $\alpha = 1$. $\qquad\square$

**Corollary 1.** *Let assumptions of Theorem 1 hold. Then* $\mathbb{E}\big\|\boldsymbol{x}^T-\boldsymbol{x}^*\big\|^2 \leq \varepsilon$ *holds after*

$$
T \geq \left(4+\frac{4\ell}{\mu}+\frac{1}{3c\delta G}\right)\ln\frac{3R^2}{\varepsilon}
$$

*iterations of* SGDA-RA *with* $\gamma = \min\left(\frac{1}{2\ell},\frac{1}{2\mu+\frac{\mu}{6c\delta G}}\right)$ *and* $b \geq \frac{72c\delta\sigma^2}{\mu^2\varepsilon}$.

*Proof.* If $\zeta = 0$, $\rho^2 = 24\sigma^2$ the result of Theorem 1 can be simplified as

$$
\mathbb{E}\big\|\boldsymbol{x}^T-\boldsymbol{x}^*\big\|^2 \leq \left(1-\frac{\gamma\mu}{2}\right)^T\big\|\boldsymbol{x}^0-\boldsymbol{x}^*\big\|^2 + \frac{2\gamma\sigma^2}{\mu G}+\frac{48\gamma c\delta\sigma^2}{\mu}+\frac{24c\delta\sigma^2}{\mu^2}.
$$

Applying Lemma C.3 to the last bound we get the result of the corollary. $\qquad\square$

### D.2  Proofs for SEG-RA

---
**Algorithm 2** SEG-RA

---
**Input:** RAGG, $\gamma$
1: **for** $t = 1,...$ **do**
2:     **for** worker $i \in [n]$ **in parallel**
3:         $\boldsymbol{g}^t_{\boldsymbol{\xi}_i} \leftarrow \boldsymbol{g}_i(\boldsymbol{x}^t,\boldsymbol{\xi}_i)$
4:         **send** $\boldsymbol{g}^t_{\boldsymbol{\xi}_i}$ if $i \in \mathcal{G}$, else **send** $*$ if Byzantine
5:     $\widehat{\boldsymbol{g}}_{\boldsymbol{\xi}^t}(\boldsymbol{x}^t) = \text{RAGG}\,(\boldsymbol{g}^t_{\boldsymbol{\xi}_1},\ldots,\boldsymbol{g}^t_{\boldsymbol{\xi}_n})$
6:     $\widetilde{\boldsymbol{x}}^t \leftarrow \boldsymbol{x}^t - \gamma_1\widehat{\boldsymbol{g}}_{\boldsymbol{\xi}^t}(\boldsymbol{x}^t).$           // update params using robust aggregate
7:     **for** worker $i \in [n]$ **in parallel**
8:         $\boldsymbol{g}^t_{\boldsymbol{\eta}_i} \leftarrow \boldsymbol{g}_i(\widetilde{\boldsymbol{x}}^t,\boldsymbol{\eta}_i)$
9:         **send** $\boldsymbol{g}^t_{\boldsymbol{\eta}_i}$ if $i \in \mathcal{G}$, else **send** $*$ if Byzantine
10:    $\widehat{\boldsymbol{g}}_{\boldsymbol{\eta}^t}(\widetilde{\boldsymbol{x}}^t) = \text{RAGG}\,(\boldsymbol{g}^t_{\boldsymbol{\eta}_1},\ldots,\boldsymbol{g}^t_{\boldsymbol{\eta}_n})$
11:    $\boldsymbol{x}^{t+1} \leftarrow \boldsymbol{x}^t - \gamma_2\widehat{\boldsymbol{g}}_{\boldsymbol{\eta}^t}(\widetilde{\boldsymbol{x}}^t).$        // update params using robust aggregate

---

To analyze the convergence of SEG introduce the following notation

$$
\overline{\boldsymbol{g}}_{\boldsymbol{\xi}^k}(\boldsymbol{x}^k) = \boldsymbol{g}_{\boldsymbol{\xi}^k}(\boldsymbol{x}^k) = \frac{1}{G}\sum_{i\in\mathcal{G}}\boldsymbol{g}_i(\boldsymbol{x}^k,\boldsymbol{\xi}^k_i)
$$

$$\widehat{\boldsymbol{g}}_{\boldsymbol{\xi}^k}(\boldsymbol{x}^k) = \mathrm{RAGG}\big(\boldsymbol{g}_1(\boldsymbol{x}^k, \boldsymbol{\xi}_1^k), \ldots, \boldsymbol{g}_n(\boldsymbol{x}^k, \boldsymbol{\xi}_n^k)\big),$$

$$\widetilde{\boldsymbol{x}}^k = \boldsymbol{x}^k - \gamma_1 \widehat{\boldsymbol{g}}_{\boldsymbol{\xi}^k}(\boldsymbol{x}^k),$$

$$\overline{\boldsymbol{g}}_{\boldsymbol{\eta}^k}(\widetilde{\boldsymbol{x}}^k) = \overline{\boldsymbol{g}}_{\boldsymbol{\eta}^k}(\boldsymbol{x}^k) = \frac{1}{G} \sum_{i \in \mathcal{G}} \boldsymbol{g}_i(\boldsymbol{x}^k, \boldsymbol{\eta}_i^k)$$

where $\boldsymbol{\xi}_i^k$, $i \in \mathcal{G}$ and $\boldsymbol{\eta}_j^k$, $j \in \mathcal{G}$ are i.i.d. samples satisfying Assumption 1, i.e., due to the independence we have

**Corollary 2.** *Suppose that the operator $F$ is given in the form* (1) *and Assumption 1 holds. Then*

$$\mathbb{E}_{\boldsymbol{\xi}^k}\left[\|\overline{\boldsymbol{g}}_{\boldsymbol{\xi}^k}(\boldsymbol{x}^k) - F(\boldsymbol{x}^k)\|^2\right] \quad \leq \quad \frac{\sigma^2}{G}, \tag{27}$$

$$\mathbb{E}_{\boldsymbol{\eta}^k}\left[\|\overline{\boldsymbol{g}}_{\boldsymbol{\eta}^k}(\widetilde{\boldsymbol{x}}^k) - F(\widetilde{\boldsymbol{x}}^k)\|^2\right] \quad \leq \quad \frac{\sigma^2}{G}. \tag{28}$$

### D.2.1 Auxilary results

**Lemma D.2.** *Let Assumptions 2, 3, 4 and Corollary 2 hold. If*

$$\gamma_1 \leq \frac{1}{2L} \tag{29}$$

*for* SEG-RA *(Algorithm 2), then $\overline{\boldsymbol{g}}_{\boldsymbol{\eta}^k}(\widetilde{\boldsymbol{x}}^k) = \overline{\boldsymbol{g}}_{\boldsymbol{\eta}^k}\left(\boldsymbol{x}^k - \gamma_1 \widehat{\boldsymbol{g}}_{\boldsymbol{\xi}^k}(\boldsymbol{x}^k)\right)$ satisfies the following inequality*

$$\gamma_1^2 \mathbb{E}\left[\left\|\overline{\boldsymbol{g}}_{\boldsymbol{\eta}^k}(\widetilde{\boldsymbol{x}}^k)\right\|^2 \mid \boldsymbol{x}^k\right] \quad \leq \quad 2\widehat{P}_k + \frac{8\gamma_1^2 \sigma^2}{G} + 4\gamma_1^2 c\delta\rho^2, \tag{30}$$

*where $\widehat{P}_k = \gamma_1 \mathbb{E}_{\boldsymbol{\xi}^k, \boldsymbol{\eta}^k}\left[\langle \overline{\boldsymbol{g}}_{\boldsymbol{\eta}^k}(\widetilde{\boldsymbol{x}}^k), \boldsymbol{x}^k - \boldsymbol{x}^*\rangle\right]$ and $\rho^2 = 24\sigma^2 + 12\zeta^2$ by Lemma D.1 with $\alpha = 1$.*

*Proof.* Using the auxiliary iterate $\widehat{\boldsymbol{x}}^{k+1} = \boldsymbol{x}^k - \gamma_1 \overline{\boldsymbol{g}}_{\boldsymbol{\eta}^k}(\widetilde{\boldsymbol{x}}^k)$, we get

$$\begin{aligned}
\left\|\widehat{\boldsymbol{x}}^{k+1} - \boldsymbol{x}^*\right\|^2 &= \left\|\boldsymbol{x}^k - \boldsymbol{x}^*\right\|^2 - 2\gamma_1 \langle \boldsymbol{x}^k - \boldsymbol{x}^*, \overline{\boldsymbol{g}}_{\boldsymbol{\eta}^k}(\widetilde{\boldsymbol{x}}^k)\rangle + \gamma_1^2 \left\|\overline{\boldsymbol{g}}_{\boldsymbol{\eta}^k}(\widetilde{\boldsymbol{x}}^k)\right\|^2 & (31) \\
&= \left\|\boldsymbol{x}^k - \boldsymbol{x}^*\right\|^2 - 2\gamma_1 \left\langle \boldsymbol{x}^k - \gamma \widehat{\boldsymbol{g}}_{\boldsymbol{\xi}^k}(\boldsymbol{x}^k) - \boldsymbol{x}^*, \overline{\boldsymbol{g}}_{\boldsymbol{\eta}^k}(\widetilde{\boldsymbol{x}}^k)\right\rangle & (32) \\
&\quad - 2\gamma_1^2 \langle \widehat{\boldsymbol{g}}_{\boldsymbol{\xi}^k}(\boldsymbol{x}^k), \overline{\boldsymbol{g}}_{\boldsymbol{\eta}^k}(\widetilde{\boldsymbol{x}}^k)\rangle + \gamma_1^2 \left\|\overline{\boldsymbol{g}}_{\boldsymbol{\eta}^k}(\widetilde{\boldsymbol{x}}^k)\right\|^2. & (33)
\end{aligned}$$

Taking the expectation $\mathbb{E}_{\boldsymbol{\xi}^k, \boldsymbol{\eta}^k}[\cdot] = \mathbb{E}[\cdot \mid \boldsymbol{x}^k]$ conditioned on $\boldsymbol{x}^k$ from the above identity, using tower property $\mathbb{E}_{\boldsymbol{\xi}^k, \boldsymbol{\eta}^k}[\cdot] = \mathbb{E}_{\boldsymbol{\xi}^k}[\mathbb{E}_{\boldsymbol{\eta}^k}[\cdot]]$, and $\mu$-quasi strong monotonicity of $F(x)$, we derive

$$\begin{aligned}
&\mathbb{E}_{\boldsymbol{\xi}^k, \boldsymbol{\eta}^k}\left[\left\|\widehat{\boldsymbol{x}}^{k+1} - \boldsymbol{x}^*\right\|^2\right] \\
&= \left\|\boldsymbol{x}^k - \boldsymbol{x}^*\right\|^2 + \gamma_1^2 \mathbb{E}_{\boldsymbol{\xi}^k, \boldsymbol{\eta}^k}\left[\left\|\overline{\boldsymbol{g}}_{\boldsymbol{\eta}^k}(\widetilde{\boldsymbol{x}}^k)\right\|^2\right] \\
&\quad - 2\gamma_1 \mathbb{E}_{\boldsymbol{\xi}^k, \boldsymbol{\eta}^k}\left[\langle \boldsymbol{x}^k - \gamma_1 \widehat{\boldsymbol{g}}_{\boldsymbol{\xi}^k}(\boldsymbol{x}^k) - \boldsymbol{x}^*, \overline{\boldsymbol{g}}_{\boldsymbol{\eta}^k}(\widetilde{\boldsymbol{x}}^k)\rangle\right] \\
&\quad - 2\gamma_1^2 \mathbb{E}_{\boldsymbol{\xi}^k, \boldsymbol{\eta}^k}\left[\langle \widehat{\boldsymbol{g}}_{\boldsymbol{\xi}^k}(\boldsymbol{x}^k), \overline{\boldsymbol{g}}_{\boldsymbol{\eta}^k}(\widetilde{\boldsymbol{x}}^k)\rangle\right] \\
&= \left\|\boldsymbol{x}^k - \boldsymbol{x}^*\right\|^2 \\
&\quad - 2\gamma_1 \mathbb{E}_{\boldsymbol{\xi}^k}\left[\langle \boldsymbol{x}^k - \gamma_1 \widehat{\boldsymbol{g}}_{\boldsymbol{\xi}^k}(\boldsymbol{x}^k) - \boldsymbol{x}^*, F\left(\boldsymbol{x}^k - \gamma_1 \widehat{\boldsymbol{g}}_{\boldsymbol{\xi}^k}(\boldsymbol{x}^k)\right)\rangle\right] \\
&\quad - 2\gamma_1^2 \mathbb{E}_{\boldsymbol{\xi}^k}\left[\langle \widehat{\boldsymbol{g}}_{\boldsymbol{\xi}^k}(\boldsymbol{x}^k), \overline{\boldsymbol{g}}_{\boldsymbol{\eta}^k}(\widetilde{\boldsymbol{x}}^k)\rangle\right] + \gamma_1^2 \mathbb{E}_{\boldsymbol{\xi}^k, \boldsymbol{\eta}^k}\left[\left\|\overline{\boldsymbol{g}}_{\boldsymbol{\eta}^k}(\widetilde{\boldsymbol{x}}^k)\right\|^2\right] \\
&\overset{\mathrm{(QSM),(14)}}{\leq} \left\|\boldsymbol{x}^k - \boldsymbol{x}^*\right\|^2 - \gamma_1^2 \mathbb{E}_{\boldsymbol{\xi}^k, \boldsymbol{\eta}^k}\left[\left\|\widehat{\boldsymbol{g}}_{\boldsymbol{\xi}^k}(\boldsymbol{x}^k)\right\|^2\right] \\
&\quad + \gamma_1^2 \mathbb{E}_{\boldsymbol{\xi}^k, \boldsymbol{\eta}^k}\left[\left\|\widehat{\boldsymbol{g}}_{\boldsymbol{\xi}^k}(\boldsymbol{x}^k) - \overline{\boldsymbol{g}}_{\boldsymbol{\eta}^k}(\widetilde{\boldsymbol{x}}^k)\right\|^2\right].
\end{aligned}$$

To upper bound the last term we use simple inequality (16), and apply $L$-Lipschitzness of $F(x)$:

$$
\begin{aligned}
\mathbb{E}_{\boldsymbol{\xi}^k, \boldsymbol{\eta}^k} \left[ \left\| \widehat{\boldsymbol{x}}^{k+1} - \boldsymbol{x}^* \right\|^2 \right] & \overset{(16)}{\leq} \left\| \boldsymbol{x}^k - \boldsymbol{x}^* \right\|^2 - \gamma_1^2 \mathbb{E}_{\boldsymbol{\xi}^k} \left[ \left\| \widehat{\boldsymbol{g}}_{\boldsymbol{\xi}^k}(\boldsymbol{x}^k) \right\|^2 \right] \\
& \quad + 4\gamma_1^2 \mathbb{E}_{\boldsymbol{\xi}^k} \left[ \left\| \overline{\boldsymbol{g}}_{\boldsymbol{\xi}^k}(\boldsymbol{x}^k) - \widehat{\boldsymbol{g}}_{\boldsymbol{\xi}^k}(\boldsymbol{x}^k) \right\|^2 \right] \\
& \quad + 4\gamma_1^2 \mathbb{E}_{\boldsymbol{\xi}^k} \left[ \left\| F(\boldsymbol{x}^k) - F\left(\widetilde{\boldsymbol{x}}^k\right) \right\|^2 \right] \\
& \quad + 4\gamma_1^2 \mathbb{E}_{\boldsymbol{\xi}^k} \left[ \left\| \overline{\boldsymbol{g}}_{\boldsymbol{\xi}^k}(\boldsymbol{x}^k) - F(\boldsymbol{x}^k) \right\|^2 \right] \\
& \quad + 4\gamma_1^2 \mathbb{E}_{\boldsymbol{\xi}^k, \boldsymbol{\eta}^k} \left[ \left\| \overline{\boldsymbol{g}}_{\boldsymbol{\eta}^k}\left(\widetilde{\boldsymbol{x}}^k\right) - F\left(\widetilde{\boldsymbol{x}}^k\right) \right\|^2 \right] \\
& \overset{(\text{Lip}),(27),(28)}{\leq} \left\| \boldsymbol{x}^k - \boldsymbol{x}^* \right\|^2 - \gamma_1^2 \left( 1 - 4L^2 \gamma_1^2 \right) \mathbb{E}_{\boldsymbol{\xi}^k} \left[ \left\| \widehat{\boldsymbol{g}}_{\boldsymbol{\xi}^k}(\boldsymbol{x}^k) \right\|^2 \right] \\
& \quad + 4\gamma_1^2 \mathbb{E}_{\boldsymbol{\xi}^k} \left[ \left\| \overline{\boldsymbol{g}}_{\boldsymbol{\xi}^k}(\boldsymbol{x}^k) - \widehat{\boldsymbol{g}}_{\boldsymbol{\xi}^k}(\boldsymbol{x}^k) \right\|^2 \right] \\
& \quad + \frac{4\gamma_1^2 \sigma^2}{G} + \frac{4\gamma_1^2 \sigma^2}{G} \\
& \overset{(16),Lemma\ D.1}{\leq} \left\| \boldsymbol{x}^k - \boldsymbol{x}^* \right\|^2 - \gamma_1^2 \left( 1 - 4\gamma_1^2 L^2 \right) \mathbb{E}_{\boldsymbol{\xi}^k} \left[ \left\| \widehat{\boldsymbol{g}}_{\boldsymbol{\xi}^k}(\boldsymbol{x}^k) \right\|^2 \right] \\
& \quad + \frac{8\gamma_1^2 \sigma^2}{G} + 4\gamma_1^2 c\delta \rho^2 \\
& \overset{(29)}{\leq} \left\| \boldsymbol{x}^k - \boldsymbol{x}^* \right\|^2 + \frac{8\gamma_1^2 \sigma^2}{G} + 4\gamma_1^2 c\delta \rho^2.
\end{aligned}
$$

Finally, we use the above inequality together with (31):

$$
\left\| \boldsymbol{x}^k - \boldsymbol{x}^* \right\|^2 - 2\widehat{P}_k + \gamma_1^2 \mathbb{E} \left[ \left\| \overline{\boldsymbol{g}}_{\boldsymbol{\eta}^k}(\widetilde{\boldsymbol{x}}^k) \right\|^2 \mid \boldsymbol{x}^k \right] \leq \left\| \boldsymbol{x}^k - \boldsymbol{x}^* \right\|^2 + \frac{8\gamma_1^2 \sigma^2}{G} + 4\gamma_1^2 c\delta \rho^2,
$$

where $\widehat{P}_k = \gamma_1 \mathbb{E}_{\boldsymbol{\xi}^k, \boldsymbol{\eta}^k} \left[ \left\langle \overline{\boldsymbol{g}}_{\boldsymbol{\eta}^k}(\widetilde{\boldsymbol{x}}^k), \boldsymbol{x}^k - \boldsymbol{x}^* \right\rangle \right]$. Rearranging the terms, we obtain (30). $\square$

**Lemma D.3.** *Consider* SEG-RA *(Algorithm 2). Let Assumptions 2, 3, 4 and Corollary 2 hold. If*

$$
\gamma_1 \leq \frac{1}{2\mu + 2L}, \tag{34}
$$

*then* $\overline{\boldsymbol{g}}_{\boldsymbol{\eta}^k}(\widetilde{\boldsymbol{x}}^k) = \overline{\boldsymbol{g}}_{\boldsymbol{\eta}^k}\left(\boldsymbol{x}^k - \gamma_1 \widehat{\boldsymbol{g}}_{\boldsymbol{\xi}^k}(\boldsymbol{x}^k)\right)$ *satisfies the following inequality*

$$
\widehat{P}_k \geq \frac{\mu\gamma_1}{2} \left\| \boldsymbol{x}^k - \boldsymbol{x}^* \right\|^2 + \frac{\gamma_1^2}{4} \mathbb{E}_{\boldsymbol{\xi}^k} \left[ \left\| \overline{\boldsymbol{g}}_{\boldsymbol{\xi}^k}(\boldsymbol{x}^k) \right\|^2 \right] - \frac{8\gamma_1^2 \sigma^2}{G} - \frac{9\gamma_1^2 c\delta \rho^2}{2}, \tag{35}
$$

*or simply*

$$
-\widehat{P}_k \leq -\frac{\mu\gamma_1}{2} \left\| \boldsymbol{x}^k - \boldsymbol{x}^* \right\|^2 + \frac{4\gamma_1^2 \sigma^2}{G} + 4\gamma_1^2 c\delta \rho^2
$$

*where* $\widehat{P}_k = \gamma_1 \mathbb{E}_{\boldsymbol{\xi}^k, \boldsymbol{\eta}^k} \left[ \left\langle \overline{\boldsymbol{g}}_{\boldsymbol{\eta}^k}(\widetilde{\boldsymbol{x}}^k), \boldsymbol{x}^k - \boldsymbol{x}^* \right\rangle \right]$ *and* $\rho^2 = 24\sigma^2 + 12\zeta^2$ *by Lemma D.1 with* $\alpha = 1$.

*Proof.* Since $\mathbb{E}_{\boldsymbol{\xi}^k, \boldsymbol{\eta}^k}[\cdot] = \mathbb{E}[\cdot \mid \boldsymbol{x}^k]$ and $\overline{\boldsymbol{g}}_{\boldsymbol{\eta}^k}(\widetilde{\boldsymbol{x}}^k) = \overline{\boldsymbol{g}}_{\boldsymbol{\eta}^k}(\boldsymbol{x}^k - \gamma_1 \widehat{\boldsymbol{g}}_{\boldsymbol{\xi}^k}(\boldsymbol{x}^k))$, we have

$$
\begin{aligned}
-\widehat{P}_k \\
= \quad & -\gamma_1 \mathbb{E}_{\boldsymbol{\xi}^k, \boldsymbol{\eta}^k}\left[\langle \overline{\boldsymbol{g}}_{\boldsymbol{\eta}^k}(\widetilde{\boldsymbol{x}}^k), \boldsymbol{x}^k - \boldsymbol{x}^*\rangle\right] \\
= \quad & -\gamma_1 \mathbb{E}_{\boldsymbol{\xi}^k}\left[\langle \mathbb{E}_{\boldsymbol{\eta}^k}[\overline{\boldsymbol{g}}_{\boldsymbol{\eta}^k}(\widetilde{\boldsymbol{x}}^k)], \boldsymbol{x}^k - \gamma_1 \widehat{\boldsymbol{g}}_{\boldsymbol{\xi}^k}(\boldsymbol{x}^k) - \boldsymbol{x}^*\rangle\right] \\
& -\gamma_1^2 \mathbb{E}\left[\langle \overline{\boldsymbol{g}}_{\boldsymbol{\eta}^k}(\widetilde{\boldsymbol{x}}^k), \widehat{\boldsymbol{g}}_{\boldsymbol{\xi}^k}(\boldsymbol{x}^k)\rangle\right] \\
\overset{(14)}{=} \quad & -\gamma_1 \mathbb{E}_{\boldsymbol{\xi}^k}\left[\langle F(\boldsymbol{x}^k - \gamma_1 \widehat{\boldsymbol{g}}_{\boldsymbol{\xi}^k}(\boldsymbol{x}^k)), \boldsymbol{x}^k - \gamma_1 \widehat{\boldsymbol{g}}_{\boldsymbol{\xi}^k}(\boldsymbol{x}^k) - \boldsymbol{x}^*\rangle\right] \\
& -\frac{\gamma_1^2}{2}\mathbb{E}_{\boldsymbol{\xi}^k, \boldsymbol{\eta}^k}\left[\|\overline{\boldsymbol{g}}_{\boldsymbol{\eta}^k}(\widetilde{\boldsymbol{x}}^k)\|^2\right] - \frac{\gamma_1^2}{2}\mathbb{E}_{\boldsymbol{\xi}^k}\left[\|\widehat{\boldsymbol{g}}_{\boldsymbol{\xi}^k}(\boldsymbol{x}^k)\|^2\right] \\
& +\frac{\gamma_1^2}{2}\mathbb{E}_{\boldsymbol{\xi}^k, \boldsymbol{\eta}^k}\left[\|\overline{\boldsymbol{g}}_{\boldsymbol{\eta}^k}(\widetilde{\boldsymbol{x}}^k) - \widehat{\boldsymbol{g}}_{\boldsymbol{\xi}^k}(\boldsymbol{x}^k)\|^2\right] \\
\overset{(QSM),(16)}{\leq} \quad & -\mu\gamma_1 \mathbb{E}_{\boldsymbol{\xi}^k, \boldsymbol{\eta}^k}\left[\|\boldsymbol{x}^k - \boldsymbol{x}^* - \gamma_1 \widehat{\boldsymbol{g}}_{\boldsymbol{\xi}^k}(\boldsymbol{x}^k)\|^2\right] - \frac{\gamma_1^2}{2}\mathbb{E}_{\boldsymbol{\xi}^k}\left[\|\widehat{\boldsymbol{g}}_{\boldsymbol{\xi}^k}(\boldsymbol{x}^k)\|^2\right] \\
& +\frac{4\gamma_1^2}{2}\mathbb{E}_{\boldsymbol{\xi}^k}\left[\|\overline{\boldsymbol{g}}_{\boldsymbol{\xi}^k}(\boldsymbol{x}^k) - \widehat{\boldsymbol{g}}_{\boldsymbol{\xi}^k}(\boldsymbol{x}^k)\|^2\right] \\
& +\frac{4\gamma_1^2}{2}\mathbb{E}_{\boldsymbol{\xi}^k}\left[\|F(\boldsymbol{x}^k) - F(\widetilde{\boldsymbol{x}}^k)\|^2\right] \\
& +\frac{4\gamma_1^2}{2}\mathbb{E}_{\boldsymbol{\xi}^k}\left[\|\overline{\boldsymbol{g}}_{\boldsymbol{\xi}^k}(\boldsymbol{x}^k) - F(\boldsymbol{x}^k)\|^2\right] \\
& +\frac{4\gamma_1^2}{2}\mathbb{E}_{\boldsymbol{\xi}^k, \boldsymbol{\eta}^k}\left[\|\overline{\boldsymbol{g}}_{\boldsymbol{\eta}^k}(\widetilde{\boldsymbol{x}}^k) - F(\widetilde{\boldsymbol{x}}^k)\|^2\right] \\
\overset{(17),(Lip),Lem.\ D.1,Cor.\ 2}{\leq} \quad & -\frac{\mu\gamma_1}{2}\|\boldsymbol{x}^k - \boldsymbol{x}^*\|^2 - \frac{\gamma_1^2}{2}(1 - 2\gamma_1\mu - 4\gamma_1^2 L^2)\mathbb{E}_{\boldsymbol{\xi}^k}\left[\|\widehat{\boldsymbol{g}}_{\boldsymbol{\xi}^k}(\boldsymbol{x}^k)\|^2\right] \\
& +\frac{4\gamma_1^2\sigma^2}{2G} + \frac{4\gamma_1^2\sigma^2}{2G} + 4\gamma_1^2 c\delta\rho^2 \\
\overset{(34)}{\leq} \quad & -\frac{\mu\gamma_1}{2}\|\boldsymbol{x}^k - \boldsymbol{x}^*\|^2 - \frac{\gamma_1^2}{2}\mathbb{E}_{\boldsymbol{\xi}^k}\left[\|\widehat{\boldsymbol{g}}_{\boldsymbol{\xi}^k}(\boldsymbol{x}^k)\|^2\right] + \frac{4\gamma_1^2\sigma^2}{G} + 4\gamma_1^2 c\delta\rho^2
\end{aligned}
$$

So one have

$$
-\widehat{P}_k \quad \leq \quad -\frac{\mu\gamma_1}{2}\|\boldsymbol{x}^k - \boldsymbol{x}^*\|^2 - \frac{\gamma_1^2}{4}\mathbb{E}_{\boldsymbol{\xi}^k}\left[\|\overline{\boldsymbol{g}}_{\boldsymbol{\xi}^k}(\boldsymbol{x}^k)\|^2\right] + \frac{4\gamma_1^2\sigma^2}{G} + \frac{9\gamma_1^2 c\delta\rho^2}{2}
$$

or simply

$$
-\widehat{P}_k \quad \leq \quad -\frac{\mu\gamma_1}{2}\|\boldsymbol{x}^k - \boldsymbol{x}^*\|^2 + \frac{4\gamma_1^2\sigma^2}{G} + 4\gamma_1^2 c\delta\rho^2
$$

that concludes the proof. $\qquad\square$

### D.2.2 Quasi-Strongly Monotone Case

Combining Lemmas D.2 and D.3, we get the following result.

**Theorem** (Theorem 2 duplicate)**.** *Let Assumptions 1, 2, 3 and 4 hold. Then after $T$ iterations* SEG-RA *(Algorithm 2) with $(\delta, c)$-RAGG, $\gamma_1 \leq \frac{1}{2\mu + 2L}$ and $\beta = \gamma_2/\gamma_1 \leq 1/4$ outputs $\boldsymbol{x}^T$ such that*

$$
\mathbb{E}\|\boldsymbol{x}^T - \boldsymbol{x}^*\|^2 \leq \left(1 - \frac{\mu\beta\gamma_1}{4}\right)^T \|\boldsymbol{x}^0 - \boldsymbol{x}^*\|^2 + \frac{8\gamma_1\sigma^2}{\mu\beta G} + 8c\delta\rho^2\left(\frac{\gamma_1}{\beta\mu} + \frac{2}{\mu^2}\right),
$$

*where $\rho^2 = 24\sigma^2 + 12\zeta^2$ by Lemma D.1 with $\alpha = 1$.*

*Proof of Theorem 2.* Since $\boldsymbol{x}^{k+1} = \boldsymbol{x}^k - \gamma_2 \widehat{\boldsymbol{g}}_{\boldsymbol{\eta}^k}(\widetilde{\boldsymbol{x}}^k)$, we have

$$
\begin{aligned}
\left\|\boldsymbol{x}^{k+1} - \boldsymbol{x}^*\right\|^2 &= \left\|\boldsymbol{x}^k - \gamma_2 \widehat{\boldsymbol{g}}_{\boldsymbol{\eta}^k}(\widetilde{\boldsymbol{x}}^k) - \boldsymbol{x}^*\right\|^2 \\
&= \left\|\boldsymbol{x}^k - \boldsymbol{x}^*\right\|^2 - 2\gamma_2 \langle \widehat{\boldsymbol{g}}_{\boldsymbol{\eta}^k}(\widetilde{\boldsymbol{x}}^k), \boldsymbol{x}^k - \boldsymbol{x}^* \rangle + \gamma_2^2 \left\|\widehat{\boldsymbol{g}}_{\boldsymbol{\eta}^k}(\widetilde{\boldsymbol{x}}^k)\right\|^2 \\
&\leq \left\|\boldsymbol{x}^k - \boldsymbol{x}^*\right\|^2 - 2\gamma_2 \langle \overline{\boldsymbol{g}}_{\boldsymbol{\eta}^k}(\widetilde{\boldsymbol{x}}^k), \boldsymbol{x}^k - \boldsymbol{x}^* \rangle + 2\gamma_2^2 \left\|\overline{\boldsymbol{g}}_{\boldsymbol{\eta}^k}(\widetilde{\boldsymbol{x}}^k)\right\|^2 \\
&\quad + 2\gamma_2^2 \left\|\overline{\boldsymbol{g}}_{\boldsymbol{\eta}^k}(\widetilde{\boldsymbol{x}}^k) - \widehat{\boldsymbol{g}}_{\boldsymbol{\eta}^k}(\widetilde{\boldsymbol{x}}^k)\right\|^2 + 2\gamma_2 \langle \overline{\boldsymbol{g}}_{\boldsymbol{\eta}^k}(\widetilde{\boldsymbol{x}}^k) - \widehat{\boldsymbol{g}}_{\boldsymbol{\eta}^k}(\widetilde{\boldsymbol{x}}^k), \boldsymbol{x}^k - \boldsymbol{x}^* \rangle \\
&\leq (1 + \lambda) \left\|\boldsymbol{x}^k - \boldsymbol{x}^*\right\|^2 - 2\gamma_2 \langle \overline{\boldsymbol{g}}_{\boldsymbol{\eta}^k}(\widetilde{\boldsymbol{x}}^k), \boldsymbol{x}^k - \boldsymbol{x}^* \rangle + 2\gamma_2^2 \left\|\overline{\boldsymbol{g}}_{\boldsymbol{\eta}^k}(\widetilde{\boldsymbol{x}}^k)\right\|^2 \\
&\quad + \gamma_2^2 \left(2 + \frac{1}{\lambda}\right) \left\|\overline{\boldsymbol{g}}_{\boldsymbol{\eta}^k}(\widetilde{\boldsymbol{x}}^k) - \widehat{\boldsymbol{g}}_{\boldsymbol{\eta}^k}(\widetilde{\boldsymbol{x}}^k)\right\|^2
\end{aligned}
$$

Taking the expectation, conditioned on $\boldsymbol{x}^k$,

$$
\begin{aligned}
\mathbb{E}_{\boldsymbol{\xi}^k, \boldsymbol{\eta}^k} \left\|\boldsymbol{x}^{k+1} - \boldsymbol{x}^*\right\|^2 &\leq (1 + \lambda) \left\|\boldsymbol{x}^k - \boldsymbol{x}^*\right\|^2 - 2\beta\gamma_1 \mathbb{E}_{\boldsymbol{\xi}^k, \boldsymbol{\eta}^k} \langle \overline{\boldsymbol{g}}_{\boldsymbol{\eta}^k}(\widetilde{\boldsymbol{x}}^k), \boldsymbol{x}^k - \boldsymbol{x}^* \rangle \\
&\quad + 2\beta^2 \gamma_1^2 \mathbb{E}_{\boldsymbol{\xi}^k, \boldsymbol{\eta}^k} \left\|\overline{\boldsymbol{g}}_{\boldsymbol{\eta}^k}(\widetilde{\boldsymbol{x}}^k)\right\|^2 + \gamma_2^2 c\delta\rho^2 \left(2 + \frac{1}{\lambda}\right),
\end{aligned}
$$

using the definition of $\widehat{P}_k = \gamma_1 \mathbb{E}_{\boldsymbol{\xi}^k, \boldsymbol{\eta}^k} \left[\langle \overline{\boldsymbol{g}}_{\boldsymbol{\eta}^k}(\widetilde{\boldsymbol{x}}^k), \boldsymbol{x}^k - \boldsymbol{x}^* \rangle\right]$, we continue our derivation:

$$
\mathbb{E}_{\boldsymbol{\xi}^k, \boldsymbol{\eta}^k} \left[\left\|\boldsymbol{x}^{k+1} - \boldsymbol{x}^*\right\|^2\right] \tag{36}
$$

$$
\begin{aligned}
&= (1 + \lambda) \left\|\boldsymbol{x}^k - \boldsymbol{x}^*\right\|^2 - 2\beta\widehat{P}_k + 2\beta^2 \gamma_1^2 \mathbb{E}_{\boldsymbol{\xi}^k, \boldsymbol{\eta}^k} \left\|\overline{\boldsymbol{g}}_{\boldsymbol{\eta}^k}(\widetilde{\boldsymbol{x}}^k)\right\|^2 \\
&\quad + \gamma_2^2 c\delta\rho^2 \left(2 + \frac{1}{\lambda}\right) \\
&\overset{(30)}{\leq} (1 + \lambda) \left\|\boldsymbol{x}^k - \boldsymbol{x}^*\right\|^2 - 2\beta\widehat{P}_k + 2\beta^2 \left(2\widehat{P}_k + \frac{8\gamma_1^2 \sigma^2}{G} + 4\gamma_1^2 c\delta\rho^2\right) \\
&\quad + \gamma_2^2 c\delta\rho^2 \left(2 + \frac{1}{\lambda}\right) \\
&\overset{0 \leq \beta \leq 1/2}{\leq} (1 + \lambda) \left\|\boldsymbol{x}^k - \boldsymbol{x}^*\right\|^2 - 2\widehat{P}_k(\beta - 2\beta^2) + \frac{16\gamma_2^2 \sigma^2}{G} + 8\gamma_2^2 c\delta\rho^2 \\
&\quad + \gamma_2^2 c\delta\rho^2 \left(2 + \frac{1}{\lambda}\right) \\
&\overset{(35)}{\leq} (1 + \lambda) \left\|\boldsymbol{x}^k - \boldsymbol{x}^*\right\|^2 \\
&\quad + 2\beta(1 - 2\beta) \left(-\frac{\mu\gamma_1}{2} \left\|\boldsymbol{x}^k - \boldsymbol{x}^*\right\|^2 + \frac{4\gamma_1^2 \sigma^2}{G} + 4\gamma_1^2 c\delta\rho^2\right) \\
&\quad + \frac{16\gamma_2^2 \sigma^2}{G} + 8\gamma_2^2 c\delta\rho^2 + \gamma_2^2 c\delta\rho^2 \left(2 + \frac{1}{\lambda}\right) \\
&\leq \left(1 + \lambda - 2\beta(1 - 2\beta)\frac{\mu\gamma_1}{2}\right) \left\|\boldsymbol{x}^k - \boldsymbol{x}^*\right\|^2 \\
&\quad + \frac{\gamma_1^2 \sigma^2}{G} + \gamma_1^2 c\delta\rho^2 + \frac{16\gamma_2^2 \sigma^2}{G} + 8\gamma_2^2 c\delta\rho^2 + \gamma_2^2 c\delta\rho^2 \left(2 + \frac{1}{\lambda}\right) \tag{37} \\
&\overset{0 \leq \beta \leq 1/4}{\leq} \left(1 + \lambda - \frac{\mu\gamma_2}{2}\right) \left\|\boldsymbol{x}^k - \boldsymbol{x}^*\right\|^2 + \frac{\sigma^2}{G} \left(\gamma_1^2 + 16\gamma_2^2\right) \\
&\quad + c\delta\rho^2 \left(\gamma_1^2 + 10\gamma_2^2 + \frac{\gamma_2^2}{\lambda}\right) \\
&\overset{\lambda = \mu\gamma_2/4}{\leq} \left(1 - \frac{\mu\gamma_2}{4}\right) \left\|\boldsymbol{x}^k - \boldsymbol{x}^*\right\|^2 + \frac{\sigma^2}{G} \left(\gamma_1^2 + 16\gamma_2^2\right) \\
&\quad + c\delta\rho^2 \left(\gamma_1^2 + 10\gamma_2^2 + \frac{4\gamma_2}{\mu}\right)
\end{aligned}
$$

Next, we take the full expectation from the both sides

$$\mathbb{E}\left[\left\|\boldsymbol{x}^{k+1} - \boldsymbol{x}^*\right\|^2\right] \leq \left(1 - \frac{\mu\gamma_2}{4}\right)\mathbb{E}\left[\left\|\boldsymbol{x}^k - \boldsymbol{x}^*\right\|^2\right] + \frac{\sigma^2}{G}(\gamma_1^2 + 16\gamma_2^2) + c\delta\rho^2\left(\gamma_1^2 + 10\gamma_2^2 + \frac{4\gamma_2}{\mu}\right).$$
(38)

Unrolling the recurrence, together with the bound on $\rho$ given by Lemma D.1 we derive the result of the theorem:

$$\mathbb{E}\left[\left\|\boldsymbol{x}^K - \boldsymbol{x}^*\right\|^2\right] \leq \left(1 - \frac{\mu\gamma_2}{4}\right)^K \left\|\boldsymbol{x}^0 - \boldsymbol{x}^*\right\|^2 + \frac{4\sigma^2(\gamma_1^2 + 16\gamma_2^2)}{\mu\gamma_2 G} + \frac{4c\delta\rho^2(\gamma_1^2 + 10\gamma_2^2 + \frac{4\gamma_2}{\mu})}{\mu\gamma_2}.$$

$\square$

**Corollary 3.** *Let assumptions of Theorem 2 hold. Then $\mathbb{E}\left\|\boldsymbol{x}^T - \boldsymbol{x}^*\right\|^2 \leq \varepsilon$ holds after*

$$T \geq 4\left(\frac{2}{\beta} + \frac{2\ell}{\beta\mu} + \frac{1}{3\beta c\delta G}\right)\ln\frac{3R^2}{\varepsilon}$$

*iterations of SEG-RA with $\gamma_1 = \min\left(\frac{1}{2\mu+2L}, \frac{1}{2\mu+\frac{\mu}{12c\delta G}}\right)$ and $b \geq \frac{288c\delta\sigma^2}{\beta\mu^2\varepsilon}$.*

*Proof.* Next, we plug $\gamma_2 = \beta\gamma_1 \leq \gamma_1/4$ into the result of Theorem 2 and obtain

$$\mathbb{E}\left[\left\|\boldsymbol{x}^{k+1} - \boldsymbol{x}^*\right\|^2\right] \leq \left(1 - \frac{\mu\beta\gamma_1}{4}\right)\left\|\boldsymbol{x}^0 - \boldsymbol{x}^*\right\|^2 + \frac{8\gamma_1^2\sigma^2}{\mu\beta G} + \frac{8\gamma_1 c\delta\rho^2}{\beta\mu} + \frac{16c\delta\rho^2}{\beta\mu^2}.$$ (39)

If $\zeta = 0$, $\rho^2 = 24\sigma^2$ the last reccurence can be unrolled as

$$\mathbb{E}\left\|\boldsymbol{x}^T - \boldsymbol{x}^*\right\|^2 \leq \left(1 - \frac{\mu\beta\gamma_1}{4}\right)^T \left\|\boldsymbol{x}^0 - \boldsymbol{x}^*\right\|^2 + \frac{8\gamma_1\sigma^2}{\mu\beta G} + \frac{8\cdot24\gamma_1 c\delta\sigma^2}{\beta\mu} + \frac{16\cdot24c\delta\sigma^2}{\beta\mu^2}.$$

Applying Lemma C.3 to the last bound we get the result of the corollary. $\square$

### D.3 Proofs for M-SGDA-RA

---
**Algorithm 3** M-SGDA-RA
---
**Input:** RAGG, $\gamma$, $\alpha \in [0,1]$
1: **for** $t = 0, \ldots$ **do**
2:     **for** worker $i \in [n]$ **in parallel**
3:         $\boldsymbol{g}_i^t \leftarrow \boldsymbol{g}_i(\boldsymbol{x}^t, \boldsymbol{\xi}_i)$ and $\boldsymbol{m}_i^t \leftarrow (1-\alpha)\boldsymbol{m}_i^{t-1} + \alpha\boldsymbol{g}_i^t$         // worker momentum
4:         **send** $\boldsymbol{m}_i^t$ if $i \in \mathcal{G}$, else **send** $*$ if Byzantine
5:     $\widehat{\boldsymbol{m}}^t = \text{RAGG}(\boldsymbol{m}_1^t, \ldots, \boldsymbol{m}_n^t)$ and $\boldsymbol{x}^{t+1} \leftarrow \boldsymbol{x}^t - \gamma\widehat{\boldsymbol{m}}^t$.     // update params using robust aggregate
---

#### D.3.1 Quasi-Strongly Monotone Case

**Theorem** (Theorem 3 duplicate)**.** *Let Assumptions 1, 2, 4, and 6 hold. Then after $T$ iterations M-SGDA-RA (Algorithm 3) with $(\delta, c)$-RAGG outputs $\overline{\boldsymbol{x}}^T$ such that*

$$\mathbb{E}\left[\left\|\overline{\boldsymbol{x}}^T - \boldsymbol{x}^*\right\|^2\right] \leq \frac{2\left\|\boldsymbol{x}^0 - \boldsymbol{x}^*\right\|^2}{\mu\gamma\alpha W_T} + \frac{4c\delta\rho^2}{\mu^2\alpha^2} + \frac{8\gamma c\delta\rho^2}{\mu\alpha^2} + \frac{6\gamma\sigma^2}{\mu\alpha^2 G},$$

*where $\overline{\boldsymbol{x}}^T = \frac{1}{W_T}\sum_{t=0}^T w_t\widehat{\boldsymbol{x}}^t$, $\widehat{\boldsymbol{x}}^t = \frac{\alpha}{1-(1-\alpha)^{t+1}}\sum_{j=0}^t(1-\alpha)^{t-j}\boldsymbol{x}^j$, $w_t = \left(1 - \frac{\mu\gamma\alpha}{2}\right)^{-t-1}$, and $W_T = \sum_{t=0}^T w_t$ and $\rho^2 = 24\sigma^2 + 12\zeta^2$ by Lemma D.1.*

*Proof of Theorem 3.* Since $\widehat{\boldsymbol{m}}^t = \widehat{\boldsymbol{m}}^t - \overline{\boldsymbol{m}}^t + \overline{\boldsymbol{m}}^t$ and $\overline{\boldsymbol{m}}^t = \alpha\overline{\boldsymbol{g}}^t + (1-\alpha)\overline{\boldsymbol{m}}^{t-1}$ one has

$$
\begin{aligned}
\left\|\boldsymbol{x}^{t+1} - \boldsymbol{x}^*\right\|^2 &= \left\|\boldsymbol{x}^t - \boldsymbol{x}^* - \gamma\widehat{\boldsymbol{m}}^t\right\|^2 = \left\|\boldsymbol{x}^t - \boldsymbol{x}^*\right\|^2 - 2\gamma\langle\widehat{\boldsymbol{m}}^t, \boldsymbol{x}^t - \boldsymbol{x}^*\rangle + \gamma^2\left\|\widehat{\boldsymbol{m}}^t\right\|^2 = \\
&= \left\|\boldsymbol{x}^t - \boldsymbol{x}^*\right\|^2 - 2\gamma\langle\widehat{\boldsymbol{m}}^t - \overline{\boldsymbol{m}}^t, \boldsymbol{x}^t - \boldsymbol{x}^*\rangle + \gamma^2\left\|\widehat{\boldsymbol{m}}^t\right\|^2 \\
&\quad - 2\gamma\langle\overline{\boldsymbol{m}}^t, \boldsymbol{x}^t - \boldsymbol{x}^*\rangle.
\end{aligned}
$$

Next, unrolling the following recursion

$$
\begin{aligned}
\langle\overline{\boldsymbol{m}}^t, \boldsymbol{x}^t - \boldsymbol{x}^*\rangle &= \alpha\langle\overline{\boldsymbol{g}}^t, \boldsymbol{x}^t - \boldsymbol{x}^*\rangle + (1-\alpha)\langle\overline{\boldsymbol{m}}^{t-1}, \boldsymbol{x}^t - \boldsymbol{x}^*\rangle \\
&= \alpha\langle\overline{\boldsymbol{g}}^t, \boldsymbol{x}^t - \boldsymbol{x}^*\rangle + (1-\alpha)\langle\overline{\boldsymbol{m}}^{t-1}, \boldsymbol{x}^{t-1} - \boldsymbol{x}^*\rangle + (1-\alpha)\langle\overline{\boldsymbol{m}}^{t-1}, \boldsymbol{x}^t - \boldsymbol{x}^{t-1}\rangle \\
&= \alpha\langle\overline{\boldsymbol{g}}^t, \boldsymbol{x}^t - \boldsymbol{x}^*\rangle + (1-\alpha)\langle\overline{\boldsymbol{m}}^{t-1}, \boldsymbol{x}^{t-1} - \boldsymbol{x}^*\rangle + (1-\alpha)\gamma\langle\overline{\boldsymbol{m}}^{t-1}, \widehat{\boldsymbol{m}}^t\rangle
\end{aligned}
$$

one obtains

$$
\langle\overline{\boldsymbol{m}}^t, \boldsymbol{x}^t - \boldsymbol{x}^*\rangle = \alpha\sum_{j=0}^t (1-\alpha)^{t-j}\langle\overline{\boldsymbol{g}}^j, \boldsymbol{x}^j - \boldsymbol{x}^*\rangle - (1-\alpha)\gamma\sum_{j=1}^t (1-\alpha)^{t-j}\langle\overline{\boldsymbol{m}}^{j-1}, \widehat{\boldsymbol{m}}^j\rangle
$$

Applying the latter and (11) for $\langle\widehat{\boldsymbol{m}}^t - \overline{\boldsymbol{m}}^t, \boldsymbol{x}^t - \boldsymbol{x}^*\rangle$ with $\lambda = \frac{\mu\gamma\alpha}{2}$ we obtain

$$
\begin{aligned}
2\alpha\gamma\sum_{j=0}^t &(1-\alpha)^{t-j}\langle\overline{\boldsymbol{g}}^j, \boldsymbol{x}^j - \boldsymbol{x}^*\rangle \\
&\leq \left(1 + \frac{\mu\gamma\alpha}{2}\right)\left\|\boldsymbol{x}^t - \boldsymbol{x}^*\right\|^2 - \left\|\boldsymbol{x}^{t+1} - \boldsymbol{x}^*\right\|^2 + \frac{2\gamma}{\mu\alpha}\left\|\widehat{\boldsymbol{m}}^t - \overline{\boldsymbol{m}}^t\right\|^2 \\
&\quad + \gamma^2\left\|\widehat{\boldsymbol{m}}^t\right\|^2 + 2\gamma^2(1-\alpha)\sum_{j=1}^t (1-\alpha)^{t-j}\langle\overline{\boldsymbol{m}}^{j-1}, \widehat{\boldsymbol{m}}^j\rangle \\
&\overset{(11)}{\leq} \left(1 + \frac{\mu\gamma\alpha}{2}\right)\left\|\boldsymbol{x}^t - \boldsymbol{x}^*\right\|^2 - \left\|\boldsymbol{x}^{t+1} - \boldsymbol{x}^*\right\|^2 + \frac{2\gamma}{\mu\alpha}\left\|\widehat{\boldsymbol{m}}^t - \overline{\boldsymbol{m}}^t\right\|^2 \\
&\quad + \gamma^2\left\|\widehat{\boldsymbol{m}}^t\right\|^2 + \gamma^2\sum_{j=1}^t (1-\alpha)^{t-j}\left\|\widehat{\boldsymbol{m}}^j\right\|^2 \\
&\quad + \gamma^2(1-\alpha)^2\sum_{j=1}^t (1-\alpha)^{t-j}\left\|\overline{\boldsymbol{m}}^{j-1}\right\|^2 \\
&\overset{(12)}{\leq} \left(1 + \frac{\mu\gamma\alpha}{2}\right)\left\|\boldsymbol{x}^t - \boldsymbol{x}^*\right\|^2 - \left\|\boldsymbol{x}^{t+1} - \boldsymbol{x}^*\right\|^2 + \frac{2\gamma}{\mu\alpha}\left\|\widehat{\boldsymbol{m}}^t - \overline{\boldsymbol{m}}^t\right\|^2 \\
&\quad + 4\gamma^2\sum_{j=1}^t (1-\alpha)^{t-j}\left\|\widehat{\boldsymbol{m}}^j - \overline{\boldsymbol{m}}^j\right\|^2 \\
&\quad + 3\gamma^2\sum_{j=1}^t (1-\alpha)^{t-j}\left\|\overline{\boldsymbol{m}}^{j-1}\right\|^2
\end{aligned}
$$

Since $\overline{\boldsymbol{m}}^t = \alpha\sum_{j=0}^t (1-\alpha)^{t-j}\overline{\boldsymbol{g}}^j$ and hence $\left\|\overline{\boldsymbol{m}}^t\right\|^2 \leq \alpha\sum_{j=0}^t (1-\alpha)^{t-j}\left\|\overline{\boldsymbol{g}}^j\right\|^2$ one has

$$
\begin{aligned}
2\alpha\gamma\sum_{j=0}^t &(1-\alpha)^{t-j}\langle\overline{\boldsymbol{g}}^j, \boldsymbol{x}^j - \boldsymbol{x}^*\rangle \\
&\leq \left(1 + \frac{\mu\gamma\alpha}{2}\right)\left\|\boldsymbol{x}^t - \boldsymbol{x}^*\right\|^2 - \left\|\boldsymbol{x}^{t+1} - \boldsymbol{x}^*\right\|^2 + \frac{2\gamma}{\mu\alpha}\left\|\widehat{\boldsymbol{m}}^t - \overline{\boldsymbol{m}}^t\right\|^2 \\
&\quad + 4\gamma^2\sum_{j=1}^t (1-\alpha)^{t-j}\left\|\widehat{\boldsymbol{m}}^j - \overline{\boldsymbol{m}}^j\right\|^2 \\
&\quad + 3\gamma^2\alpha\sum_{j=1}^t (1-\alpha)^{t-j}\sum_{i=0}^j (1-\alpha)^{j-i}\left\|\overline{\boldsymbol{g}}^i\right\|^2.
\end{aligned}
$$

Next by taking an expectation $\mathbb{E}_{\boldsymbol{\xi}}$ of both sides of the above inequality and rearranging terms obtain

$$
2\alpha\gamma \sum_{j=0}^{t}(1-\alpha)^{t-j}\langle F^j, \boldsymbol{x}^j - \boldsymbol{x}^*\rangle
$$
$$
\leq \left(1+\frac{\mu\gamma\alpha}{2}\right)\left\|\boldsymbol{x}^t - \boldsymbol{x}^*\right\|^2 - \mathbb{E}_{\boldsymbol{\xi}}\left\|\boldsymbol{x}^{t+1} - \boldsymbol{x}^*\right\|^2
$$
$$
+\frac{2\gamma}{\mu\alpha}\mathbb{E}_{\boldsymbol{\xi}}\left\|\widehat{\boldsymbol{m}}^t - \overline{\boldsymbol{m}}^t\right\|^2 + 4\gamma^2 \sum_{j=1}^{t}(1-\alpha)^{t-j}\mathbb{E}_{\boldsymbol{\xi}}\left\|\widehat{\boldsymbol{m}}^j - \overline{\boldsymbol{m}}^j\right\|^2
$$
$$
+3\gamma^2\alpha \sum_{j=1}^{t}(1-\alpha)^{t-j}\sum_{i=0}^{j}(1-\alpha)^{j-i}\mathbb{E}_{\boldsymbol{\xi}}\left\|\overline{\boldsymbol{g}}^i\right\|^2.
$$

Next we use Lemma C.1 to bound $\mathbb{E}_{\boldsymbol{\xi}}\left\|\overline{\boldsymbol{g}}^j\right\|^2$ and Assumption (QSM) to obtain the following bound

$$
-\alpha \sum_{j=0}^{t}(1-\alpha)^{t-j}\langle F^j, \boldsymbol{x}^j - \boldsymbol{x}^*\rangle \leq -\alpha \sum_{j=0}^{t}(1-\alpha)^{t-j}\left\|\boldsymbol{x}^j - \boldsymbol{x}^*\right\|^2 \leq -\alpha\mu\left\|\boldsymbol{x}^t - \boldsymbol{x}^*\right\|^2.
$$

Gathering the above results we have

$$
\alpha\gamma \sum_{j=0}^{t}(1-\alpha)^{t-j}\langle F^j, \boldsymbol{x}^j - \boldsymbol{x}^*\rangle
$$
$$
\leq \left(1-\frac{\mu\gamma\alpha}{2}\right)\left\|\boldsymbol{x}^t - \boldsymbol{x}^*\right\|^2 - \mathbb{E}_{\boldsymbol{\xi}}\left\|\boldsymbol{x}^{t+1} - \boldsymbol{x}^*\right\|^2
$$
$$
+\frac{2\gamma}{\mu\alpha}\mathbb{E}_{\boldsymbol{\xi}}\left\|\widehat{\boldsymbol{m}}^t - \overline{\boldsymbol{m}}^t\right\|^2 + 4\gamma^2 \sum_{j=1}^{t}(1-\alpha)^{t-j}\mathbb{E}_{\boldsymbol{\xi}}\left\|\widehat{\boldsymbol{m}}^j - \overline{\boldsymbol{m}}^j\right\|^2
$$
$$
+3\gamma^2\alpha\ell \sum_{j=1}^{t}(1-\alpha)^{t-j}\sum_{i=0}^{j}(1-\alpha)^{j-i}\langle F(\boldsymbol{x}^i), \boldsymbol{x}^i - \boldsymbol{x}^*\rangle
$$
$$
+\frac{3\gamma^2\alpha\sigma^2}{G} \sum_{j=1}^{t}(1-\alpha)^{t-j}\sum_{i=0}^{j}(1-\alpha)^{j-i}.
$$

Next we take full expectation of both sides and obtain

$$
\alpha\gamma \sum_{j=0}^{t}(1-\alpha)^{t-j}\mathbb{E}\langle F^j, \boldsymbol{x}^j - \boldsymbol{x}^*\rangle
$$
$$
\leq \left(1-\frac{\mu\gamma\alpha}{2}\right)\mathbb{E}\left\|\boldsymbol{x}^t - \boldsymbol{x}^*\right\|^2 - \mathbb{E}\left\|\boldsymbol{x}^{t+1} - \boldsymbol{x}^*\right\|^2
$$
$$
+\frac{2\gamma}{\mu\alpha}\mathbb{E}\left\|\widehat{\boldsymbol{m}}^t - \overline{\boldsymbol{m}}^t\right\|^2 + 4\gamma^2 \sum_{j=1}^{t}(1-\alpha)^{t-j}\mathbb{E}\left\|\widehat{\boldsymbol{m}}^j - \overline{\boldsymbol{m}}^j\right\|^2
$$
$$
+3\gamma^2\alpha\ell \sum_{j=1}^{t}(1-\alpha)^{t-j}\sum_{i=0}^{j}(1-\alpha)^{j-i}\mathbb{E}\langle F(\boldsymbol{x}^i), \boldsymbol{x}^i - \boldsymbol{x}^*\rangle
$$
$$
+\frac{3\gamma^2\sigma^2}{\alpha G}.
$$

Introducing the following notation

$$
\boldsymbol{Z}^t = \sum_{j=0}^{t}(1-\alpha)^{t-j}\mathbb{E}\langle F^j, \boldsymbol{x}^j - \boldsymbol{x}^*\rangle
$$

and using that $\mathbb{E}\|\widehat{\boldsymbol{m}}^t - \overline{\boldsymbol{m}}^t\|^2 \leq c\delta\rho^2$, where $\rho$ is given by (26) we have

$$
\begin{aligned}
\alpha\gamma \boldsymbol{Z}^t \;\leq\;& \left(1 - \frac{\mu\gamma\alpha}{2}\right)\mathbb{E}\|\boldsymbol{x}^t - \boldsymbol{x}^*\|^2 - \mathbb{E}\|\boldsymbol{x}^{t+1} - \boldsymbol{x}^*\|^2 + \frac{2\gamma c\delta\rho^2}{\mu\alpha} \\
&+ \frac{4\gamma^2 c\delta\rho^2}{\alpha} + 3\gamma^2\alpha\ell \sum_{j=1}^{t}(1-\alpha)^{t-j}\boldsymbol{Z}^j + \frac{3\gamma^2\sigma^2}{\alpha G}.
\end{aligned}
$$

Next we sum the above inequality $T$ times with weights $w_t = \left(1 - \frac{\mu\gamma\alpha}{2}\right)^{-t-1}$ where $W_T = \sum_{t=0}^{T} w_t$

$$
\begin{aligned}
\alpha\gamma \sum_{t=0}^{T} w_t \boldsymbol{Z}^t \;\leq\;& \left(1 - \frac{\mu\gamma\alpha}{2}\right)\sum_{t=0}^{T} w_t\mathbb{E}\|\boldsymbol{x}^t - \boldsymbol{x}^*\|^2 - \sum_{t=0}^{T} w_t\mathbb{E}\|\boldsymbol{x}^{t+1} - \boldsymbol{x}^*\|^2 \\
&+ 3\gamma^2\alpha\ell \sum_{t=0}^{T} w_t \sum_{j=0}^{t}(1-\alpha)^{t-j}\boldsymbol{Z}^j \\
&+ W_T\left(\frac{2\gamma c\delta\rho^2}{\mu\alpha} + \frac{4\gamma^2 c\delta\rho^2}{\alpha} + \frac{3\gamma^2\sigma^2}{\alpha G}\right).
\end{aligned}
$$

Since $\left(1 - \frac{\mu\gamma\alpha}{2}\right)w_t = w_{t-1}$

$$
\begin{aligned}
\alpha\gamma \sum_{t=0}^{T} w_t \boldsymbol{Z}^t \;\leq\;& \|\boldsymbol{x}^0 - \boldsymbol{x}^*\|^2 + 3\gamma^2\alpha\ell \sum_{t=0}^{T} w_t \sum_{j=0}^{t}(1-\alpha)^{t-j}\boldsymbol{Z}^j \\
&+ W_T\left(\frac{2\gamma c\delta\rho^2}{\mu\alpha} + \frac{4\gamma^2 c\delta\rho^2}{\alpha} + \frac{3\gamma^2\sigma^2}{\alpha G}\right).
\end{aligned}
$$

If $\gamma \leq \frac{1}{\mu}$ we have

$$
w_t = \left(1 - \frac{\mu\gamma\alpha}{2}\right)^{-t+i}w_i \leq \left(1 + \frac{\mu\gamma\alpha}{2}\right)^{t-i}w_i \leq \left(1 + \frac{\alpha}{2}\right)^{t-i}w_i.
$$

So we have

$$
\begin{aligned}
\alpha\gamma \sum_{t=0}^{T} w_t \boldsymbol{Z}^t \;\leq\;& \|\boldsymbol{x}^0 - \boldsymbol{x}^*\|^2 \\
&+ 3\gamma^2\alpha\ell \sum_{t=0}^{T}\sum_{j=0}^{t}\left(1 + \frac{\alpha}{2}\right)^{t-j}(1-\alpha)^{t-j}w_j \boldsymbol{Z}^j \\
&+ W_T\left(\frac{2\gamma c\delta\rho^2}{\mu\alpha} + \frac{4\gamma^2 c\delta\rho^2}{\alpha} + \frac{3\gamma^2\sigma^2}{\alpha G}\right) \\
\;\leq\;& \|\boldsymbol{x}^0 - \boldsymbol{x}^*\|^2 + 3\gamma^2\alpha\ell \sum_{t=0}^{T}\sum_{j=0}^{t}\left(1 - \frac{\alpha}{2}\right)^{t-j}w_j \boldsymbol{Z}^j \\
&+ W_T\left(\frac{2\gamma c\delta\rho^2}{\mu\alpha} + \frac{4\gamma^2 c\delta\rho^2}{\alpha} + \frac{3\gamma^2\sigma^2}{\alpha G}\right) \\
\;\leq\;& \|\boldsymbol{x}^0 - \boldsymbol{x}^*\|^2 \\
&+ 3\gamma^2\alpha\ell\left(\sum_{t=0}^{T} w_t \boldsymbol{Z}^t\right)\left(\sum_{t=0}^{\infty}\left(1 - \frac{\alpha}{2}\right)^t\right) \\
&+ W_T\left(\frac{2\gamma c\delta\rho^2}{\mu\alpha} + \frac{4\gamma^2 c\delta\rho^2}{\alpha} + \frac{3\gamma^2\sigma^2}{\alpha G}\right) \\
\;\leq\;& \|\boldsymbol{x}^0 - \boldsymbol{x}^*\|^2 + 6\gamma^2\ell\left(\sum_{t=0}^{T} w_t \boldsymbol{Z}^t\right) \\
&+ W_T\left(\frac{2\gamma c\delta\rho^2}{\mu\alpha} + \frac{4\gamma^2 c\delta\rho^2}{\alpha} + \frac{3\gamma^2\sigma^2}{\alpha G}\right).
\end{aligned}
$$

If $\gamma \leq \frac{\alpha}{12\ell}$ then the following is true

$$\frac{\alpha\gamma}{2}\sum_{t=0}^{T}w_t \boldsymbol{Z}^t \leq \left\|\boldsymbol{x}^0 - \boldsymbol{x}^*\right\|^2 + W_T\left(\frac{2\gamma c\delta\rho^2}{\mu\alpha} + \frac{4\gamma^2 c\delta\rho^2}{\alpha} + \frac{3\gamma^2\sigma^2}{\alpha G}\right). \tag{40}$$

Using the notations for $\boldsymbol{Z}^t$ and (QSM) we have

$$\boldsymbol{Z}^t = \sum_{j=0}^{t}(1-\alpha)^{t-j}\mathbb{E}\langle F^j, \boldsymbol{x}^j - \boldsymbol{x}^*\rangle \geq \mu\sum_{j=0}^{t}(1-\alpha)^{t-j}\mathbb{E}\|\boldsymbol{x}^j - \boldsymbol{x}^*\|.$$

and consequently by Jensen's inequality

$$\boldsymbol{Z}^t \geq \mu\sum_{j=0}^{t}(1-\alpha)^{t-j}\mathbb{E}\|\boldsymbol{x}^j - \boldsymbol{x}^*\| \geq \mu\frac{1-(1-\alpha)^{t+1}}{\alpha}\mathbb{E}\left\|\boldsymbol{x}^* - \sum_{j=0}^{t}\frac{\alpha(1-\alpha)^{t-j}}{1-(1-\alpha)^{t+1}}\boldsymbol{x}^j\right\|.$$

With the definition $\widehat{\boldsymbol{x}}^t = \frac{\alpha}{1-(1-\alpha)^{t+1}}\sum_{j=0}^{t}(1-\alpha)^{t-j}\boldsymbol{x}^j$ then the above implies that

$$\boldsymbol{Z}^t \geq \mu\mathbb{E}\left\|\widehat{\boldsymbol{x}}^t - \boldsymbol{x}^*\right\|,$$

that together with (40) gives

$$\frac{\alpha\gamma\mu}{2}\sum_{t=0}^{T}w_t\mathbb{E}\left\|\widehat{\boldsymbol{x}}^t - \boldsymbol{x}^*\right\| \leq \left\|\boldsymbol{x}^0 - \boldsymbol{x}^*\right\|^2 + W_T\left(\frac{2\gamma c\delta\rho^2}{\mu\alpha} + \frac{4\gamma^2 c\delta\rho^2}{\alpha} + \frac{3\gamma^2\sigma^2}{\alpha G}\right). \tag{41}$$

Applying the Jensen inequality again with $\overline{\boldsymbol{x}}^T = \frac{1}{W_T}\sum_{t=0}^{T}w_t\widehat{\boldsymbol{x}}^t$ we derive the final result

$$\mathbb{E}\left[\left\|\overline{\boldsymbol{x}}^T - \boldsymbol{x}^*\right\|^2\right] \leq \frac{2\left\|\boldsymbol{x}^0 - \boldsymbol{x}^*\right\|^2}{\mu\gamma\alpha W_T} + \frac{4c\delta\rho^2}{\mu^2\alpha^2} + \frac{8\gamma c\delta\rho^2}{\mu\alpha^2} + \frac{6\gamma\sigma^2}{\mu\alpha^2 G}, \tag{42}$$

together with the bound on $\rho$ given by Lemma D.1. $\qquad\square$

**Corollary 4.** *Let assumptions of Theorem 3 hold. Then* $\mathbb{E}\left\|\overline{\boldsymbol{x}}^T - \boldsymbol{x}^*\right\|^2 \leq \varepsilon$ *holds after*

$$T \geq \frac{1}{\alpha}\left(4 + \frac{24\ell}{\mu\alpha} + \frac{1}{8c\delta G}\right)\ln\frac{3R^2}{\varepsilon}$$

*iterations of* M-SGDA-RA *with* $\gamma = \min\left(\frac{\alpha}{12\ell}, \frac{1}{2\mu + \frac{\mu}{16c\delta G}}\right)$.

*Proof.* If $\zeta = 0$, $\rho^2 = 24\sigma^2$ the result of Theorem 3 can be simplified as

$$\mathbb{E}\left[\left\|\overline{\boldsymbol{x}}^T - \boldsymbol{x}^*\right\|^2\right] \leq \frac{2\left\|\boldsymbol{x}^0 - \boldsymbol{x}^*\right\|^2}{\mu\gamma\alpha W_T} + \frac{4\cdot 24c\delta\sigma^2}{\mu^2\alpha^2} + \frac{8\cdot 24\gamma c\delta\sigma^2}{\mu\alpha^2} + \frac{6\gamma\sigma^2}{\mu\alpha^2 G}.$$

Since $\frac{2}{\mu\gamma\alpha W_T} \leq \left(1 - \frac{\mu\gamma\alpha}{2}\right)^{T+1}$ we can apply Lemma C.3 and get the result of the corollary.

$\qquad\square$

# E  Methods with random check of computations

We replace $(\delta_{\max}, c)$-RAGG with the simple mean, but introduce additional verification that have to be passed to accept the mean. The advantage of such aggregation that it coincides with "good" mean if there are no peers violating the protocol But if there is at least one peer violating the protocol at iteration $t$ we can bound the variance similar to Lemma D.1.

---

**Algorithm 4** CheckComputations

---

**Input:** $t$, $\mathcal{G}_t \cup \mathcal{B}_t$, $\mathcal{C}_t$, $\text{Banned}_t = \varnothing$
1: $\mathcal{C}_{t+1} = \{c_1^{t+1}, \ldots, c_m^{t+1}\}$, $\mathcal{C}_{t+1} \subset (\mathcal{G}_t \cup \mathcal{B}_t) \setminus \mathcal{C}_t$ and $\mathcal{U}_{t+1} = \{u_1^{t+1}, \ldots, u_m^{t+1}\}$, $\mathcal{U}_{t+1} \subset (\mathcal{G}_t \cup \mathcal{B}_t) \setminus \mathcal{C}_t$, where $2m$ workers $c_1^{t+1}, \ldots, c_m^{t+1}, u_1^{t+1}, \ldots, u_m^{t+1}$ are choisen uniformly at random without replacement.
2: **for** $i = 1, ..., m$ **in parallel** $c_i^{t+1}$ checks computations of $u_i^{t+1}$ during the next iteration
3:     $c_i^{t+1}$ receives a query to recalculate $g(x^t, \xi_{u_i^{t+1}}^t)$
4:     $c_i^{t+1}$ sends the recalculated $g(x^t, \xi_{u_i^{t+1}}^t)$

5: **for** $i = 1, ..., m$ **do**
6:     **if** $g(x^t, \xi_{u_i^t}^t) \neq g_{u_i^t}^t$ **then**
7:         $\text{Banned}_t = \text{Banned}_t \cup \{u_i^t, c_i^t\}$.
8:     **end if**
**Output:** $\mathcal{C}_{t+1}$, $\mathcal{G}_t \cup \mathcal{B}_t \setminus \text{Banned}_t$

---

**Lemma E.1.** *Let Assumption 2 is satisfied with $\zeta = 0$. Then the error between the ideal average $\overline{g}^t$ and the average with the recomputation rule $g^t$ can be bounded as*

$$\mathbb{E}_{\xi} \left\| \widehat{g}^t - \overline{g}^t \right\|^2 \leq \rho^2 \mathbb{1}_t,$$

*where $\rho^2 = q\sigma^2$ with $q = 2C^2 + 12 + \frac{12}{n - 2B - m}$ and $C = \mathcal{O}(1)$.*

*Proof of Lemma E.1.* Denote the set $\widetilde{\mathcal{G}}$ in the following way $\widetilde{\mathcal{G}} = \{i \in \mathcal{G}_t \setminus \mathcal{C}_t : \|\widehat{g}^t - g_i^t\| \leq C\sigma\}$.

$$\widehat{g}^t = \begin{cases} \frac{1}{n} \sum_{i=1}^n g_i^t, & \text{if number of workers} > \frac{n}{2}, \\ \textbf{recompute}, & \text{otherwise.} \end{cases}$$

So that we have

$$
\begin{aligned}
\left\| \widehat{g}^t - \overline{g}^t \right\|^2 &= \left\| \widehat{g}^t - \frac{1}{|\widetilde{\mathcal{G}}|} \sum_{i \in \widetilde{\mathcal{G}}} g_i^t + \frac{1}{|\widetilde{\mathcal{G}}|} \sum_{i \in \widetilde{\mathcal{G}}} g_i^t - \overline{g}^t \right\|^2 \\
&\leq 2 \left\| \widehat{g}^t - \frac{1}{|\widetilde{\mathcal{G}}|} \sum_{i \in \widetilde{\mathcal{G}}} g_i^t \right\|^2 + 2 \left\| \frac{1}{|\widetilde{\mathcal{G}}|} \sum_{i \in \widetilde{\mathcal{G}}} g_i^t - \overline{g}^t \right\|^2 \\
&\leq 2C^2\sigma^2 + 2 \left\| \frac{1}{|\widetilde{\mathcal{G}}|} \sum_{i \in \widetilde{\mathcal{G}}} g_i^t - \overline{g}^t \right\|^2
\end{aligned}
$$

If $\delta \leq 1/4$ then an acceptance of $\widehat{g}^t$ implies that $|\widetilde{\mathcal{G}}| > n/4$ and $|\widetilde{\mathcal{G}}| > |\mathcal{G}_t \setminus \mathcal{C}_t|/3$.

$$
\begin{aligned}
\left\| \frac{1}{|\widetilde{\mathcal{G}}|} \sum_{i \in \widetilde{\mathcal{G}}} g_i^t - \overline{g}^t \right\|^2 &\leq \frac{1}{|\widetilde{\mathcal{G}}|} \sum_{i \in \widetilde{\mathcal{G}}} \|g_i^t - \overline{g}^t\|^2 \leq \frac{1}{|\widetilde{\mathcal{G}}|} \sum_{i \in \mathcal{G}_t \setminus \mathcal{C}_t} \|g_i^t - \overline{g}^t\|^2 \\
&\leq \frac{3}{|\mathcal{G}_t \setminus \mathcal{C}_t|} \sum_{i \in \mathcal{G}_t \setminus \mathcal{C}_t} \|g_i^t - \overline{g}^t\|^2
\end{aligned}
$$

Bringing the above results together gives that

$$\mathbb{E}\big\|\widehat{\boldsymbol{g}}^t - \overline{\boldsymbol{g}}^t\big\|^2 \leq 2C^2\sigma^2 + \frac{6}{|\mathcal{G}_t \setminus \mathcal{C}_t|} \sum_{i \in \mathcal{G}} \mathbb{E}\big\|\boldsymbol{g}_i^t - \overline{\boldsymbol{g}}^t\big\|^2$$

Since checks of computations are only possible in homogeneous case ($\zeta = 0$) then $\mathbb{E}\big\|\boldsymbol{g}_i^t - F^t\big\|^2 = \sigma^2$ and

$$\mathbb{E}_{\boldsymbol{\xi}}\big\|\boldsymbol{g}_i^t - \overline{\boldsymbol{g}}^t\big\|^2 \leq 2\mathbb{E}_{\boldsymbol{\xi}}\big\|\boldsymbol{g}_i^t - F^t\big\|^2 + 2\mathbb{E}_{\boldsymbol{\xi}}\big\|F^t - \overline{\boldsymbol{g}}^t\big\|^2 \leq 2\sigma^2 + \frac{2\sigma^2}{|\mathcal{G}|}. \tag{43}$$

Since $|\mathcal{G}_t \setminus \mathcal{C}_t| > n - 2B - m$

$$\mathbb{E}_{\boldsymbol{\xi}}\big\|\widehat{\boldsymbol{g}}^t - \overline{\boldsymbol{g}}^t\big\|^2 \leq 2C^2\sigma^2 + 12\sigma^2 + \frac{12\sigma^2}{|\mathcal{G}_t \setminus \mathcal{C}_t|} \leq 2C^2\sigma^2 + 12\sigma^2 + \frac{12\sigma^2}{n - 2B - m}.$$

$\square$

## E.1 Proofs for SGDA-CC

---
**Algorithm 5** SGDA-CC
---
**Input:** $\gamma$
1: $\mathcal{C}_0 = \varnothing$
2: **for** $t = 1, \ldots$ **do**
3:     **for** worker $i \in (\mathcal{G}_t \cup \mathcal{B}_t) \setminus \mathcal{C}_t$ **in parallel**
4:         **send** $\boldsymbol{g}_i^t = \begin{cases} \boldsymbol{g}_i(\boldsymbol{x}^t, \boldsymbol{\xi}_i), & \text{if } i \in \mathcal{G}_t \setminus \mathcal{C}_t, \\ *, & \text{if } i \in \mathcal{B}_t \setminus \mathcal{C}_t, \end{cases}$
5:     $\widehat{\boldsymbol{g}}^t = \frac{1}{\mathcal{W}_t} \sum_{i \in \mathcal{W}_t} \boldsymbol{g}_i^t, \ \mathcal{W}_t = (\mathcal{G}_t \cup \mathcal{B}_t) \setminus \mathcal{C}_t$
6:
7:     **if** $|\{i \in \mathcal{W}_t \mid \|\widehat{\boldsymbol{g}}^t - \boldsymbol{g}_i^t\| \leq C\sigma\}| \geq |\mathcal{W}_t|/2$ **then**
8:         $\boldsymbol{x}^{t+1} \leftarrow \boldsymbol{x}^t - \gamma\widehat{\boldsymbol{g}}^t.$
9:     **else**
10:         **recompute**
11:     **end if**
12:     $\mathcal{C}_{t+1}, \mathcal{G}_{t+1} \cup \mathcal{B}_{t+1} = \mathsf{CheckComputations}(\mathcal{C}_t, \mathcal{G}_t \cup \mathcal{B}_t)$
---

### E.1.1 Star Co-coercieve Case

Next we provide convergence guarantees for SGDA-CC (Algorithm 5) under Assumption 6.

**Theorem 9.** *Let Assumptions 1 and 6 hold. Next, assume that*

$$\gamma = \min\left\{ \frac{1}{2\ell}, \sqrt{\frac{(n - 2B - m)R^2}{6\sigma^2 K}}, \sqrt{\frac{m^2 R^2}{72\rho^2 B^2 n^2}} \right\} \tag{44}$$

*where $\rho^2 = q\sigma^2$ with $q = 2C^2 + 12 + \frac{12}{n - 2B - m}$ and $C = \mathcal{O}(1)$ by Lemma E.1 and $R \geq \|\boldsymbol{x}^0 - \boldsymbol{x}^*\|$. Then after $K$ iterations of SGDA-CC (Algorithm 5) it outputs $\boldsymbol{x}^T$ such that*

$$\sum_{k=0}^{K-1} \mathbb{E}\big\|F(\boldsymbol{x}^k)\big\| \leq \ell \sum_{k=0}^{K-1} \mathbb{E}[\langle F(\boldsymbol{x}^k), \boldsymbol{x}^k - \boldsymbol{x}^*\rangle] \leq \frac{2\ell R^2}{\gamma}.$$

*Proof.* Since $|\mathcal{G}_t \setminus \mathcal{C}_t| \geq n - 2B - m$ one can derive using the results of Lemmas C.1 and E.1 that

$$\begin{aligned}
\mathbb{E}\left[\|\boldsymbol{x}^{k+1} - \boldsymbol{x}^*\|^2 \mid \boldsymbol{x}^k\right] &= \mathbb{E}\left[\|\boldsymbol{x}^k - \boldsymbol{x}^* - \gamma\widehat{\boldsymbol{g}}^k\|^2 \mid \boldsymbol{x}^k\right] \\
&= \|\boldsymbol{x}^k - \boldsymbol{x}^*\|^2 - 2\gamma\mathbb{E}\left[\langle \boldsymbol{x}^k - \boldsymbol{x}^*, \widehat{\boldsymbol{g}}^k\rangle \mid \boldsymbol{x}^k\right] + \gamma^2\mathbb{E}\left[\|\widehat{\boldsymbol{g}}^k\|^2 \mid \boldsymbol{x}^k\right] \\
&\leq \|\boldsymbol{x}^k - \boldsymbol{x}^*\|^2 - 2\gamma\langle \boldsymbol{x}^k - \boldsymbol{x}^*, F(\boldsymbol{x}^k)\rangle + 2\ell\gamma^2\langle \boldsymbol{x}^k - \boldsymbol{x}^*, F(\boldsymbol{x}^k)\rangle \\
&\quad - 2\gamma\mathbb{E}\left[\langle \boldsymbol{x}^k - \boldsymbol{x}^*, \widehat{\boldsymbol{g}}^k - \overline{\boldsymbol{g}}^k\rangle \mid \boldsymbol{x}^k\right] + 2\gamma^2\rho^2\mathbb{1}_k + \frac{2\gamma^2\sigma^2}{n - 2B - m},
\end{aligned}$$

where $\mathbb{1}_k$ is an indicator function of the event that at least 1 Byzantine peer violates the protocol at iteration $k$.

To estimate the inner product in the right-hand side we apply Cauchy-Schwarz inequality and then Lemma E.1

$$
\begin{aligned}
-2\gamma\mathbb{E}\left[\langle \boldsymbol{x}^k - \boldsymbol{x}^*, \widehat{\boldsymbol{g}}^k - \overline{\boldsymbol{g}}^k \rangle \mid \boldsymbol{x}^k\right] &\leq 2\gamma\|\boldsymbol{x}^k - \boldsymbol{x}^*\|\mathbb{E}\left[\|\widehat{\boldsymbol{g}}^k - \overline{\boldsymbol{g}}^k\| \mid \boldsymbol{x}^k\right] \\
&\leq 2\gamma\|\boldsymbol{x}^k - \boldsymbol{x}^*\|\sqrt{\mathbb{E}\left[\|\widehat{\boldsymbol{g}}^k - \overline{\boldsymbol{g}}^k\|^2 \mid \boldsymbol{x}^k\right]} \\
&\leq 2\gamma\rho\|\boldsymbol{x}^k - \boldsymbol{x}^*\|\mathbb{1}_k.
\end{aligned}
$$

Since $\gamma \leq \frac{1}{2\ell}$ the above results imlies

$$
\begin{aligned}
\gamma\langle \boldsymbol{x}^k - \boldsymbol{x}^*, F(\boldsymbol{x}^k)\rangle &\leq \|\boldsymbol{x}^k - \boldsymbol{x}^*\|^2 - \mathbb{E}\left[\|\boldsymbol{x}^{k+1} - \boldsymbol{x}^*\|^2 \mid \boldsymbol{x}^k\right] \\
&\quad -2\gamma\mathbb{E}\left[\langle \boldsymbol{x}^k - \boldsymbol{x}^*, \widehat{\boldsymbol{g}}^k - \overline{\boldsymbol{g}}^k \rangle \mid \boldsymbol{x}^k\right] + 2\gamma^2\rho^2\mathbb{1}_k + \frac{2\gamma^2\sigma^2}{n - 2B - m}. \\
&\leq \|\boldsymbol{x}^k - \boldsymbol{x}^*\|^2 - \mathbb{E}\left[\|\boldsymbol{x}^{k+1} - \boldsymbol{x}^*\|^2 \mid \boldsymbol{x}^k\right] \\
&\quad +2\gamma^2\rho^2\mathbb{1}_k + \frac{2\gamma^2\sigma^2}{n - 2B - m} + 2\gamma\rho\|\boldsymbol{x}^k - \boldsymbol{x}^*\|\mathbb{1}_k.
\end{aligned}
$$

Taking the full expectation from the both sides of the above inequality and summing up the results for $k = 0, 1, \ldots, K - 1$ we derive

$$
\begin{aligned}
\frac{\gamma}{K}\sum_{k=0}^{K-1}&\mathbb{E}[\langle F(\boldsymbol{x}^k), \boldsymbol{x}^k - \boldsymbol{x}^*\rangle] \\
&\leq \frac{1}{K}\sum_{k=0}^{K-1}\left(\mathbb{E}\left[\|\boldsymbol{x}^k - \boldsymbol{x}^*\|^2\right] - \mathbb{E}\left[\|\boldsymbol{x}^{k+1} - \boldsymbol{x}^*\|^2\right]\right) + \frac{2\gamma^2\sigma^2}{n - 2B - m} \\
&\quad +\frac{2\gamma\rho}{K}\sum_{k=0}^{K-1}\mathbb{E}\left[\|\boldsymbol{x}^k - \boldsymbol{x}^*\|\mathbb{1}_k\right] + \frac{2\gamma^2\rho^2}{K}\sum_{k=0}^{K-1}\mathbb{E}[\mathbb{1}_k] \\
&\leq \frac{\|\boldsymbol{x}^0 - \boldsymbol{x}^*\|^2 - \mathbb{E}[\|\boldsymbol{x}^K - \boldsymbol{x}^*\|^2]}{K} + \frac{2\gamma^2\sigma^2}{n - 2B - m} \\
&\quad +\frac{2\gamma\rho}{K}\sum_{k=0}^{K-1}\sqrt{\mathbb{E}\left[\|\boldsymbol{x}^k - \boldsymbol{x}^*\|^2\right]\mathbb{E}[\mathbb{1}_k]} + \frac{2\gamma^2\rho^2}{K}\sum_{k=0}^{K-1}\mathbb{E}[\mathbb{1}_k].
\end{aligned}
$$

Since $F$ satisfies Assumption 6, $\sum_{k=0}^{K-1}\mathbb{E}[\langle F(\boldsymbol{x}^k), \boldsymbol{x}^k - \boldsymbol{x}^*\rangle] \geq 0$. Using this and new notation $R_k = \|\boldsymbol{x}^k - \boldsymbol{x}^*\|, k > 0, R_0 \geq \|\boldsymbol{x}^0 - \boldsymbol{x}^*\|$ we get

$$
\begin{aligned}
0 \leq &\frac{R_0^2 - \mathbb{E}[R_K^2]}{K} + \frac{2\gamma^2\sigma^2}{n - 2B - m} \\
&+\frac{2\gamma\rho}{K}\sum_{k=0}^{K-1}\sqrt{\mathbb{E}\left[R_k^2\right]\mathbb{E}[\mathbb{1}_k]} + \frac{2\gamma^2\rho^2}{K}\sum_{k=0}^{K-1}\mathbb{E}[\mathbb{1}_k]
\end{aligned}
\tag{45}
$$

implying (after changing the indices) that

$$
\mathbb{E}[R_k^2] \leq R_0^2 + \frac{2\gamma^2\sigma^2 k}{n - 2B - m} + 2\gamma\rho\sum_{l=0}^{k-1}\sqrt{\mathbb{E}\left[R_l^2\right]\mathbb{E}[\mathbb{1}_l]} + 2\gamma^2\rho^2\sum_{l=0}^{k-1}\mathbb{E}[\mathbb{1}_l]
\tag{46}
$$

holds for all $k \geq 0$. In the remaining part of the proof we derive by induction that

$$
R_0^2 + \frac{2\gamma^2\sigma^2 k}{n - 2B - m} + 2\gamma\rho\sum_{l=0}^{k-1}\sqrt{\mathbb{E}\left[R_l^2\right]\mathbb{E}[\mathbb{1}_l]} + 2\gamma^2\rho^2\sum_{l=0}^{k-1}\mathbb{E}[\mathbb{1}_k] \leq 2R_0^2
\tag{47}
$$

for all $k = 0, \ldots, K$. For $k = 0$ this inequality trivially holds. Next, assume that it holds for all $k = 0, 1, \ldots, T-1, T \leq K-1$. Let us show that it holds for $k = T$ as well. From (46) and (47) we have that $\mathbb{E}[R_k^2] \leq 2R_0^2$ for all $k = 0, 1, \ldots, T-1$. Therefore,

$$
\begin{aligned}
\mathbb{E}[R_T^2] &\leq R_0^2 + \frac{2\gamma^2\sigma^2 T}{n - 2B - m} + 2\gamma\rho \sum_{l=0}^{T-1} \sqrt{\mathbb{E}[R_l^2]\,\mathbb{E}[\mathbb{1}_l]} + 2\gamma^2\rho^2 \sum_{l=0}^{T-1} \mathbb{E}[\mathbb{1}_l] \\
&\leq R_0^2 + \frac{2\gamma^2\sigma^2 T}{n - 2B - m} + 2\sqrt{2}\gamma\rho R_0 \sum_{l=0}^{T-1} \sqrt{\mathbb{E}[\mathbb{1}_l]} + 2\gamma^2\rho^2 \sum_{l=0}^{T-1} \mathbb{E}[\mathbb{1}_l].
\end{aligned}
$$

If a Byzantine peer deviates from the protocol at iteration $k$, it will be detected with some probability $p_k$ during the next iteration. One can lower bound this probability as

$$
p_k \geq m \cdot \frac{G_k}{n_k} \cdot \frac{1}{n_k} = \frac{m(1 - \delta_k)}{n_k} \geq \frac{m}{n}.
$$

Therefore, each individual Byzantine worker can violate the protocol no more than $1/p$ times on average implying that

$$
\mathbb{E}[R_T^2] \leq R_0^2 + \frac{2\gamma^2\sigma^2 T}{n - 2B - m} + \frac{2\sqrt{2}\gamma\rho R_0 nB}{m} + \frac{2\gamma^2\rho^2 nB}{m}
$$

Taking

$$
\gamma = \min\left\{ \frac{1}{2\ell}, \sqrt{\frac{(n - 2B - m)R_0^2}{6\sigma^2 K}}, \sqrt{\frac{m^2 R_0^2}{72\rho^2 B^2 n^2}} \right\}
$$

we ensure that

$$
\frac{2\gamma^2\sigma^2 T}{n - 2B - m} + \frac{2\sqrt{2}\gamma\rho R_0 nB}{m} + \frac{2\gamma^2\rho^2 nB}{m} \leq \frac{R_0^2}{3} + \frac{R_0^2}{3} + \frac{R_0^2}{3} = R_0^2,
$$

and, as a result, we get

$$
\mathbb{E}[R_T^2] \leq 2R_0^2 \equiv 2R \tag{48}
$$

. Therefore, (47) holds for all $k = 0, 1, \ldots, K$. Together with (45) it implies

$$
\sum_{k=0}^{K-1} \mathbb{E}[\langle F(\boldsymbol{x}^k), \boldsymbol{x}^k - \boldsymbol{x}^* \rangle] \leq \frac{2R_0^2}{\gamma}.
$$

The last inequality together with Assumption 6 implies

$$
\sum_{k=0}^{K-1} \mathbb{E}\big\| F(\boldsymbol{x}^k) \big\| \leq \frac{2\ell R_0^2}{\gamma}.
$$

$\square$

**Corollary 5.** *Let assumptions of Theorem 9 hold. Then $\frac{1}{K} \sum_{k=0}^{K-1} \mathbb{E}\big\| F(\boldsymbol{x}^k) \big\| \leq \varepsilon$ holds after*

$$
K = \mathcal{O}\left( \frac{\ell^2 R^2}{\varepsilon} + \frac{\sigma^2 \ell^2 R^2}{n\varepsilon^2} + \frac{\sigma n^2 \ell R}{m\varepsilon} \right)
$$

*iterations of* SGDA-CC.

*Proof.*

$$
\begin{aligned}
\frac{1}{K} \sum_{k=0}^{K-1} \mathbb{E}\big\| F(\boldsymbol{x}^k) \big\| \leq \frac{2\ell R^2}{\gamma K} &\leq \frac{2\ell R^2}{K}\left( 2\ell + \sqrt{\frac{6\sigma^2 K}{(n - 2B - m)R^2}} + \sqrt{\frac{72\rho^2 B^2 n^2}{m^2 R^2}} \right) \\
&\leq \frac{4\ell^2 R^2}{K} + \sqrt{\frac{24\sigma^2 \ell^2 R^2}{(n - 2B - m)K}} + \frac{17\rho Bn\ell R}{mK}
\end{aligned}
$$

Let us chose $K$ such that each of the last three terms less or equal $\varepsilon/3$, then

$$K = \max\left(\frac{6\ell^2 R^2}{\varepsilon}, \frac{216\sigma^2\ell^2 R^2}{(n-2B-m)\varepsilon^2}, \frac{51\rho Bn\ell R}{m\varepsilon}\right)$$

where $\rho^2 = q\sigma^2$ with $q = 2C^2 + 12 + \frac{12}{n-2B-m}$ and $C = \mathcal{O}(1)$ by Lemma E.1. The latter implies that

$$\frac{1}{K}\sum_{k=0}^{K-1}\mathbb{E}\big\|F(\boldsymbol{x}^k)\big\| \le \varepsilon.$$

Using the definition of $\rho$ from Lemma E.1 and if $B \le \frac{n}{4}$, $m << n$ the bound for $K$ can be easily derived. $\qquad\square$

### E.1.2 Quasi-Strongly Monotone Case

**Theorem** (Theorem 4 duplicate). *Let Assumptions 1, 4 and 6 hold. Then after $T$ iterations* SGDA-CC *(Algorithm 5) with $\gamma \le \frac{1}{2\ell}$ outputs $\boldsymbol{x}^T$ such that*

$$\mathbb{E}\|\boldsymbol{x}^{T+1} - \boldsymbol{x}^*\|^2 \le \left(1 - \frac{\gamma\mu}{2}\right)^{T+1}\|\boldsymbol{x}^0 - \boldsymbol{x}^*\|^2 + \frac{4\gamma\sigma^2}{\mu(n-2B-m)} + \frac{2\rho^2 nB}{m}\left(\frac{\gamma}{\mu} + \gamma^2\right).$$

*where $\rho^2 = q\sigma^2$ with $q = 2C^2 + 12 + \frac{12}{n-2B-m}$ and $C = \mathcal{O}(1)$ by Lemma E.1.*

*Proof of Theorem 4.* The proof is similar to the proof of Theorem 1

$$\mu\mathbb{E}\left[\big\|\overline{\boldsymbol{x}}^K - \boldsymbol{x}^*\big\|^2\right] = \mu\mathbb{E}\left[\left\|\frac{1}{K}\sum_{k=0}^{K-1}(\boldsymbol{x}^k - \boldsymbol{x}^*)\right\|^2\right] \le \mu\mathbb{E}\left[\frac{1}{K}\sum_{k=0}^{K-1}\|\boldsymbol{x}^k - \boldsymbol{x}^*\|^2\right]$$

$$= \frac{\mu}{K}\sum_{k=0}^{K-1}\mathbb{E}\left[\big\|\boldsymbol{x}^k - \boldsymbol{x}^*\big\|^2\right] \overset{(QSM)}{\le} \frac{1}{K}\sum_{k=0}^{K-1}\mathbb{E}\big[\langle F(\boldsymbol{x}^k), \boldsymbol{x}^k - \boldsymbol{x}^*\rangle\big]$$

Since $\widehat{\boldsymbol{g}}^t = \widehat{\boldsymbol{g}}^t - F^t + F^t$ one has

$$\|\boldsymbol{x}^{t+1} - \boldsymbol{x}^*\|^2 = \|\boldsymbol{x}^t - \boldsymbol{x}^*\|^2 - 2\gamma\langle\widehat{\boldsymbol{g}}^t - \overline{\boldsymbol{g}}^t, \boldsymbol{x}^t - \boldsymbol{x}^*\rangle - 2\gamma\langle\overline{\boldsymbol{g}}^t, \boldsymbol{x}^t - \boldsymbol{x}^*\rangle + \gamma^2\|\widehat{\boldsymbol{g}}^t\|^2.$$

Applying (11) for $\langle\overline{\boldsymbol{g}}^t, \boldsymbol{x}^t - \boldsymbol{x}^*\rangle$ with $\lambda = \frac{\gamma\mu}{2}$ and (12) for $\|\widehat{\boldsymbol{g}}^t\|^2 = \|\widehat{\boldsymbol{g}}^t - \overline{\boldsymbol{g}}^t + \overline{\boldsymbol{g}}^t\|^2$ we derive

$$\|\boldsymbol{x}^{t+1} - \boldsymbol{x}^*\|^2 \le \left(1 + \frac{\gamma\mu}{2}\right)\|\boldsymbol{x}^t - \boldsymbol{x}^*\|^2 - 2\gamma\langle\overline{\boldsymbol{g}}^t, \boldsymbol{x}^t - \boldsymbol{x}^*\rangle$$
$$+ \frac{2\gamma}{\mu}\|\widehat{\boldsymbol{g}}^t - \overline{\boldsymbol{g}}^t\|^2 + 2\gamma^2\|\widehat{\boldsymbol{g}}^t - \overline{\boldsymbol{g}}^t\|^2 + 2\gamma^2\|\overline{\boldsymbol{g}}^t\|^2.$$

Next by taking an expectation $\mathbb{E}_{\boldsymbol{\xi}}$ of both sides of the above inequality and rearranging terms obtain

$$\mathbb{E}_{\boldsymbol{\xi}}\|\boldsymbol{x}^{t+1} - \boldsymbol{x}^*\|^2 \le \left(1 + \frac{\gamma\mu}{2}\right)\|\boldsymbol{x}^t - \boldsymbol{x}^*\|^2 - 2\gamma\langle F(x^t), \boldsymbol{x}^t - \boldsymbol{x}^*\rangle$$
$$+ \frac{2\gamma}{\mu}\mathbb{E}_{\boldsymbol{\xi}}\|\widehat{\boldsymbol{g}}^t - \overline{\boldsymbol{g}}^t\|^2 + 2\gamma^2\mathbb{E}_{\boldsymbol{\xi}}\|\widehat{\boldsymbol{g}}^t - \overline{\boldsymbol{g}}^t\|^2 + 2\gamma^2\mathbb{E}_{\boldsymbol{\xi}}\|\overline{\boldsymbol{g}}^t\|^2.$$

The difference with the proof of Theorem 1 is that we suppose that the number of peer violating the protocol at an iteration $t$ is known to any "good" peer. So the result of Lemma E.1 writes as follows

$$\mathbb{E}_{\boldsymbol{\xi}}\|\widehat{\boldsymbol{g}}^t - \overline{\boldsymbol{g}}^t\|^2 \le \rho^2\mathbb{1}_t,$$

where $\mathbb{1}_t$ is an indicator function of the event that at least 1 Byzantine peer violates the protocol at iteration $t$.

Together with Lemma C.1 we can proceed as follows m

$$\mathbb{E}_{\boldsymbol{\xi}}\|\boldsymbol{x}^{t+1} - \boldsymbol{x}^*\|^2 \le \left(1 + \frac{\gamma\mu}{2}\right)\|\boldsymbol{x}^t - \boldsymbol{x}^*\|^2 + \left(2\gamma^2\ell - 2\gamma\right)\langle F(\boldsymbol{x}^t), \boldsymbol{x}^t - \boldsymbol{x}^*\rangle$$
$$+ \frac{2\gamma^2\sigma^2}{G} + 2\mathbb{1}_t\rho^2\left(\frac{\gamma}{\mu} + \gamma^2\right),$$

Since $\gamma \leq \frac{1}{2\ell}$ and Assumption (QSM) holds we derive

$$\mathbb{E}_{\boldsymbol{\xi}}\left\|\boldsymbol{x}^{t+1} - \boldsymbol{x}^*\right\|^2 \leq \left(1 - \frac{\gamma\mu}{2}\right)\left\|\boldsymbol{x}^t - \boldsymbol{x}^*\right\|^2 + \frac{2\gamma^2\sigma^2}{G} + 2\mathbb{1}_t\rho^2\left(\frac{\gamma}{\mu} + \gamma^2\right).$$

Next we take full expectation of both sides and obtain

$$\mathbb{E}\left\|\boldsymbol{x}^{t+1} - \boldsymbol{x}^*\right\|^2 \leq \left(1 - \frac{\gamma\mu}{2}\right)\mathbb{E}\left\|\boldsymbol{x}^t - \boldsymbol{x}^*\right\|^2 + \frac{2\gamma^2\sigma^2}{n - 2B - m} + 2\rho^2\left(\frac{\gamma}{\mu} + \gamma^2\right)\mathbb{E}\mathbb{1}_t.$$

The latter implies

$$\mathbb{E}\left\|\boldsymbol{x}^{T+1} - \boldsymbol{x}^*\right\|^2 \leq \left(1 - \frac{\gamma\mu}{2}\right)^{T+1}\left\|\boldsymbol{x}^0 - \boldsymbol{x}^*\right\|^2 + \frac{4\gamma\sigma^2}{\mu(n - 2B - m)}$$
$$+ 2\rho^2\left(\frac{\gamma}{\mu} + \gamma^2\right)\sum_i^T \mathbb{E}\mathbb{1}_i\left(1 - \frac{\gamma\mu}{2}\right)^{T-i}.$$

If a Byzantine peer deviates from the protocol at iteration $t$, it will be detected with some probability $p_t$ during the next iteration. One can lower bound this probability as

$$p_t \geq m \cdot \frac{G_t}{n_t} \cdot \frac{1}{n_t} = \frac{m(1 - \delta_t)}{n_t} \geq \frac{m}{n}.$$

Therefore, each individual Byzantine worker can violate the protocol no more than $^1/_p$ times on average implying that

$$\mathbb{E}\left[\sum_{t=0}^{\infty}\mathbb{1}_t\right] \leq \frac{nB}{m}$$

that implies

$$\mathbb{E}\left[\sum_i^T\mathbb{1}_i\left(1 - \frac{\gamma\mu}{2}\right)^{T-i}\right] \leq \mathbb{E}\left[\sum_i^T\mathbb{1}_i\right] \leq \frac{nB}{m}. \tag{49}$$

The latter together with the above bound on $\mathbb{E}\left\|\boldsymbol{x}^{T+1} - \boldsymbol{x}^*\right\|^2$ implies the result of the theorem.

$$\mathbb{E}\left\|\boldsymbol{x}^{T+1} - \boldsymbol{x}^*\right\|^2 \leq \left(1 - \frac{\gamma\mu}{2}\right)^{T+1}\left\|\boldsymbol{x}^0 - \boldsymbol{x}^*\right\|^2 + \frac{4\gamma\sigma^2}{\mu(n - 2B - m)} + \frac{2\rho^2 nB}{m}\left(\frac{\gamma}{\mu} + \gamma^2\right).$$

$\square$

**Corollary 6.** *Let assumptions of Theorem 4 hold. Then* $\mathbb{E}\left\|\boldsymbol{x}^T - \boldsymbol{x}^*\right\|^2 \leq \varepsilon$ *holds after*

$$T = \widetilde{\mathcal{O}}\left(\frac{\ell}{\mu} + \frac{\sigma^2}{\mu^2(n - 2B - m)\varepsilon} + \frac{q\sigma^2 Bn}{\mu^2 m\varepsilon} + \frac{q\sigma^2 Bn}{\mu^2 m\sqrt{\varepsilon}}\right)$$

*iterations of* SGDA-CC *(Alg. 5) with*

$$\gamma = \min\left\{\frac{1}{2\ell}, \frac{2\ln\left(\max\left\{2, \min\left\{\frac{m(n - 2B - m)\mu^2 R^2 K}{8m\sigma^2 + 4q\sigma^2 nB(n - 2B - m)}, \frac{m\mu^2 R^2 K^2}{8qnB\sigma^2}\right\}\right\}\right)}{\mu(K + 1)}\right\}.$$

*Proof.* Using the definition of $\rho$ $(\rho^2 = q\sigma^2 = \mathcal{O}(\sigma^2))$ from Lemma E.1 and if $B \leq \frac{n}{4}$, $m << n$ the result of Theorem 4 can be simplified as

$$\mathbb{E}\left\|\boldsymbol{x}^{T+1} - \boldsymbol{x}^*\right\|^2 \leq \left(1 - \frac{\gamma\mu}{2}\right)^{T+1}\left\|\boldsymbol{x}^0 - \boldsymbol{x}^*\right\|^2 + \frac{4\gamma\sigma^2}{\mu(n - 2B - m)} + \frac{2q\sigma^2 nB}{m}\left(\frac{\gamma}{\mu} + \gamma^2\right).$$

Applying Lemma C.4 to the last bound we get the result of the corollary. $\square$

### E.1.3 Monotone Case

**Theorem 10.** *Suppose the assumptions of Theorem 9 and Assumption 5 hold. Next, assume that*

$$\gamma = \min\left\{\frac{1}{2\ell}, \sqrt{\frac{(n - 2B - m)R^2}{6\sigma^2 K}}, \sqrt{\frac{m^2 R^2}{72\rho^2 B^2 n^2}}\right\} \tag{50}$$

*where $\rho^2 = q\sigma^2$ with $q = 2C^2 + 12 + \frac{12}{n-2B-m}$ and $C = \mathcal{O}(1)$ by Lemma E.1 and $R \geq \|\boldsymbol{x}^0 - \boldsymbol{x}^*\|$. Then after $K$ iterations of* SGDA-CC *(Algorithm 5)*

$$\mathbb{E}\left[\text{Gap}_{B_R(x^*)}\left(\overline{\boldsymbol{x}}^K\right)\right] \leq \frac{3R^2}{\gamma K}, \tag{51}$$

*where* $\text{Gap}_{B_R(x^*)}\left(\overline{\boldsymbol{x}}^K\right) = \max\limits_{u \in B_R(x^*)} \left\langle F(\boldsymbol{u}), \overline{\boldsymbol{x}}^K - \boldsymbol{u}\right\rangle$, $\overline{\boldsymbol{x}}^K = \frac{1}{K}\sum\limits_{k=0}^{K-1} \boldsymbol{x}^k$ and $R \geq \|\boldsymbol{x}^0 - \boldsymbol{x}^*\|$.

*Proof.* Combining (16), (14) one can derive

$$\begin{aligned}
2\gamma\left\langle F(\boldsymbol{x}^k), \boldsymbol{x}^k - \boldsymbol{u}\right\rangle &\leq \|\boldsymbol{x}^k - \boldsymbol{u}\|^2 - \|\boldsymbol{x}^{k+1} - \boldsymbol{u}\|^2 \\
&\quad -2\gamma\left\langle\widehat{\boldsymbol{g}}^k - \overline{\boldsymbol{g}}^k, \boldsymbol{x}^k - \boldsymbol{u}\right\rangle - 2\gamma\left\langle\overline{\boldsymbol{g}}^k - F^k, \boldsymbol{x}^k - \boldsymbol{u}\right\rangle \\
&\quad +2\gamma^2\|\widehat{\boldsymbol{g}}^k - \overline{\boldsymbol{g}}^k\|^2 + 2\gamma^2\|\overline{\boldsymbol{g}}^k\|^2.
\end{aligned}$$

Assumption 5 implies that

$$\left\langle F(\boldsymbol{u}), \boldsymbol{x}^k - \boldsymbol{u}\right\rangle \leq \left\langle F(\boldsymbol{x}^k), \boldsymbol{x}^k - \boldsymbol{u}\right\rangle \tag{52}$$

and consequently by Jensen inequality

$$\begin{aligned}
2\gamma K\left\langle F(\boldsymbol{u}), \overline{\boldsymbol{x}}^K - \boldsymbol{u}\right\rangle &\leq \|\boldsymbol{x}^0 - \boldsymbol{u}\|^2 - 2\gamma\sum_{k=0}^{K-1}\left\langle\widehat{\boldsymbol{g}}^k - \overline{\boldsymbol{g}}^k, \boldsymbol{x}^k - \boldsymbol{u}\right\rangle \\
&\quad -2\gamma\sum_{k=0}^{K-1}\left\langle\overline{\boldsymbol{g}}^k - F^k, \boldsymbol{x}^k - \boldsymbol{u}\right\rangle + 2\gamma^2\sum_{k=0}^{K-1}\left(\|\widehat{\boldsymbol{g}}^k - \overline{\boldsymbol{g}}^k\|^2 + \|\overline{\boldsymbol{g}}^k\|^2\right).
\end{aligned}$$

Then maximization in $\boldsymbol{u}$ gives

$$\begin{aligned}
2\gamma K\text{Gap}_{B_R(x^*)}\left(\overline{\boldsymbol{x}}^K\right) &\leq \max_{u \in B_R(x^*)}\|\boldsymbol{x}^0 - \boldsymbol{u}\|^2 + 2\gamma^2\sum_{k=0}^{K-1}\left(\|\widehat{\boldsymbol{g}}^k - \overline{\boldsymbol{g}}^k\|^2 + \|\overline{\boldsymbol{g}}^k\|^2\right) \\
&\quad +2\gamma\max_{u \in B_R(x^*)}\left(\sum_{k=0}^{K-1}\left\langle\overline{\boldsymbol{g}}^k - \widehat{\boldsymbol{g}}^k, \boldsymbol{x}^k - \boldsymbol{u}\right\rangle\right) \\
&\quad +2\gamma\max_{u \in B_R(x^*)}\left(\sum_{k=0}^{K-1}\left\langle F^k - \overline{\boldsymbol{g}}^k, \boldsymbol{x}^k - \boldsymbol{u}\right\rangle\right).
\end{aligned}$$

Taking the full expectation from the both sides of the previous inequality gives

$$\begin{aligned}
2\gamma K\mathbb{E}\left[\text{Gap}_{B_R(x^*)}\left(\overline{\boldsymbol{x}}^K\right)\right] &\leq \max_{u \in B_R(x^*)}\|\boldsymbol{x}^0 - \boldsymbol{u}\|^2 \\
&\quad +2\gamma\mathbb{E}\left[\max_{u \in B_R(x^*)}\left(\sum_{k=0}^{K-1}\left\langle\overline{\boldsymbol{g}}^k - \widehat{\boldsymbol{g}}^k, \boldsymbol{x}^k - \boldsymbol{u}\right\rangle\right)\right] \\
&\quad +2\gamma\mathbb{E}\left[\max_{u \in B_R(x^*)}\left(\sum_{k=0}^{K-1}\left\langle F^k - \overline{\boldsymbol{g}}^k, \boldsymbol{x}^k - \boldsymbol{u}\right\rangle\right)\right] \\
&\quad +2\gamma^2\mathbb{E}\left[\sum_{k=0}^{K-1}\left(\|\widehat{\boldsymbol{g}}^k - \overline{\boldsymbol{g}}^k\|^2 + \|\overline{\boldsymbol{g}}^k\|^2\right)\right]
\end{aligned}$$

Firstly obtain the bound for the terms that do not depend on $\boldsymbol{u}$ using Assumption 1, Lemma E.1 and Theorem 9

$$2\gamma^2 \mathbb{E}\left[\sum_{k=0}^{K-1}\left(\left\|\widehat{\boldsymbol{g}}^k - \overline{\boldsymbol{g}}^k\right\|^2 + \left\|\overline{\boldsymbol{g}}^k\right\|^2\right)\right]$$

$$\leq 2\gamma^2 \mathbb{E}\left[\sum_{k=0}^{K-1}\left\|\widehat{\boldsymbol{g}}^k - \overline{\boldsymbol{g}}^k\right\|^2\right] + 2\gamma^2 \mathbb{E}\left[\sum_{k=0}^{K-1}\left\|\overline{\boldsymbol{g}}^k\right\|^2\right]$$

$$\leq 2\gamma^2 \rho^2 \mathbb{E}\left[\sum_{k=0}^{K-1}\mathbb{1}_k\right] + \frac{2\gamma^2 K\sigma^2}{|\mathcal{G}_t \setminus \mathcal{C}_t|} + 2\gamma^2\ell\sum_{k=0}^{K-1}\mathbb{E}\left\langle F^k, \boldsymbol{x}^k - \boldsymbol{x}^*\right\rangle$$

$$\leq \frac{2\gamma^2 nBc\rho^2}{m} + \frac{2\gamma^2 K\sigma^2}{|\mathcal{G}_t \setminus \mathcal{C}_t|} + 4\ell\gamma R^2.$$

Since $\mathbb{E}\left[\left\|\boldsymbol{x}^k - \boldsymbol{u}\right\|\right] \leq \mathbb{E}\left[\left\|\boldsymbol{x}^k - \boldsymbol{x}^*\right\|\right] + \left\|\boldsymbol{x}^* - \boldsymbol{u}\right\| \leq \mathbb{E}\left[\left\|\boldsymbol{x}^k - \boldsymbol{x}^*\right\|\right] + \max_{u \in B_R(x^*)}\left\|\boldsymbol{x}^* - \boldsymbol{u}\right\| \overset{(48)}{\leq} 3R$ one can derive that

$$2\gamma\mathbb{E}\left[\max_{u \in B_R(x^*)}\left(\sum_{k=0}^{K-1}\left\langle \overline{\boldsymbol{g}}^k - \widehat{\boldsymbol{g}}^k, \boldsymbol{x}^k - \boldsymbol{u}\right\rangle\right)\right]$$

$$\leq 2\gamma\mathbb{E}\left[\max_{u \in B_R(x^*)}\left(\sum_{k=0}^{K-1}\left\langle \overline{\boldsymbol{g}}^k - \widehat{\boldsymbol{g}}^k, \boldsymbol{x}^* - \boldsymbol{u}\right\rangle\right) + \sum_{k=0}^{K-1}\left\langle \overline{\boldsymbol{g}}^k - \widehat{\boldsymbol{g}}^k, \boldsymbol{x}^k - \boldsymbol{x}^*\right\rangle\right]$$

$$\leq 2\gamma\sum_{k=0}^{K-1}\mathbb{E}\left[\max_{u \in B_R(x^*)}\left\langle \overline{\boldsymbol{g}}^k - \widehat{\boldsymbol{g}}^k, \boldsymbol{x}^* - \boldsymbol{u}\right\rangle\right] + 2\gamma\mathbb{E}\left[\sum_{k=0}^{K-1}\left\langle \overline{\boldsymbol{g}}^k - \widehat{\boldsymbol{g}}^k, \boldsymbol{x}^k - \boldsymbol{x}^*\right\rangle\right]$$

$$\leq 2\gamma\sum_{k=0}^{K-1}\mathbb{E}\left[\max_{u \in B_R(x^*)}\left\|\overline{\boldsymbol{g}}^k - \widehat{\boldsymbol{g}}^k\right\|\left\|\boldsymbol{x}^* - \boldsymbol{u}\right\|\right] + 2\gamma\mathbb{E}\left[\mathbb{E}\left[\sum_{k=0}^{K-1}\left\|\overline{\boldsymbol{g}}^k - \widehat{\boldsymbol{g}}^k\right\|\left\|\boldsymbol{x}^k - \boldsymbol{x}^*\right\| \mid \boldsymbol{x}^k\right]\right]$$

$$\leq 2\gamma\sum_{k=0}^{K-1}\mathbb{E}\left[R\left\|\overline{\boldsymbol{g}}^k - \widehat{\boldsymbol{g}}^k\right\|\right] + 2\gamma\mathbb{E}\left[\sum_{k=0}^{K-1}\rho\mathbb{1}_k\left\|\boldsymbol{x}^k - \boldsymbol{x}^*\right\|\right]$$

$$\leq 2\gamma\rho R\mathbb{E}\left[\sum_{k=0}^{K-1}\mathbb{1}_k\right] + 4\gamma\rho R\mathbb{E}\left[\sum_{k=0}^{K-1}\mathbb{1}_k\right] \leq 6\gamma R\rho\mathbb{E}\left[\sum_{k=0}^{K-1}\mathbb{1}_k\right] \leq \frac{6nB\gamma R\rho}{m}$$

Following Beznosikov et al. [2023] one can derive he bound for the next term:

$$\mathbb{E}\left[\sum_{k=0}^{K-1}\left\langle F^k - \overline{\boldsymbol{g}}^k, \boldsymbol{x}^k\right\rangle\right] = \mathbb{E}\left[\sum_{k=0}^{K-1}\left\langle \mathbb{E}[F^k - \overline{\boldsymbol{g}}^k \mid x^k], \boldsymbol{x}^k\right\rangle\right] = 0,$$

$$\mathbb{E}\left[\sum_{k=0}^{K-1}\left\langle F^k - \overline{\boldsymbol{g}}^k, \boldsymbol{x}^0\right\rangle\right] = \sum_{k=0}^{K-1}\left\langle \mathbb{E}[F^k - \overline{\boldsymbol{g}}^k], \boldsymbol{x}^0\right\rangle = 0,$$

we have

$$2\gamma\mathbb{E}\left[\max_{\boldsymbol{u}\in B_R(x^*)}\sum_{k=0}^{K-1}\langle F^k-\overline{\boldsymbol{g}}^k,\boldsymbol{x}^k-\boldsymbol{u}\rangle\right]$$

$$= 2\gamma\mathbb{E}\left[\sum_{k=0}^{K-1}\langle F^k-\overline{\boldsymbol{g}}^k,\boldsymbol{x}^k\rangle\right]+2\gamma\mathbb{E}\left[\max_{\boldsymbol{u}\in B_R(x^*)}\sum_{k=0}^{K-1}\langle F^k-\overline{\boldsymbol{g}}^k,-\boldsymbol{u}\rangle\right]$$

$$= 2\gamma\mathbb{E}\left[\max_{\boldsymbol{u}\in B_R(x^*)}\sum_{k=0}^{K-1}\langle F^k-\overline{\boldsymbol{g}}^k,-\boldsymbol{u}\rangle\right]$$

$$= 2\gamma\mathbb{E}\left[\sum_{k=0}^{K-1}\langle F^k-\overline{\boldsymbol{g}}^k,\boldsymbol{x}^0\rangle\right]$$

$$\quad +2\gamma\mathbb{E}\left[\max_{\boldsymbol{u}\in B_R(x^*)}\sum_{k=0}^{K-1}\langle F^k-\overline{\boldsymbol{g}}^k,-\boldsymbol{u}\rangle\right]$$

$$= 2\gamma K\mathbb{E}\left[\max_{\boldsymbol{u}\in B_R(x^*)}\left\langle\frac{1}{K}\sum_{k=0}^{K-1}(F^k-\overline{\boldsymbol{g}}^k),\boldsymbol{x}^0-\boldsymbol{u}\right\rangle\right]$$

$$\overset{(11)}{\leq} 2\gamma K\mathbb{E}\left[\max_{\boldsymbol{u}\in B_R(x^*)}\left\{\frac{\gamma}{2}\left\|\frac{1}{K}\sum_{k=0}^{K-1}(F^k-\overline{\boldsymbol{g}}^k)\right\|^2+\frac{1}{2\gamma}\|\boldsymbol{x}^0-\boldsymbol{u}\|^2\right\}\right]$$

$$= \gamma^2\mathbb{E}\left[\left\|\sum_{k=0}^{K-1}(F^k-\overline{\boldsymbol{g}}^k)\right\|^2\right]+\max_{\boldsymbol{u}\in B_R(x^*)}\|\boldsymbol{x}^0-\boldsymbol{u}\|^2.$$

We notice that $\mathbb{E}[F^k-\overline{\boldsymbol{g}}^k\mid F^0-\overline{\boldsymbol{g}}^0,\ldots,F^{k-1}-\overline{\boldsymbol{g}}^{k-1}]=0$ for all $k\geq 1$, i.e., conditions of Lemma C.2 are satisfied. Therefore, applying Lemma C.2, we get

$$2\gamma\mathbb{E}\left[\max_{\boldsymbol{u}\in B_R(x^*)}\sum_{k=0}^{K-1}\langle F^k-\overline{\boldsymbol{g}}^k,\boldsymbol{x}^k-\boldsymbol{u}\rangle\right] \leq \gamma^2\sum_{k=0}^{K-1}\mathbb{E}[\|F^k-\overline{\boldsymbol{g}}^k\|^2]$$

$$\quad +\max_{\boldsymbol{u}\in B_R(x^*)}\|\boldsymbol{x}^0-\boldsymbol{u}\|^2 \tag{53}$$

$$\leq \frac{\gamma^2 K\sigma^2}{|\mathcal{G}_t\setminus\mathcal{C}_t|}+\max_{\boldsymbol{u}\in B_R(x^*)}\|\boldsymbol{x}^0-\boldsymbol{u}\|^2. \tag{54}$$

Assembling the above results together gives

$$2\gamma K\mathbb{E}\left[\texttt{Gap}_{B_R(x^*)}(\overline{\boldsymbol{x}}^K)\right]$$

$$\leq 2\max_{u\in B_R(x^*)}\|\boldsymbol{x}^0-\boldsymbol{u}\|^2+\frac{2\gamma^2 nB\rho^2}{m}+\frac{3\gamma^2 K\sigma^2}{|\mathcal{G}_t\setminus\mathcal{C}_t|}+4\ell\gamma R^2+\frac{6nB\gamma R\rho}{m}$$

$$\leq 2\max_{u\in B_R(x^*)}\|\boldsymbol{x}^0-\boldsymbol{u}\|^2+\frac{2\gamma^2 nB\rho^2}{m}+\frac{3\gamma^2 K\sigma^2}{n-2B-m}$$

$$\quad +4\ell\gamma R^2+\frac{6nB\gamma R\rho}{m}\overset{(44)}{\leq} 6R^2.$$

$\square$

**Corollary 7.** *Let assumptions of Theorem 10 hold. Then* $\mathbb{E}\left[\texttt{Gap}_{B_R(x^*)}(\overline{\boldsymbol{x}}^K)\right]\leq\varepsilon$ *holds after*

$$K=\mathcal{O}\left(\frac{\ell R^2}{\varepsilon}+\frac{\sigma^2 R^2}{n\varepsilon^2}+\frac{\sigma n^2 R}{m\varepsilon}\right)$$

*iterations of* SGDA-CC.

*Proof.*

$$\mathbb{E}\left[\mathtt{Gap}_{B_R(x^*)}\left(\overline{\boldsymbol{x}}^K\right)\right] \leq \frac{3R^2}{\gamma K} \quad \leq \quad \frac{3R^2}{K}\left(2\ell + \sqrt{\frac{6\sigma^2 K}{(n-2B-m)R^2}} + \sqrt{\frac{72\rho^2 B^2 n^2}{m^2 R^2}}\right)$$

$$\leq \quad \frac{6\ell R^2}{K} + \sqrt{\frac{54\sigma^2 R^2}{(n-2B-m)K}} + \frac{26\rho BnR}{mK}$$

Let us chose $K$ such that each of the last three terms less or equal $\varepsilon/3$, then

$$K = \max\left(\frac{18\ell R^2}{\varepsilon}, \frac{9 \cdot 54\sigma^2 R^2}{(n-2B-m)\varepsilon^2}, \frac{78\rho BnR}{m\varepsilon}\right)$$

guarantees that

$$\mathbb{E}\left[\mathtt{Gap}_{B_R(x^*)}\left(\overline{\boldsymbol{x}}^K\right)\right] \leq \varepsilon.$$

Using the definition of $\rho$ from Lemma E.1 and if $B \leq \frac{n}{4}$, $m << n$ the bound for $K$ can be easily derived. $\qquad\square$

## E.2 Proofs for R-SGDA-CC

### E.2.1 Quasi-Strongly Monotone Case

---
**Algorithm 6** R-SGDA-CC
---
**Input:** $\boldsymbol{x}^0$ – starting point, $r$ – number of restarts, $\{\gamma_t\}_{t=1}^r$ – stepsizes for SGDA-CC (Alg. 5), $\{K_t\}_{t=1}^r$ – number of iterations for SGDA-CC (Alg. 5)
1: $\widehat{\boldsymbol{x}}^0 = \boldsymbol{x}^0$
2: **for** $t = 1, 2, \ldots, r$ **do**
3:     Run SGDA-CC (Alg. 5) for $K_t$ iterations with stepsize $\gamma_t$, starting point $\widehat{\boldsymbol{x}}^{t-1}$.
4:     Define $\widehat{\boldsymbol{x}}^t$ as $\widehat{\boldsymbol{x}}^t = \frac{1}{K_t}\sum_{k=0}^{K_t} \boldsymbol{x}^{k,t}$, where $\boldsymbol{x}^{0,t}, \boldsymbol{x}^{1,t}, \ldots, \boldsymbol{x}^{K_t,t}$ are the iterates produced by SGDA-CC (Alg. 5).
5: **end for**
**Output:** $\widehat{\boldsymbol{x}}^r$

---

**Theorem** (Theorem 5 duplicate). *Let Assumptions 1, 4 and 6 hold. Then, after $r = \left\lceil \log_2 \frac{R^2}{\varepsilon} \right\rceil - 1$ restarts R-SGDA-CC (Algorithm 6) with $\gamma_t = \min\left\{\frac{1}{2\ell}, \sqrt{\frac{(n-2B-m)R^2}{6\sigma^2 2^t K_t}}, \sqrt{\frac{m^2 R^2}{72 q \sigma^2 2^t B^2 n^2}}\right\}$ and $K_t = \left\lceil \max\left\{\frac{8\ell}{\mu}, \frac{96\sigma^2 2^t}{(n-2B-m)\mu^2 R^2}, \frac{34n\sigma B\sqrt{q2^t}}{m\mu R}\right\}\right\rceil$, where $R \geq \|\boldsymbol{x}^0 - \boldsymbol{x}^*\|$, outputs $\widehat{\boldsymbol{x}}^r$ such that $\mathbb{E}\|\widehat{\boldsymbol{x}}^r - \boldsymbol{x}^*\|^2 \leq \varepsilon$. Moreover, the total number of executed iterations of SGDA-CC is*

$$\sum_{t=1}^r K_t = \mathcal{O}\left(\frac{\ell}{\mu}\log\frac{\mu R_0^2}{\varepsilon} + \frac{\sigma^2}{(n-2B-m)\mu\varepsilon} + \frac{nB\sigma}{m\sqrt{\mu\varepsilon}}\right). \qquad (55)$$

*With $q = 2C^2 + 12 + \frac{12}{n-2B-m}$ and $C = \mathcal{O}(1)$ by Lemma E.1.*

*Proof of Theorem 5.* $\overline{\boldsymbol{x}}^K = \frac{1}{K}\sum_{k=0}^{K-1} \boldsymbol{x}^k$

$$\mu\mathbb{E}\left[\|\overline{\boldsymbol{x}}^K - \boldsymbol{x}^*\|^2\right] = \mu\mathbb{E}\left[\left\|\frac{1}{K}\sum_{k=0}^{K-1}(\boldsymbol{x}^k - \boldsymbol{x}^*)\right\|^2\right] \leq \mu\mathbb{E}\left[\frac{1}{K}\sum_{k=0}^{K-1}\|\boldsymbol{x}^k - \boldsymbol{x}^*\|^2\right]$$

$$= \frac{\mu}{K}\sum_{k=0}^{K-1}\mathbb{E}\left[\|\boldsymbol{x}^k - \boldsymbol{x}^*\|^2\right] \overset{\text{(QSM)}}{\leq} \frac{1}{K}\sum_{k=0}^{K-1}\mathbb{E}\left[\langle F(\boldsymbol{x}^k), \boldsymbol{x}^k - \boldsymbol{x}^*\rangle\right]$$

Theorem 9 implies that SGDA-CC with

$$\gamma = \min\left\{\frac{1}{2\ell}, \sqrt{\frac{(n-2B-m)R_0^2}{6\sigma^2 K}}, \sqrt{\frac{m^2 R_0^2}{72\rho^2 B^2 n^2}}\right\}$$

guarantees

$$\mu\mathbb{E}\left[\left\|\overline{\boldsymbol{x}}^K - \boldsymbol{x}^*\right\|^2\right] \leq \frac{2}{R_0^2}\gamma K$$

after $K$ iterations.

After the first restart we have

$$\mathbb{E}\left[\left\|\widehat{\boldsymbol{x}}^1 - \boldsymbol{x}^*\right\|^2\right] \leq \frac{2R_0^2}{\mu\gamma_1 K_1} \leq \frac{R_0^2}{2}.$$

Next, assume that we have $\mathbb{E}[\|\widehat{\boldsymbol{x}}^t - \boldsymbol{x}^*\|^2] \leq \frac{R_0^2}{2^t}$ for some $t \leq r-1$. Then, Theorem 9 implies that

$$\mathbb{E}\left[\left\|\widehat{\boldsymbol{x}}^{t+1} - \boldsymbol{x}^*\right\|^2 \mid \widehat{\boldsymbol{x}}^t\right] \leq \frac{2\|\widehat{\boldsymbol{x}}^t - \boldsymbol{x}^*\|^2}{\mu\gamma_t K_t}.$$

Taking the full expectation from the both sides of previous inequality we get

$$\mathbb{E}\left[\left\|\widehat{\boldsymbol{x}}^{t+1} - \boldsymbol{x}^*\right\|^2\right] \leq \frac{2\mathbb{E}[\|\widehat{\boldsymbol{x}}^t - \boldsymbol{x}^*\|^2]}{\mu\gamma_t K_t} \leq \frac{2R_0^2}{2^t\mu\gamma_t K_t} \leq \frac{R_0^2}{2^{t+1}}.$$

Therefore, by mathematical induction we have that for all $t = 1,\ldots,r$

$$\mathbb{E}\left[\|\widehat{\boldsymbol{x}}^t - \boldsymbol{x}^*\|^2\right] \leq \frac{R_0^2}{2^t}.$$

Then, after $r = \left\lceil\log_2\frac{R_0^2}{\varepsilon}\right\rceil - 1$ restarts of SGDA-CC we have $\mathbb{E}\left[\|\widehat{\boldsymbol{x}}^r - \boldsymbol{x}^*\|^2\right] \leq \varepsilon$. The total number of iterations executed by SGDA-CC is

$$\begin{aligned}
\sum_{t=1}^{r} K_t &= \mathcal{O}\left(\sum_{t=1}^{r}\max\left\{\frac{\ell}{\mu}, \frac{\sigma^2 2^t}{(n-2B-m)\mu^2 R_0^2}, \frac{nB\rho 2^{\frac{t}{2}}}{m\mu R_0}\right\}\right) \\
&= \mathcal{O}\left(\frac{\ell}{\mu}r + \frac{\sigma^2 2^r}{(n-2B-m)\mu^2 R_0^2} + \frac{nB\rho 2^{\frac{r}{2}}}{m\mu R_0}\right) \\
&= \mathcal{O}\left(\frac{\ell}{\mu}\log\frac{\mu R_0^2}{\varepsilon} + \frac{\sigma^2}{(n-2B-m)\mu^2 R_0^2}\cdot\frac{\mu R_0^2}{\varepsilon} + \frac{nB\rho}{m\mu R_0}\cdot\sqrt{\frac{\mu R_0^2}{\varepsilon}}\right) \\
&= \mathcal{O}\left(\frac{\ell}{\mu}\log\frac{\mu R_0^2}{\varepsilon} + \frac{\sigma^2}{(n-2B-m)\mu\varepsilon} + \frac{nB\rho}{m\sqrt{\mu\varepsilon}}\right).
\end{aligned}$$

$\square$

**Corollary 8.** *Let assumptions of 5 hold. Then* $\mathbb{E}\left[\|\widehat{\boldsymbol{x}}^r - \boldsymbol{x}^*\|^2\right] \leq \varepsilon$ *holds after*

$$\sum_{t=1}^{r} K_t = \mathcal{O}\left(\frac{\ell}{\mu}\log\frac{\mu R^2}{\varepsilon} + \frac{\sigma^2}{n\mu\varepsilon} + \frac{n^2\sigma}{m\sqrt{\mu\varepsilon}}\right) \tag{56}$$

*iterations of* SGDA-CC.

*Proof.* Using the definition of $\rho$ from Lemma E.1 and if $B \leq \frac{n}{4}$, $m << n$ the bound for $\sum_{t=1}^{r} K_t$ can be easily derived. $\square$

## E.3 Proofs for SEG-CC

---

**Algorithm 7** SEG-CC

---

**Input:** RAGG, $\gamma$

1: **for** $t = 1, \ldots$ **do**
2:      **for** worker $i \in [n]$ **in parallel**
3:          $g_{\xi_i}^t \leftarrow g_i(x^t, \xi_i)$
4:          **send** $g_{\xi_i}^t$ if $i \in \mathcal{G}_t$, else **send** $*$ if Byzantine
5:      $\widehat{g}_{\xi^t}(x^t) = \frac{1}{\mathcal{W}_{t-\frac{1}{2}}} \sum_{i \in \mathcal{W}_{t-\frac{1}{2}}} g_{\xi_i}^t, \quad \mathcal{W}_{t-\frac{1}{2}} = (\mathcal{G}_t \cup \mathcal{B}_t) \setminus \mathcal{C}_t$
6:      **if** $\left| \left\{ i \in \mathcal{W}_{t-\frac{1}{2}} \mid \left\| \widehat{g}_{\xi^t}(x^t) - g_{\xi_i}^t \right\| \leq C\sigma \right\} \right| \geq \left| \mathcal{W}_{t-\frac{1}{2}} \right|/2$ **then**
7:          $\widetilde{x}^t \leftarrow x^t - \gamma_1 \widehat{g}_{\xi^t}(x^t)$
8:      **else**
9:          **recompute**
10:      **end if**
11:      $\mathcal{C}_{t+\frac{1}{2}}, \mathcal{G}_{t+\frac{1}{2}} \cup \mathcal{B}_{t+\frac{1}{2}} = \mathsf{CheckComputations}(\mathcal{C}_t, \mathcal{G}_t \cup \mathcal{B}_t)$
12:      **for** worker $i \in [n]$ **in parallel**
13:          $g_{\eta_i}^t \leftarrow g_i(\widetilde{x}^t, \eta_i)$
14:          **send** $g_{\eta_i}^t$ if $i \in \mathcal{G}$, else **send** $*$ if Byzantine
15:      $\widehat{g}_{\eta^t}(\widetilde{x}^t) = \frac{1}{\mathcal{W}_t} \sum_{i \in \mathcal{W}_t} g_{\eta_i}^t, \quad \mathcal{W}_t = (\mathcal{G}_{t+\frac{1}{2}} \cup \mathcal{B}_{t+\frac{1}{2}}) \setminus \mathcal{C}_{t+\frac{1}{2}}$
16:      **if** $\left| \left\{ i \in \mathcal{W}_t \mid \left\| \widehat{g}_{\eta^t}(\widetilde{x}^t) - g_{\eta_i}^t \right\| \leq C\sigma \right\} \right| \geq |\mathcal{W}_t|/2$ **then**
17:          $x^{t+1} \leftarrow x^t - \gamma_2 \widehat{g}_{\eta^t}(\widetilde{x}^t)$
18:      **else**
19:          **recompute**
20:      **end if**
21:      $\mathcal{C}_{t+1}, \mathcal{G}_{t+1} \cup \mathcal{B}_{t+1} = \mathsf{CheckComputations}(\mathcal{C}_{t+\frac{1}{2}}, \mathcal{G}_{t+\frac{1}{2}} \cup \mathcal{B}_{t+\frac{1}{2}})$

---

### E.3.1 Auxilary results

Similarly to Section E.1 we state the following. If a Byzantine peer deviates from the protocol at iteration $k$, it will be detected with some probability $p_k$ during the next iteration. One can lower bound this probability as

$$ p_k \geq m \cdot \frac{G_k}{n_k} \cdot \frac{1}{n_k} = \frac{m(1 - \delta_k)}{n_k} \geq \frac{m}{n}. $$

Therefore, each individual Byzantine worker can violate the protocol no more than $1/p$ times on average implying that

$$ \sum_{l=0}^{\infty} \mathbb{E}[\mathbb{1}_l] + \sum_{l=0}^{\infty} \mathbb{E}\left[\mathbb{1}_{l-\frac{1}{2}}\right] \leq \frac{nB}{m}. \tag{57} $$

**Lemma E.2.** *Let Assumption 3 holds. Let Algorithm 7 is run with $\gamma_1 \leq 1/2L$ and $\beta = \gamma_2/\gamma_1 \leq 1/2$. Then its iterations satisfy*

$$
\begin{aligned}
2\gamma_2 \langle \overline{g}_{\eta^k}(\widetilde{x}^k), \widetilde{x}^k - u \rangle \leq \ & \left\| x^k - u \right\|^2 - \left\| x^{k+1} - u \right\|^2 - 2\gamma_2 \langle \widehat{g}_{\eta^k}(\widetilde{x}^k) - \overline{g}_{\eta^k}(\widetilde{x}^k), x^k - u \rangle \\
& + 2\gamma_2^2 \left\| \widehat{g}_{\eta^k}(\widetilde{x}^k) - \overline{g}_{\eta^k}(\widetilde{x}^k) \right\|^2 + 4\gamma_1\gamma_2 \left\| \overline{g}_{\eta^k}(\widetilde{x}^k) - F(\widetilde{x}^k) \right\|^2 \\
& + 4\gamma_1\gamma_2 \left\| F(x^k) - \overline{g}_{\xi^k}(x^k) \right\|^2 + 4\gamma_1\gamma_2 \left\| \overline{g}_{\xi^k}(x^k) - \widehat{g}_{\xi^k}(x^k) \right\|^2.
\end{aligned}
$$

*Proof.* Since $x^{k+1} = x^k - \gamma_2 \widehat{g}_{\eta^k}(\widetilde{x}^k)$, we have

$$
\begin{aligned}
\left\| x^{k+1} - u \right\|^2 &= \left\| x^k - \gamma_2 \widehat{g}_{\eta^k}(\widetilde{x}^k) - u \right\|^2 \\
&= \left\| x^k - u \right\|^2 - 2\gamma_2 \langle \widehat{g}_{\eta^k}(\widetilde{x}^k), x^k - u \rangle + \gamma_2^2 \left\| \widehat{g}_{\eta^k}(\widetilde{x}^k) \right\|^2.
\end{aligned}
$$

Rearranging the terms gives that

$$2\gamma_2\langle \overline{\boldsymbol{g}}_{\boldsymbol{\eta}^k}(\widetilde{\boldsymbol{x}}^k), \boldsymbol{x}^k - \boldsymbol{u}\rangle = \left\|\boldsymbol{x}^k - \boldsymbol{u}\right\|^2 - \left\|\boldsymbol{x}^{k+1} - \boldsymbol{u}\right\|^2$$
$$-2\gamma_2\langle \widehat{\boldsymbol{g}}_{\boldsymbol{\eta}^k}(\widetilde{\boldsymbol{x}}^k) - \overline{\boldsymbol{g}}_{\boldsymbol{\eta}^k}(\widetilde{\boldsymbol{x}}^k), \boldsymbol{x}^k - \boldsymbol{u}\rangle + \gamma_2^2\left\|\widehat{\boldsymbol{g}}_{\boldsymbol{\eta}^k}(\widetilde{\boldsymbol{x}}^k)\right\|^2.$$

Next we use (14)

$$2\langle \overline{\boldsymbol{g}}_{\boldsymbol{\eta}^k}(\widetilde{\boldsymbol{x}}^k), \boldsymbol{x}^k - \boldsymbol{u}\rangle = 2\langle \overline{\boldsymbol{g}}_{\boldsymbol{\eta}^k}(\widetilde{\boldsymbol{x}}^k), \boldsymbol{x}^k - \widetilde{\boldsymbol{x}}^k\rangle + 2\langle \overline{\boldsymbol{g}}_{\boldsymbol{\eta}^k}(\widetilde{\boldsymbol{x}}^k), \widetilde{\boldsymbol{x}}^k - \boldsymbol{u}\rangle$$
$$= 2\gamma_1\langle \overline{\boldsymbol{g}}_{\boldsymbol{\eta}^k}(\widetilde{\boldsymbol{x}}^k), \widehat{\boldsymbol{g}}_{\boldsymbol{\xi}^k}(\boldsymbol{x}^k)\rangle + 2\langle \overline{\boldsymbol{g}}_{\boldsymbol{\eta}^k}(\widetilde{\boldsymbol{x}}^k), \widetilde{\boldsymbol{x}}^k - \boldsymbol{u}\rangle$$
$$\overset{(14)}{=} -\gamma_1\left(\left\|\overline{\boldsymbol{g}}_{\boldsymbol{\eta}^k}(\widetilde{\boldsymbol{x}}^k) - \widehat{\boldsymbol{g}}_{\boldsymbol{\xi}^k}(\boldsymbol{x}^k)\right\|^2 - \left\|\widehat{\boldsymbol{g}}_{\boldsymbol{\xi}^k}(\boldsymbol{x}^k)\right\|^2 - \left\|\overline{\boldsymbol{g}}_{\boldsymbol{\eta}^k}(\widetilde{\boldsymbol{x}}^k)\right\|^2\right)$$
$$+2\langle \overline{\boldsymbol{g}}_{\boldsymbol{\eta}^k}(\widetilde{\boldsymbol{x}}^k), \widetilde{\boldsymbol{x}}^k - \boldsymbol{u}\rangle$$

and obtain the following

$$2\gamma_2\langle \overline{\boldsymbol{g}}_{\boldsymbol{\eta}^k}(\widetilde{\boldsymbol{x}}^k), \widetilde{\boldsymbol{x}}^k - \boldsymbol{u}\rangle = \left\|\boldsymbol{x}^k - \boldsymbol{u}\right\|^2 - \left\|\boldsymbol{x}^{k+1} - \boldsymbol{u}\right\|^2 - 2\gamma_2\langle \widehat{\boldsymbol{g}}_{\boldsymbol{\eta}^k}(\widetilde{\boldsymbol{x}}^k) - \overline{\boldsymbol{g}}_{\boldsymbol{\eta}^k}(\widetilde{\boldsymbol{x}}^k), \boldsymbol{x}^k - \boldsymbol{u}\rangle$$
$$+\gamma_2^2\left\|\widehat{\boldsymbol{g}}_{\boldsymbol{\eta}^k}(\widetilde{\boldsymbol{x}}^k)\right\|^2$$
$$+\gamma_1\gamma_2\left(\left\|\overline{\boldsymbol{g}}_{\boldsymbol{\eta}^k}(\widetilde{\boldsymbol{x}}^k) - \widehat{\boldsymbol{g}}_{\boldsymbol{\xi}^k}(\boldsymbol{x}^k)\right\|^2 - \left\|\widehat{\boldsymbol{g}}_{\boldsymbol{\xi}^k}(\boldsymbol{x}^k)\right\|^2 - \left\|\overline{\boldsymbol{g}}_{\boldsymbol{\eta}^k}(\widetilde{\boldsymbol{x}}^k)\right\|^2\right)$$
$$\overset{(16)}{\leq} \left\|\boldsymbol{x}^k - \boldsymbol{u}\right\|^2 - \left\|\boldsymbol{x}^{k+1} - \boldsymbol{u}\right\|^2 - 2\gamma_2\langle \widehat{\boldsymbol{g}}_{\boldsymbol{\eta}^k}(\widetilde{\boldsymbol{x}}^k) - \overline{\boldsymbol{g}}_{\boldsymbol{\eta}^k}(\widetilde{\boldsymbol{x}}^k), \boldsymbol{x}^k - \boldsymbol{u}\rangle$$
$$+2\gamma_2^2\left\|\widehat{\boldsymbol{g}}_{\boldsymbol{\eta}^k}(\widetilde{\boldsymbol{x}}^k) - \overline{\boldsymbol{g}}_{\boldsymbol{\eta}^k}(\widetilde{\boldsymbol{x}}^k)\right\|^2 + 2\gamma_2^2\left\|\overline{\boldsymbol{g}}_{\boldsymbol{\eta}^k}(\widetilde{\boldsymbol{x}}^k)\right\|^2$$
$$+\gamma_1\gamma_2\left(\left\|\overline{\boldsymbol{g}}_{\boldsymbol{\eta}^k}(\widetilde{\boldsymbol{x}}^k) - \widehat{\boldsymbol{g}}_{\boldsymbol{\xi}^k}(\boldsymbol{x}^k)\right\|^2 - \left\|\widehat{\boldsymbol{g}}_{\boldsymbol{\xi}^k}(\boldsymbol{x}^k)\right\|^2 - \left\|\overline{\boldsymbol{g}}_{\boldsymbol{\eta}^k}(\widetilde{\boldsymbol{x}}^k)\right\|^2\right).$$

If $\beta = \gamma_2/\gamma_1 \leq 1/2$

$$2\gamma_2\langle \overline{\boldsymbol{g}}_{\boldsymbol{\eta}^k}(\widetilde{\boldsymbol{x}}^k), \widetilde{\boldsymbol{x}}^k - \boldsymbol{u}\rangle \leq \left\|\boldsymbol{x}^k - \boldsymbol{u}\right\|^2 - \left\|\boldsymbol{x}^{k+1} - \boldsymbol{u}\right\|^2 - 2\gamma_2\langle \widehat{\boldsymbol{g}}_{\boldsymbol{\eta}^k}(\widetilde{\boldsymbol{x}}^k) - \overline{\boldsymbol{g}}_{\boldsymbol{\eta}^k}(\widetilde{\boldsymbol{x}}^k), \boldsymbol{x}^k - \boldsymbol{u}\rangle$$
$$+2\gamma_2^2\left\|\widehat{\boldsymbol{g}}_{\boldsymbol{\eta}^k}(\widetilde{\boldsymbol{x}}^k) - \overline{\boldsymbol{g}}_{\boldsymbol{\eta}^k}(\widetilde{\boldsymbol{x}}^k)\right\|^2$$
$$+\gamma_1\gamma_2\left(\left\|\overline{\boldsymbol{g}}_{\boldsymbol{\eta}^k}(\widetilde{\boldsymbol{x}}^k) - \widehat{\boldsymbol{g}}_{\boldsymbol{\xi}^k}(\boldsymbol{x}^k)\right\|^2 - \left\|\widehat{\boldsymbol{g}}_{\boldsymbol{\xi}^k}(\boldsymbol{x}^k)\right\|^2\right).$$

Combining the latter with the result of the following chain

$$\left\|\overline{\boldsymbol{g}}_{\boldsymbol{\eta}^k}(\widetilde{\boldsymbol{x}}^k) - \widehat{\boldsymbol{g}}_{\boldsymbol{\xi}^k}(\boldsymbol{x}^k)\right\|^2 \overset{(16)}{\leq} 4\left\|\overline{\boldsymbol{g}}_{\boldsymbol{\eta}^k}(\widetilde{\boldsymbol{x}}^k) - F(\widetilde{\boldsymbol{x}}^k)\right\|^2 + 4\left\|F(\widetilde{\boldsymbol{x}}^k) - F(\boldsymbol{x}^k)\right\|^2$$
$$+4\left\|F(\boldsymbol{x}^k) - \overline{\boldsymbol{g}}_{\boldsymbol{\xi}^k}(\boldsymbol{x}^k)\right\|^2 + 4\left\|\overline{\boldsymbol{g}}_{\boldsymbol{\xi}^k}(\boldsymbol{x}^k) - \widehat{\boldsymbol{g}}_{\boldsymbol{\xi}^k}(\boldsymbol{x}^k)\right\|^2$$
$$\overset{(3)}{\leq} 4\left\|\overline{\boldsymbol{g}}_{\boldsymbol{\eta}^k}(\widetilde{\boldsymbol{x}}^k) - F(\widetilde{\boldsymbol{x}}^k)\right\|^2 + 4L^2\left\|\widetilde{\boldsymbol{x}}^k - \boldsymbol{x}^k\right\|^2$$
$$+4\left\|F(\boldsymbol{x}^k) - \overline{\boldsymbol{g}}_{\boldsymbol{\xi}^k}(\boldsymbol{x}^k)\right\|^2 + 4\left\|\overline{\boldsymbol{g}}_{\boldsymbol{\xi}^k}(\boldsymbol{x}^k) - \widehat{\boldsymbol{g}}_{\boldsymbol{\xi}^k}(\boldsymbol{x}^k)\right\|^2$$
$$= 4\left\|\overline{\boldsymbol{g}}_{\boldsymbol{\eta}^k}(\widetilde{\boldsymbol{x}}^k) - F(\widetilde{\boldsymbol{x}}^k)\right\|^2 + 4\left\|F(\boldsymbol{x}^k) - \overline{\boldsymbol{g}}_{\boldsymbol{\xi}^k}(\boldsymbol{x}^k)\right\|^2$$
$$+4\left\|\overline{\boldsymbol{g}}_{\boldsymbol{\xi}^k}(\boldsymbol{x}^k) - \widehat{\boldsymbol{g}}_{\boldsymbol{\xi}^k}(\boldsymbol{x}^k)\right\|^2 + 4\gamma_1^2 L^2\left\|\widehat{\boldsymbol{g}}_{\boldsymbol{\xi}^k}(\boldsymbol{x}^k)\right\|^2$$

we obtain if $\gamma_1 \leq 1/2L$

$$2\gamma_2\langle \overline{\boldsymbol{g}}_{\boldsymbol{\eta}^k}(\widetilde{\boldsymbol{x}}^k), \widetilde{\boldsymbol{x}}^k - \boldsymbol{u}\rangle \leq \left\|\boldsymbol{x}^k - \boldsymbol{u}\right\|^2 - \left\|\boldsymbol{x}^{k+1} - \boldsymbol{u}\right\|^2 - 2\gamma_2\langle \widehat{\boldsymbol{g}}_{\boldsymbol{\eta}^k}(\widetilde{\boldsymbol{x}}^k) - \overline{\boldsymbol{g}}_{\boldsymbol{\eta}^k}(\widetilde{\boldsymbol{x}}^k), \boldsymbol{x}^k - \boldsymbol{u}\rangle$$
$$+2\gamma_2^2\left\|\widehat{\boldsymbol{g}}_{\boldsymbol{\eta}^k}(\widetilde{\boldsymbol{x}}^k) - \overline{\boldsymbol{g}}_{\boldsymbol{\eta}^k}(\widetilde{\boldsymbol{x}}^k)\right\|^2 + 4\gamma_1\gamma_2\left\|\overline{\boldsymbol{g}}_{\boldsymbol{\eta}^k}(\widetilde{\boldsymbol{x}}^k) - F(\widetilde{\boldsymbol{x}}^k)\right\|^2$$
$$+4\gamma_1\gamma_2\left\|F(\boldsymbol{x}^k) - \overline{\boldsymbol{g}}_{\boldsymbol{\xi}^k}(\boldsymbol{x}^k)\right\|^2 + 4\gamma_1\gamma_2\left\|\overline{\boldsymbol{g}}_{\boldsymbol{\xi}^k}(\boldsymbol{x}^k) - \widehat{\boldsymbol{g}}_{\boldsymbol{\xi}^k}(\boldsymbol{x}^k)\right\|^2. \tag{58}$$

$$\square$$

### E.3.2 Lipschitz Case

**Theorem 11.** *Let Assumptions 1, 3, 5 hold. And let*

$$\gamma_1 = \min\left\{\frac{1}{2L}, \sqrt{\frac{(n-2B-m)R^2}{16\sigma^2 K}}, \sqrt{\frac{mR^2}{8\rho^2 Bn}}\right\}, \tag{59}$$

$$\gamma_2 = \min\left\{\frac{1}{4L}, \sqrt{\frac{m^2 R^2}{64\rho^2 B^2 n^2}}, \sqrt{\frac{(n-2B-m)R^2}{64\sigma^2 K}}\right\}, \tag{60}$$

*where $\rho^2 = q\sigma^2$ with $q = 2C^2 + 12 + \frac{12}{n-2B-m}$ and $C = \mathcal{O}(1)$ by Lemma E.1. Then iterations of SEG-CC (Algorithm 7) satisfy for $k \geq 1$*

$$\mathbb{E}[R_k^2] \leq 2R, \tag{61}$$

*and*

$$\sum_{k=0}^{K-1} \mathbb{E}\big[\big\langle F(\widetilde{\boldsymbol{x}}^k), \widetilde{\boldsymbol{x}}^k - \boldsymbol{x}^*\big\rangle\big] \leq \frac{R^2}{\gamma_2}.$$

*where $R_k = \|\boldsymbol{x}^k - \boldsymbol{x}^*\|$ and $R \geq \|\boldsymbol{x}^0 - \boldsymbol{x}^*\|$.*

*Proof.* Substituting $\boldsymbol{u} = \boldsymbol{x}^*$ into the result of Lemma E.2 and taking expectation over $\boldsymbol{\eta}^k$ one obtains

$$\begin{aligned}
&2\gamma_2\mathbb{E}_{\boldsymbol{\eta}^k}\big[\big\langle \overline{\boldsymbol{g}}_{\boldsymbol{\eta}^k}(\widetilde{\boldsymbol{x}}^k), \widetilde{\boldsymbol{x}}^k - \boldsymbol{x}^*\big\rangle\big] \\
&\overset{(16)}{\leq} \left\|\boldsymbol{x}^k - \boldsymbol{x}^*\right\|^2 - \mathbb{E}_{\boldsymbol{\eta}^k}\left[\left\|\boldsymbol{x}^{k+1} - \boldsymbol{x}^*\right\|^2\right] - 2\gamma_2\mathbb{E}_{\boldsymbol{\eta}^k}\big[\big\langle \widehat{\boldsymbol{g}}_{\boldsymbol{\eta}^k}(\widetilde{\boldsymbol{x}}^k) - \overline{\boldsymbol{g}}_{\boldsymbol{\eta}^k}(\widetilde{\boldsymbol{x}}^k), \boldsymbol{x}^k - \boldsymbol{x}^*\big\rangle\big] \\
&\quad + \gamma_1\gamma_2 4\mathbb{E}_{\boldsymbol{\eta}^k}\left[\left\|\overline{\boldsymbol{g}}_{\boldsymbol{\eta}^k}(\widetilde{\boldsymbol{x}}^k) - F(\widetilde{\boldsymbol{x}}^k)\right\|^2\right] + \gamma_1\gamma_2 4\left\|F(\boldsymbol{x}^k) - \overline{\boldsymbol{g}}_{\boldsymbol{\xi}^k}(\boldsymbol{x}^k)\right\|^2 \\
&\quad + \gamma_1\gamma_2 4\left\|\overline{\boldsymbol{g}}_{\boldsymbol{\xi}^k}(\boldsymbol{x}^k) - \widehat{\boldsymbol{g}}_{\boldsymbol{\xi}^k}(\boldsymbol{x}^k)\right\|^2 + 2\gamma_2^2\mathbb{E}_{\boldsymbol{\eta}^k}\left[\left\|\widehat{\boldsymbol{g}}_{\boldsymbol{\eta}^k}(\widetilde{\boldsymbol{x}}^k) - \overline{\boldsymbol{g}}_{\boldsymbol{\eta}^k}(\widetilde{\boldsymbol{x}}^k)\right\|^2\right]
\end{aligned}$$

To estimate the inner product in the right-hand side we apply Cauchy-Schwarz inequality:

$$\begin{aligned}
&-2\gamma_2\mathbb{E}_{\boldsymbol{\eta}^k}\big[\big\langle \widehat{\boldsymbol{g}}_{\boldsymbol{\eta}^k}(\widetilde{\boldsymbol{x}}^k) - \overline{\boldsymbol{g}}_{\boldsymbol{\eta}^k}(\widetilde{\boldsymbol{x}}^k), \boldsymbol{x}^k - \boldsymbol{x}^*\big\rangle\big] \\
&\qquad\leq \quad 2\gamma_2\mathbb{E}_{\boldsymbol{\eta}^k}\big[\left\|\boldsymbol{x}^k - \boldsymbol{x}^*\right\|\big\|\widehat{\boldsymbol{g}}_{\boldsymbol{\eta}^k}(\widetilde{\boldsymbol{x}}^k) - \overline{\boldsymbol{g}}_{\boldsymbol{\eta}^k}(\widetilde{\boldsymbol{x}}^k)\big\|\big] \\
&\qquad\overset{Lemma\ E.1}{\leq} \quad 2\gamma_2\rho\left\|\boldsymbol{x}^k - \boldsymbol{x}^*\right\|\mathbb{1}_k.
\end{aligned}$$

Then Assumption 6 implies

$$\begin{aligned}
2\gamma_2\big\langle F(\widetilde{\boldsymbol{x}}^k), \widetilde{\boldsymbol{x}}^k - \boldsymbol{x}^*\big\rangle \quad\leq\quad & \left\|\boldsymbol{x}^k - \boldsymbol{x}^*\right\|^2 - \mathbb{E}_{\boldsymbol{\eta}^k}\left[\left\|\boldsymbol{x}^{k+1} - \boldsymbol{x}^*\right\|^2\right] \\
& + 2\gamma_2\rho\left\|\boldsymbol{x}^k - \boldsymbol{x}^*\right\|\mathbb{1}_k + 2\gamma_2^2\rho^2\mathbb{1}_k + \frac{4\gamma_1\gamma_2\sigma^2}{|\mathcal{G}_k \setminus \mathcal{C}_k|} \\
& + 4\gamma_1\gamma_2\left\|F(\boldsymbol{x}^k) - \overline{\boldsymbol{g}}_{\boldsymbol{\xi}^k}(\boldsymbol{x}^k)\right\|^2 + 4\gamma_1\gamma_2\left\|\overline{\boldsymbol{g}}_{\boldsymbol{\xi}^k}(\boldsymbol{x}^k) - \widehat{\boldsymbol{g}}_{\boldsymbol{\xi}^k}(\boldsymbol{x}^k)\right\|^2.
\end{aligned}$$

Taking expectation over $\boldsymbol{\xi}^k$ one obtains that

$$2\gamma_2 \mathbb{E}_{\boldsymbol{\xi}^k, \boldsymbol{\eta}^k}\left[\left\langle F(\widetilde{\boldsymbol{x}}^k), \widetilde{\boldsymbol{x}}^k - \boldsymbol{x}^* \right\rangle\right]$$

$$\leq \left\|\boldsymbol{x}^k - \boldsymbol{x}^*\right\|^2 - \mathbb{E}_{\boldsymbol{\xi}^k, \boldsymbol{\eta}^k}\left[\left\|\boldsymbol{x}^{k+1} - \boldsymbol{x}^*\right\|^2\right] + 2\gamma_2\rho\|\boldsymbol{x}^k - \boldsymbol{x}^*\|\mathbb{1}_k + 2\gamma_2^2\rho^2\mathbb{1}_k$$

$$+ \gamma_1\gamma_2\left(\frac{4\sigma^2}{|\mathcal{G}_k \setminus \mathcal{C}_k|} + 4\mathbb{E}_{\boldsymbol{\xi}^k}\left[\left\|F(\boldsymbol{x}^k) - \overline{\boldsymbol{g}}_{\boldsymbol{\xi}^k}(\boldsymbol{x}^k)\right\|^2\right] + 4\mathbb{E}_{\boldsymbol{\xi}^k}\left[\left\|\overline{\boldsymbol{g}}_{\boldsymbol{\xi}^k}(\boldsymbol{x}^k) - \widehat{\boldsymbol{g}}_{\boldsymbol{\xi}^k}(\boldsymbol{x}^k)\right\|^2\right]\right)$$

$$\leq \left\|\boldsymbol{x}^k - \boldsymbol{x}^*\right\|^2 - \mathbb{E}_{\boldsymbol{\xi}^k, \boldsymbol{\eta}^k}\left[\left\|\boldsymbol{x}^{k+1} - \boldsymbol{x}^*\right\|^2\right] + 2\gamma_2\rho\|\boldsymbol{x}^k - \boldsymbol{x}^*\|\mathbb{1}_k + 2\gamma_2^2\rho^2\mathbb{1}_k$$

$$+ \gamma_1\gamma_2\left(\frac{4\sigma^2}{|\mathcal{G}_k \setminus \mathcal{C}_k|} + \frac{4\sigma^2}{\left|\mathcal{G}_{k+\frac{1}{2}} \setminus \mathcal{C}_{k+\frac{1}{2}}\right|} + 4\rho^2\mathbb{1}_{k-\frac{1}{2}}\right)$$

$$\leq \left\|\boldsymbol{x}^k - \boldsymbol{x}^*\right\|^2 - \mathbb{E}_{\boldsymbol{\xi}^k, \boldsymbol{\eta}^k}\left[\left\|\boldsymbol{x}^{k+1} - \boldsymbol{x}^*\right\|^2\right] + 2\gamma_2\rho\|\boldsymbol{x}^k - \boldsymbol{x}^*\|\mathbb{1}_k + 2\gamma_2^2\rho^2\mathbb{1}_k$$

$$+ \gamma_1\gamma_2\left(\frac{4\sigma^2}{n - 2B - m} + \frac{4\sigma^2}{n - 2B - m} + 4\rho^2\mathbb{1}_{k-\frac{1}{2}}\right).$$

Finally taking the full expectation gives that

$$2\gamma_2\mathbb{E}\left[\left\langle F(\widetilde{\boldsymbol{x}}^k), \widetilde{\boldsymbol{x}}^k - \boldsymbol{x}^*\right\rangle\right]$$

$$\leq \mathbb{E}\left[\left\|\boldsymbol{x}^k - \boldsymbol{x}^*\right\|^2\right] - \mathbb{E}\left[\left\|\boldsymbol{x}^{k+1} - \boldsymbol{x}^*\right\|^2\right] + 2\gamma_2\rho\mathbb{E}\left[\|\boldsymbol{x}^k - \boldsymbol{x}^*\|\mathbb{1}_k\right] + 2\gamma_2^2\rho^2\mathbb{E}[\mathbb{1}_k]$$

$$+ \gamma_1\gamma_2\left(\frac{8\sigma^2}{n - 2B - m} + 4\rho^2\mathbb{E}\left[\mathbb{1}_{k-\frac{1}{2}}\right]\right).$$

Summing up the results for $k = 0, 1, \ldots, K - 1$ we derive

$$\frac{2\gamma_2}{K}\sum_{k=0}^{K-1}\mathbb{E}\left[\left\langle F(\widetilde{\boldsymbol{x}}^k), \widetilde{\boldsymbol{x}}^k - \boldsymbol{x}^*\right\rangle\right]$$

$$\leq \frac{1}{K}\sum_{k=0}^{K-1}\left(\mathbb{E}\left[\|\boldsymbol{x}^k - \boldsymbol{x}^*\|^2\right] - \mathbb{E}\left[\|\boldsymbol{x}^{k+1} - \boldsymbol{x}^*\|^2\right]\right) + \frac{8\gamma_1\gamma_2\sigma^2}{n - 2B - m}$$

$$+ \frac{2\gamma_2\rho}{K}\sum_{k=0}^{K-1}\mathbb{E}\left[\|\boldsymbol{x}^k - \boldsymbol{x}^*\|\mathbb{1}_k\right] + \frac{2\gamma_2^2\rho^2}{K}\sum_{k=0}^{K-1}\mathbb{E}[\mathbb{1}_k] + \frac{4\gamma_1\gamma_2\rho^2}{K}\sum_{k=0}^{K-1}\mathbb{E}\left[\mathbb{1}_{k-\frac{1}{2}}\right]$$

$$\leq \frac{\|\boldsymbol{x}^0 - \boldsymbol{x}^*\|^2 - \mathbb{E}[\|\boldsymbol{x}^K - \boldsymbol{x}^*\|^2]}{K} + \frac{8\gamma_1\gamma_2\sigma^2}{n - 2B - m}$$

$$+ \frac{2\gamma_2\rho}{K}\sum_{k=0}^{K-1}\sqrt{\mathbb{E}\left[\|\boldsymbol{x}^k - \boldsymbol{x}^*\|^2\right]\mathbb{E}[\mathbb{1}_k]} + \frac{2\gamma_2^2\rho^2}{K}\sum_{k=0}^{K-1}\mathbb{E}[\mathbb{1}_k] + \frac{4\gamma_1\gamma_2\rho^2}{K}\sum_{k=0}^{K-1}\mathbb{E}\left[\mathbb{1}_{k-\frac{1}{2}}\right].$$

Assumption 5 implies that $0 \leq \left\langle F(\boldsymbol{x}^*), \widetilde{\boldsymbol{x}}^k - \boldsymbol{x}^*\right\rangle \leq \left\langle F(\widetilde{\boldsymbol{x}}^k), \widetilde{\boldsymbol{x}}^k - \boldsymbol{x}^*\right\rangle$. Using this and a the notation $R_k = \|\boldsymbol{x}^k - \boldsymbol{x}^*\|$, $k > 0$, $R_0 \geq \|\boldsymbol{x}^0 - \boldsymbol{x}^*\|$ we get

$$0 \leq \frac{R_0^2 - \mathbb{E}[R_K^2]}{K} + \frac{8\gamma_1\gamma_2\sigma^2}{n - 2B - m}$$

$$+ \frac{2\gamma_2\rho}{K}\sum_{k=0}^{K-1}\sqrt{\mathbb{E}\left[\|\boldsymbol{x}^k - \boldsymbol{x}^*\|^2\right]\mathbb{E}[\mathbb{1}_k]} + \frac{2\gamma_2^2\rho^2}{K}\sum_{k=0}^{K-1}\mathbb{E}[\mathbb{1}_k] + \frac{4\gamma_1\gamma_2\rho^2}{K}\sum_{k=0}^{K-1}\mathbb{E}\left[\mathbb{1}_{k-\frac{1}{2}}\right] \quad (62)$$

implying (after changing the indices) that

$$\mathbb{E}[R_k^2] \leq R_0^2 + \frac{8\gamma_1\gamma_2\sigma^2 k}{n - 2B - m} + 2\gamma_2\rho\sum_{l=0}^{k-1}\sqrt{\mathbb{E}[R_l^2]\mathbb{E}[\mathbb{1}_l]} \quad (63)$$

$$+ 2\gamma_2^2\rho^2\sum_{l=0}^{k-1}\mathbb{E}[\mathbb{1}_l] + 4\gamma_1\gamma_2\rho^2\sum_{l=0}^{k-1}\mathbb{E}\left[\mathbb{1}_{l-\frac{1}{2}}\right] \quad (64)$$

holds for all $k \geq 0$. In the remaining part of the proof we derive by induction that

$$\mathbb{E}[R_k^2] \quad \leq \quad R_0^2 + \frac{8\gamma_1\gamma_2\sigma^2 k}{n - 2B - m} + 2\gamma_2\rho \sum_{l=0}^{k-1} \sqrt{\mathbb{E}[R_l^2]\mathbb{E}[\mathbb{1}_l]} \tag{65}$$

$$+ 2\gamma_2^2\rho^2 \sum_{l=0}^{k-1} \mathbb{E}[\mathbb{1}_l] + 4\gamma_1\gamma_2\rho^2 \sum_{l=0}^{k-1} \mathbb{E}\left[\mathbb{1}_{l-\frac{1}{2}}\right] \leq 2R_0^2 \tag{66}$$

for all $k = 0, \ldots, K$. For $k = 0$ this inequality trivially holds. Next, assume that it holds for all $k = 0, 1, \ldots, T - 1, T \leq K - 1$. Let us show that it holds for $k = T$ as well. From (64) and (66) we have that $\mathbb{E}[R_k^2] \leq 2\bar{R}_0^2$ for all $k = 0, 1, \ldots, T - 1$. Therefore,

$$\mathbb{E}[R_T^2] \quad \leq \quad R_0^2 + \frac{8\gamma_1\gamma_2\sigma^2 k}{n - 2B - m} + 2\gamma_2\rho \sum_{l=0}^{k-1} \sqrt{\mathbb{E}[R_l^2]\mathbb{E}[\mathbb{1}_l]} + 2\gamma_2^2\rho^2 \sum_{l=0}^{k-1} \mathbb{E}[\mathbb{1}_l]$$

$$+ 4\gamma_1\gamma_2\rho^2 \sum_{l=0}^{k-1} \mathbb{E}\left[\mathbb{1}_{l-\frac{1}{2}}\right]$$

$$\leq \quad R_0^2 + \frac{8\gamma_1\gamma_2\sigma^2 k}{n - 2B - m} + 2\gamma_2\rho R_0 \sum_{l=0}^{k-1} \sqrt{\mathbb{E}[\mathbb{1}_l]} + 2\gamma_2^2\rho^2 \sum_{l=0}^{k-1} \mathbb{E}[\mathbb{1}_l]$$

$$+ 4\gamma_1\gamma_2\rho^2 \sum_{l=0}^{k-1} \mathbb{E}\left[\mathbb{1}_{l-\frac{1}{2}}\right].$$

The latter together with the expected number of at least one peer violations (57) implies

$$\mathbb{E}[R_T^2] \quad \leq \quad R_0^2 + \frac{8\gamma_1\gamma_2\sigma^2 k}{n - 2B - m} + 2\gamma_2\rho R_0 \frac{nB}{m} + 2\gamma_2^2\rho^2 \frac{nB}{m} + 4\gamma_1\gamma_2\rho^2 \frac{nB}{m}.$$

Taking

$$\gamma_1 = \min\left\{\frac{1}{2L}, \sqrt{\frac{(n - 2B - m)R_0^2}{16\sigma^2 K}}, \sqrt{\frac{mR_0^2}{8\rho^2 Bn}}\right\}$$

$$\gamma_2 \quad = \quad \min\left\{\sqrt{\frac{m^2 R_0^2}{64\rho^2 B^2 n^2}}, \frac{1}{4L}, \sqrt{\frac{(n - 2B - m)R_0^2}{64\sigma^2 K}}, \sqrt{\frac{mR_0^2}{32\rho^2 Bn}}\right\}$$

$$= \quad \min\left\{\frac{1}{4L}, \sqrt{\frac{m^2 R_0^2}{64\rho^2 B^2 n^2}}, \sqrt{\frac{(n - 2B - m)R_0^2}{64\sigma^2 K}}\right\}$$

we satisfy conditions of Lemma E.2 and ensure that

$$\frac{8\gamma_1\gamma_2\sigma^2 k}{n - 2B - m} + 2\gamma_2\rho R_0 \frac{nB}{m} + 2\gamma_2^2\rho^2 \frac{nB}{m} + 4\gamma_1\gamma_2\rho^2 \frac{nB}{m} \leq \frac{R_0^2}{4} + \frac{R_0^2}{4} + \frac{R_0^2}{4} + \frac{R_0^2}{4} = R_0^2,$$

and, as a result, we get

$$\mathbb{E}[R_T^2] \leq 2R_0^2 \equiv 2R. \tag{67}$$

Therefore, (66) holds for all $k = 0, 1, \ldots, K$. Together with (62) it implies

$$\sum_{k=0}^{K-1} \mathbb{E}\left[\left\langle F(\widetilde{\boldsymbol{x}}^k), \widetilde{\boldsymbol{x}}^k - \boldsymbol{x}^*\right\rangle\right] \leq \frac{R_0^2}{\gamma_2}.$$

$\square$

### E.3.3 Quasi-Strongly Monotone Case

**Lemma E.3.** *Let Assumptions 3, 4 and Corollary 2 hold. If*

$$\gamma_1 \leq \frac{1}{2L} \tag{68}$$

*then* $\overline{\boldsymbol{g}}_{\boldsymbol{\eta}^k}(\widetilde{\boldsymbol{x}}^k) = \overline{\boldsymbol{g}}_{\boldsymbol{\eta}^k}\left(\boldsymbol{x}^k - \gamma_1 \widehat{\boldsymbol{g}}_{\boldsymbol{\xi}^k}(\boldsymbol{x}^k)\right)$ *satisfies the following inequality*

$$\gamma_1^2 \mathbb{E}\left[\left\|\overline{\boldsymbol{g}}_{\boldsymbol{\eta}^k}(\widetilde{\boldsymbol{x}}^k)\right\|^2 \mid \boldsymbol{x}^k\right] \leq 2\widehat{P}_k + \frac{8\gamma_1^2\sigma^2}{G} + 4\gamma_1^2\rho^2 \mathbb{1}_{k-\frac{1}{2}}, \tag{69}$$

*where* $\widehat{P}_k = \gamma_1 \mathbb{E}_{\boldsymbol{\xi}^k, \boldsymbol{\eta}^k}\left[\langle \overline{\boldsymbol{g}}_{\boldsymbol{\eta}^k}(\widetilde{\boldsymbol{x}}^k), \boldsymbol{x}^k - \boldsymbol{x}^* \rangle\right]$ *and* $\rho^2 = q\sigma^2$ *with* $q = 2C^2 + 12 + \frac{12}{n-2B-m}$ *and* $C = \mathcal{O}(1)$ *by Lemma E.1.*

*Proof.* Using the auxiliary iterate $\widehat{\boldsymbol{x}}^{k+1} = \boldsymbol{x}^k - \gamma_1 \overline{\boldsymbol{g}}_{\boldsymbol{\eta}^k}(\widetilde{\boldsymbol{x}}^k)$, we get

$$\left\|\widehat{\boldsymbol{x}}^{k+1} - \boldsymbol{x}^*\right\|^2 = \left\|\boldsymbol{x}^k - \boldsymbol{x}^*\right\|^2 - 2\gamma_1\langle \boldsymbol{x}^k - \boldsymbol{x}^*, \overline{\boldsymbol{g}}_{\boldsymbol{\eta}^k}(\widetilde{\boldsymbol{x}}^k)\rangle + \gamma_1^2\left\|\overline{\boldsymbol{g}}_{\boldsymbol{\eta}^k}(\widetilde{\boldsymbol{x}}^k)\right\|^2 \tag{70}$$

$$= \left\|\boldsymbol{x}^k - \boldsymbol{x}^*\right\|^2 - 2\gamma_1\left\langle \boldsymbol{x}^k - \gamma\widehat{\boldsymbol{g}}_{\boldsymbol{\xi}^k}(\boldsymbol{x}^k) - \boldsymbol{x}^*, \overline{\boldsymbol{g}}_{\boldsymbol{\eta}^k}(\widetilde{\boldsymbol{x}}^k)\right\rangle \tag{71}$$

$$-2\gamma_1^2\langle \widehat{\boldsymbol{g}}_{\boldsymbol{\xi}^k}(\boldsymbol{x}^k), \overline{\boldsymbol{g}}_{\boldsymbol{\eta}^k}(\widetilde{\boldsymbol{x}}^k)\rangle + \gamma_1^2\left\|\overline{\boldsymbol{g}}_{\boldsymbol{\eta}^k}(\widetilde{\boldsymbol{x}}^k)\right\|^2. \tag{72}$$

Taking the expectation $\mathbb{E}_{\boldsymbol{\xi}^k, \boldsymbol{\eta}^k}[\cdot] = \mathbb{E}\left[\cdot \mid \boldsymbol{x}^k\right]$ conditioned on $\boldsymbol{x}^k$ from the above identity, using tower property $\mathbb{E}_{\boldsymbol{\xi}^k, \boldsymbol{\eta}^k}[\cdot] = \mathbb{E}_{\boldsymbol{\xi}^k}[\mathbb{E}_{\boldsymbol{\eta}^k}[\cdot]]$, and $\mu$-quasi strong monotonicity of $F(x)$, we derive

$$\mathbb{E}_{\boldsymbol{\xi}^k, \boldsymbol{\eta}^k}\left[\left\|\widehat{\boldsymbol{x}}^{k+1} - \boldsymbol{x}^*\right\|^2\right]$$

$$= \left\|\boldsymbol{x}^k - \boldsymbol{x}^*\right\|^2 - 2\gamma_1\mathbb{E}_{\boldsymbol{\xi}^k, \boldsymbol{\eta}^k}\left[\langle \boldsymbol{x}^k - \gamma_1\widehat{\boldsymbol{g}}_{\boldsymbol{\xi}^k}(\boldsymbol{x}^k) - \boldsymbol{x}^*, \overline{\boldsymbol{g}}_{\boldsymbol{\eta}^k}(\widetilde{\boldsymbol{x}}^k)\rangle\right]$$

$$-2\gamma_1^2\mathbb{E}_{\boldsymbol{\xi}^k, \boldsymbol{\eta}^k}\left[\langle \widehat{\boldsymbol{g}}_{\boldsymbol{\xi}^k}(\boldsymbol{x}^k), \overline{\boldsymbol{g}}_{\boldsymbol{\eta}^k}(\widetilde{\boldsymbol{x}}^k)\rangle\right] + \gamma_1^2\mathbb{E}_{\boldsymbol{\xi}^k, \boldsymbol{\eta}^k}\left[\left\|\overline{\boldsymbol{g}}_{\boldsymbol{\eta}^k}(\widetilde{\boldsymbol{x}}^k)\right\|^2\right]$$

$$= \left\|\boldsymbol{x}^k - \boldsymbol{x}^*\right\|^2$$

$$-2\gamma_1\mathbb{E}_{\boldsymbol{\xi}^k}\left[\langle \boldsymbol{x}^k - \gamma_1\widehat{\boldsymbol{g}}_{\boldsymbol{\xi}^k}(\boldsymbol{x}^k) - \boldsymbol{x}^*, F\left(\boldsymbol{x}^k - \gamma_1\widehat{\boldsymbol{g}}_{\boldsymbol{\xi}^k}(\boldsymbol{x}^k)\right)\rangle\right]$$

$$-2\gamma_1^2\mathbb{E}_{\boldsymbol{\xi}^k}\left[\langle \widehat{\boldsymbol{g}}_{\boldsymbol{\xi}^k}(\boldsymbol{x}^k), \overline{\boldsymbol{g}}_{\boldsymbol{\eta}^k}(\widetilde{\boldsymbol{x}}^k)\rangle\right] + \gamma_1^2\mathbb{E}_{\boldsymbol{\xi}^k, \boldsymbol{\eta}^k}\left[\left\|\overline{\boldsymbol{g}}_{\boldsymbol{\eta}^k}(\widetilde{\boldsymbol{x}}^k)\right\|^2\right]$$

$$\overset{\text{(QSM),(14)}}{\leq} \left\|\boldsymbol{x}^k - \boldsymbol{x}^*\right\|^2 - \gamma_1^2\mathbb{E}_{\boldsymbol{\xi}^k, \boldsymbol{\eta}^k}\left[\left\|\widehat{\boldsymbol{g}}_{\boldsymbol{\xi}^k}(\boldsymbol{x}^k)\right\|^2\right]$$

$$+\gamma_1^2\mathbb{E}_{\boldsymbol{\xi}^k, \boldsymbol{\eta}^k}\left[\left\|\widehat{\boldsymbol{g}}_{\boldsymbol{\xi}^k}(\boldsymbol{x}^k) - \overline{\boldsymbol{g}}_{\boldsymbol{\eta}^k}(\widetilde{\boldsymbol{x}}^k)\right\|^2\right].$$

To upper bound the last term we use simple inequality (16), and apply $L$-Lipschitzness of $F(x)$:

$$
\begin{aligned}
\mathbb{E}_{\boldsymbol{\xi}^k,\boldsymbol{\eta}^k}\left[\left\|\widehat{\boldsymbol{x}}^{k+1}-\boldsymbol{x}^*\right\|^2\right] \quad &\overset{(16)}{\leq}\quad \left\|\boldsymbol{x}^k-\boldsymbol{x}^*\right\|^2-\gamma_1^2\mathbb{E}_{\boldsymbol{\xi}^k}\left[\left\|\widehat{\boldsymbol{g}}_{\boldsymbol{\xi}^k}(\boldsymbol{x}^k)\right\|^2\right] \\
&\quad +4\gamma_1^2\mathbb{E}_{\boldsymbol{\xi}^k}\left[\left\|\overline{\boldsymbol{g}}_{\boldsymbol{\xi}^k}(\boldsymbol{x}^k)-\widehat{\boldsymbol{g}}_{\boldsymbol{\xi}^k}(\boldsymbol{x}^k)\right\|^2\right] \\
&\quad +4\gamma_1^2\mathbb{E}_{\boldsymbol{\xi}^k}\left[\left\|F(\boldsymbol{x}^k)-F\left(\widetilde{\boldsymbol{x}}^k\right)\right\|^2\right] \\
&\quad +4\gamma_1^2\mathbb{E}_{\boldsymbol{\xi}^k}\left[\left\|\overline{\boldsymbol{g}}_{\boldsymbol{\xi}^k}(\boldsymbol{x}^k)-F(\boldsymbol{x}^k)\right\|^2\right] \\
&\quad +4\gamma_1^2\mathbb{E}_{\boldsymbol{\xi}^k,\boldsymbol{\eta}^k}\left[\left\|\overline{\boldsymbol{g}}_{\boldsymbol{\eta}^k}\left(\widetilde{\boldsymbol{x}}^k\right)-F\left(\widetilde{\boldsymbol{x}}^k\right)\right\|^2\right] \\
&\overset{(Lip),(27),(28)}{\leq}\quad \left\|\boldsymbol{x}^k-\boldsymbol{x}^*\right\|^2-\gamma_1^2\left(1-4L^2\gamma_1^2\right)\mathbb{E}_{\boldsymbol{\xi}^k}\left[\left\|\widehat{\boldsymbol{g}}_{\boldsymbol{\xi}^k}(\boldsymbol{x}^k)\right\|^2\right] \\
&\quad +4\gamma_1^2\mathbb{E}_{\boldsymbol{\xi}^k}\left[\left\|\overline{\boldsymbol{g}}_{\boldsymbol{\xi}^k}(\boldsymbol{x}^k)-\widehat{\boldsymbol{g}}_{\boldsymbol{\xi}^k}(\boldsymbol{x}^k)\right\|^2\right] \\
&\quad +\frac{4\gamma_1^2\sigma^2}{G}+\frac{4\gamma_1^2\sigma^2}{G} \\
&\overset{(16),Lem.\ D.1}{\leq}\quad \left\|\boldsymbol{x}^k-\boldsymbol{x}^*\right\|^2-\gamma_1^2\left(1-4\gamma_1^2L^2\right)\mathbb{E}_{\boldsymbol{\xi}^k}\left[\left\|\widehat{\boldsymbol{g}}_{\boldsymbol{\xi}^k}(\boldsymbol{x}^k)\right\|^2\right] \\
&\quad +\frac{8\gamma_1^2\sigma^2}{G}+4\gamma_1^2\rho^2\mathbb{1}_{k-\frac{1}{2}} \\
&\overset{(68)}{\leq}\quad \left\|\boldsymbol{x}^k-\boldsymbol{x}^*\right\|^2+\frac{8\gamma_1^2\sigma^2}{G}+4\gamma_1^2\rho^2\mathbb{1}_{k-\frac{1}{2}}.
\end{aligned}
$$

Finally, we use the above inequality together with (70):

$$
\left\|\boldsymbol{x}^k-\boldsymbol{x}^*\right\|^2-2\widehat{P}_k+\gamma_1^2\mathbb{E}\left[\left\|\overline{\boldsymbol{g}}_{\boldsymbol{\eta}^k}(\widetilde{\boldsymbol{x}}^k)\right\|^2\mid\boldsymbol{x}^k\right]\quad\leq\quad\left\|\boldsymbol{x}^k-\boldsymbol{x}^*\right\|^2+\frac{8\gamma_1^2\sigma^2}{G}+4\gamma_1^2\rho^2\mathbb{1}_{k-\frac{1}{2}},
$$

where $\widehat{P}_k=\gamma_1\mathbb{E}_{\boldsymbol{\xi}^k,\boldsymbol{\eta}^k}\left[\left\langle\overline{\boldsymbol{g}}_{\boldsymbol{\eta}^k}(\widetilde{\boldsymbol{x}}^k),\boldsymbol{x}^k-\boldsymbol{x}^*\right\rangle\right]$. Rearranging the terms, we obtain (69). $\qquad\square$

**Lemma E.4.** *Let Assumptions 3, 4 and Corollary 2 hold. If*

$$
\gamma_1\leq\frac{1}{2\mu+2L}, \tag{73}
$$

*then $\overline{\boldsymbol{g}}_{\boldsymbol{\eta}^k}(\widetilde{\boldsymbol{x}}^k)=\overline{\boldsymbol{g}}_{\boldsymbol{\eta}^k}\left(\boldsymbol{x}^k-\gamma_1\widehat{\boldsymbol{g}}_{\boldsymbol{\xi}^k}(\boldsymbol{x}^k)\right)$ satisfies the following inequality*

$$
\widehat{P}_k\quad\geq\quad\frac{\mu\gamma_1}{2}\left\|\boldsymbol{x}^k-\boldsymbol{x}^*\right\|^2+\frac{\gamma_1^2}{4}\mathbb{E}_{\boldsymbol{\xi}^k}\left[\left\|\overline{\boldsymbol{g}}_{\boldsymbol{\xi}^k}(\boldsymbol{x}^k)\right\|^2\right]-\frac{8\gamma_1^2\sigma^2}{G}-\frac{9\gamma_1^2\rho^2\mathbb{1}_{k-\frac{1}{2}}}{2}, \tag{74}
$$

*or simply*

$$
-\widehat{P}_k\quad\leq\quad-\frac{\mu\gamma_1}{2}\left\|\boldsymbol{x}^k-\boldsymbol{x}^*\right\|^2+\frac{4\gamma_1^2\sigma^2}{G}+4\gamma_1^2\rho^2\mathbb{1}_{k-\frac{1}{2}}
$$

*where $\widehat{P}_k=\gamma_1\mathbb{E}_{\boldsymbol{\xi}^k,\boldsymbol{\eta}^k}\left[\left\langle\overline{\boldsymbol{g}}_{\boldsymbol{\eta}^k}(\widetilde{\boldsymbol{x}}^k),\boldsymbol{x}^k-\boldsymbol{x}^*\right\rangle\right]$, where $\rho^2=q\sigma^2$ with $q=2C^2+12+\frac{12}{n-2B-m}$ and $C=\mathcal{O}(1)$ by Lemma E.1.*

*Proof.* Since $\mathbb{E}_{\boldsymbol{\xi}^k,\boldsymbol{\eta}^k}[\cdot] = \mathbb{E}[\cdot \mid \boldsymbol{x}^k]$ and $\overline{\boldsymbol{g}}_{\boldsymbol{\eta}^k}(\widetilde{\boldsymbol{x}}^k) = \overline{\boldsymbol{g}}_{\boldsymbol{\eta}^k}\left(\boldsymbol{x}^k - \gamma_1 \widehat{\boldsymbol{g}}_{\boldsymbol{\xi}^k}(\boldsymbol{x}^k)\right)$, we have

$$
\begin{aligned}
-\widehat{P}_k \quad &= \quad -\gamma_1 \mathbb{E}_{\boldsymbol{\xi}^k,\boldsymbol{\eta}^k}\left[\langle \overline{\boldsymbol{g}}_{\boldsymbol{\eta}^k}(\widetilde{\boldsymbol{x}}^k), \boldsymbol{x}^k - \boldsymbol{x}^*\rangle\right] \\
&= \quad -\gamma_1 \mathbb{E}_{\boldsymbol{\xi}^k}\left[\langle \mathbb{E}_{\boldsymbol{\eta}^k}[\overline{\boldsymbol{g}}_{\boldsymbol{\eta}^k}(\widetilde{\boldsymbol{x}}^k)], \boldsymbol{x}^k - \gamma_1 \widehat{\boldsymbol{g}}_{\boldsymbol{\xi}^k}(\boldsymbol{x}^k) - \boldsymbol{x}^*\rangle\right] \\
&\qquad -\gamma_1^2 \mathbb{E}\left[\langle \overline{\boldsymbol{g}}_{\boldsymbol{\eta}^k}(\widetilde{\boldsymbol{x}}^k), \widehat{\boldsymbol{g}}_{\boldsymbol{\xi}^k}(\boldsymbol{x}^k)\rangle\right] \\
&\overset{(14)}{=} \quad -\gamma_1 \mathbb{E}_{\boldsymbol{\xi}^k}\left[\langle F(\boldsymbol{x}^k - \gamma_1 \widehat{\boldsymbol{g}}_{\boldsymbol{\xi}^k}(\boldsymbol{x}^k)), \boldsymbol{x}^k - \gamma_1 \widehat{\boldsymbol{g}}_{\boldsymbol{\xi}^k}(\boldsymbol{x}^k) - \boldsymbol{x}^*\rangle\right] \\
&\qquad -\frac{\gamma_1^2}{2}\mathbb{E}_{\boldsymbol{\xi}^k,\boldsymbol{\eta}^k}\left[\left\|\overline{\boldsymbol{g}}_{\boldsymbol{\eta}^k}(\widetilde{\boldsymbol{x}}^k)\right\|^2\right] - \frac{\gamma_1^2}{2}\mathbb{E}_{\boldsymbol{\xi}^k}\left[\left\|\widehat{\boldsymbol{g}}_{\boldsymbol{\xi}^k}(\boldsymbol{x}^k)\right\|^2\right] \\
&\qquad +\frac{\gamma_1^2}{2}\mathbb{E}_{\boldsymbol{\xi}^k,\boldsymbol{\eta}^k}\left[\left\|\overline{\boldsymbol{g}}_{\boldsymbol{\eta}^k}(\widetilde{\boldsymbol{x}}^k) - \widehat{\boldsymbol{g}}_{\boldsymbol{\xi}^k}(\boldsymbol{x}^k)\right\|^2\right] \\
&\overset{(\mathrm{QSM}),(16)}{\leq} \quad -\mu\gamma_1 \mathbb{E}_{\boldsymbol{\xi}^k,\boldsymbol{\eta}^k}\left[\left\|\boldsymbol{x}^k - \boldsymbol{x}^* - \gamma_1 \widehat{\boldsymbol{g}}_{\boldsymbol{\xi}^k}(\boldsymbol{x}^k)\right\|^2\right] - \frac{\gamma_1^2}{2}\mathbb{E}_{\boldsymbol{\xi}^k}\left[\left\|\widehat{\boldsymbol{g}}_{\boldsymbol{\xi}^k}(\boldsymbol{x}^k)\right\|^2\right] \\
&\qquad +\frac{4\gamma_1^2}{2}\mathbb{E}_{\boldsymbol{\xi}^k}\left[\left\|\overline{\boldsymbol{g}}_{\boldsymbol{\xi}^k}(\boldsymbol{x}^k) - \widehat{\boldsymbol{g}}_{\boldsymbol{\xi}^k}(\boldsymbol{x}^k)\right\|^2\right] \\
&\qquad +\frac{4\gamma_1^2}{2}\mathbb{E}_{\boldsymbol{\xi}^k}\left[\left\|F(\boldsymbol{x}^k) - F(\widetilde{\boldsymbol{x}}^k)\right\|^2\right] \\
&\qquad +\frac{4\gamma_1^2}{2}\mathbb{E}_{\boldsymbol{\xi}^k}\left[\left\|\overline{\boldsymbol{g}}_{\boldsymbol{\xi}^k}(\boldsymbol{x}^k) - F(\boldsymbol{x}^k)\right\|^2\right] \\
&\qquad +\frac{4\gamma_1^2}{2}\mathbb{E}_{\boldsymbol{\xi}^k,\boldsymbol{\eta}^k}\left[\left\|\overline{\boldsymbol{g}}_{\boldsymbol{\eta}^k}(\widetilde{\boldsymbol{x}}^k) - F(\widetilde{\boldsymbol{x}}^k)\right\|^2\right] \\
&\overset{(17),(\mathrm{Lip}),Lem.D.1,Cor.2}{\leq} \quad -\frac{\mu\gamma_1}{2}\left\|\boldsymbol{x}^k - \boldsymbol{x}^*\right\|^2 - \frac{\gamma_1^2}{2}(1 - 2\gamma_1\mu - 4\gamma_1^2 L^2)\mathbb{E}_{\boldsymbol{\xi}^k}\left[\left\|\widehat{\boldsymbol{g}}_{\boldsymbol{\xi}^k}(\boldsymbol{x}^k)\right\|^2\right] \\
&\qquad +\frac{4\gamma_1^2\sigma^2}{2g} + \frac{4\gamma_1^2\sigma^2}{2g} + 4\gamma_1^2\rho^2 \mathbb{1}_{k-\frac{1}{2}} \\
&\overset{(73)}{\leq} \quad -\frac{\mu\gamma_1}{2}\left\|\boldsymbol{x}^k - \boldsymbol{x}^*\right\|^2 - \frac{\gamma_1^2}{2}\mathbb{E}_{\boldsymbol{\xi}^k}\left[\left\|\widehat{\boldsymbol{g}}_{\boldsymbol{\xi}^k}(\boldsymbol{x}^k)\right\|^2\right] \\
&\qquad +\frac{4\gamma_1^2\sigma^2}{G} + 4\gamma_1^2\rho^2 \mathbb{1}_{k-\frac{1}{2}}
\end{aligned}
$$

So one have

$$
-\widehat{P}_k \quad \leq \quad -\frac{\mu\gamma_1}{2}\left\|\boldsymbol{x}^k - \boldsymbol{x}^*\right\|^2 - \frac{\gamma_1^2}{4}\mathbb{E}_{\boldsymbol{\xi}^k}\left[\left\|\overline{\boldsymbol{g}}_{\boldsymbol{\xi}^k}(\boldsymbol{x}^k)\right\|^2\right] + \frac{4\gamma_1^2\sigma^2}{G} + \frac{9\gamma_1^2\rho^2 \mathbb{1}_{k-\frac{1}{2}}}{2}
$$

or simply

$$
-\widehat{P}_k \quad \leq \quad -\frac{\mu\gamma_1}{2}\left\|\boldsymbol{x}^k - \boldsymbol{x}^*\right\|^2 + \frac{4\gamma_1^2\sigma^2}{G} + 4\gamma_1^2\rho^2 \mathbb{1}_{k-\frac{1}{2}}
$$

that concludes the proof. $\qquad\square$

Combining Lemmas E.3 and E.4, we get the following result.

**Theorem** (Theorem 6 duplicate)**.** *Let Assumptions 1, 3 and 4 hold. Then after $T$ iterations* SEG-CC *(Algorithm 7) with $\gamma_1 \leq \frac{1}{2\mu+2L}$ and $\beta = \gamma_2/\gamma_1 \leq 1/4$ outputs $\boldsymbol{x}^T$ such that*

$$
\mathbb{E}\left\|\boldsymbol{x}^T - \boldsymbol{x}^*\right\|^2 \leq \left(1 - \frac{\mu\beta\gamma_1}{4}\right)^T \left\|\boldsymbol{x}^0 - \boldsymbol{x}^*\right\|^2 + 2\sigma^2\left(\frac{4\gamma_1}{\beta\mu^2(n-2B-m)} + \frac{\gamma_1 qnB}{m}\right),
$$

*where $q = 2C^2 + 12 + \frac{12}{n-2B-m}$; $q = \mathcal{O}(1)$ since $C = \mathcal{O}(1)$.*

*Proof of Theorem 6.* Since $x^{k+1} = x^k - \gamma_2\widehat{g}_{\eta^k}(\widetilde{x}^k)$, we have

$$
\begin{aligned}
\left\|x^{k+1} - x^*\right\|^2 &= \left\|x^k - \gamma_2\widehat{g}_{\eta^k}(\widetilde{x}^k) - x^*\right\|^2 \\
&= \left\|x^k - x^*\right\|^2 - 2\gamma_2\langle\widehat{g}_{\eta^k}(\widetilde{x}^k), x^k - x^*\rangle + \gamma_2^2\left\|\widehat{g}_{\eta^k}(\widetilde{x}^k)\right\|^2 \\
&\leq \left\|x^k - x^*\right\|^2 - 2\gamma_2\langle\overline{g}_{\eta^k}(\widetilde{x}^k), x^k - x^*\rangle + 2\gamma_2^2\left\|\overline{g}_{\eta^k}(\widetilde{x}^k)\right\|^2 \\
&\quad + 2\gamma_2^2\left\|\overline{g}_{\eta^k}(\widetilde{x}^k) - \widehat{g}_{\eta^k}(\widetilde{x}^k)\right\|^2 + 2\gamma_2\langle\overline{g}_{\eta^k}(\widetilde{x}^k) - \widehat{g}_{\eta^k}(\widetilde{x}^k), x^k - x^*\rangle \\
&\leq (1+\lambda)\left\|x^k - x^*\right\|^2 - 2\gamma_2\langle\overline{g}_{\eta^k}(\widetilde{x}^k), x^k - x^*\rangle + 2\gamma_2^2\left\|\overline{g}_{\eta^k}(\widetilde{x}^k)\right\|^2 \\
&\quad + \gamma_2^2\left(2 + \frac{1}{\lambda}\right)\left\|\overline{g}_{\eta^k}(\widetilde{x}^k) - \widehat{g}_{\eta^k}(\widetilde{x}^k)\right\|^2
\end{aligned}
$$

Taking the expectation, conditioned on $x^k$,

$$
\begin{aligned}
\mathbb{E}_{\xi^k,\eta^k}\left\|x^{k+1} - x^*\right\|^2 &\leq (1+\lambda)\left\|x^k - x^*\right\|^2 - 2\beta\gamma_1\mathbb{E}_{\xi^k,\eta^k}\langle\overline{g}_{\eta^k}(\widetilde{x}^k), x^k - x^*\rangle \\
&\quad + 2\beta^2\gamma_1^2\mathbb{E}_{\xi^k,\eta^k}\left\|\overline{g}_{\eta^k}(\widetilde{x}^k)\right\|^2 + \gamma_2^2\rho^2\left(2 + \frac{1}{\lambda}\right)\mathbb{1}_k,
\end{aligned}
$$

using the definition of $\widehat{P}_k = \gamma_1\mathbb{E}_{\xi^k,\eta^k}\left[\langle\overline{g}_{\eta^k}(\widetilde{x}^k), x^k - x^*\rangle\right]$, we continue our derivation:

$$
\begin{aligned}
&\mathbb{E}_{\xi^k,\eta^k}\left[\left\|x^{k+1} - x^*\right\|^2\right] \\
&= (1+\lambda)\left\|x^k - x^*\right\|^2 - 2\beta\widehat{P}_k + 2\beta^2\gamma_1^2\mathbb{E}_{\xi^k,\eta^k}\left\|\overline{g}_{\eta^k}(\widetilde{x}^k)\right\|^2 \\
&\quad + \gamma_2^2\rho^2\left(2 + \frac{1}{\lambda}\right)\mathbb{1}_k \\
&\overset{(69)}{\leq} (1+\lambda)\left\|x^k - x^*\right\|^2 - 2\beta\widehat{P}_k + 2\beta^2\left(2\widehat{P}_k + \frac{8\gamma_1^2\sigma^2}{G} + 4\gamma_1^2\rho^2\mathbb{1}_{k-\frac{1}{2}}\right) \\
&\quad + \gamma_2^2\rho^2\left(2 + \frac{1}{\lambda}\right)\mathbb{1}_k \\
&\overset{0\leq\beta\leq 1/2}{\leq} (1+\lambda)\left\|x^k - x^*\right\|^2 - 2\widehat{P}_k(\beta - 2\beta^2) + \frac{16\gamma_2^2\sigma^2}{G} + 8\gamma_2^2\rho^2\mathbb{1}_{k-\frac{1}{2}} \\
&\quad + \gamma_2^2\rho^2\left(2 + \frac{1}{\lambda}\right)\mathbb{1}_k \\
&\overset{(74)}{\leq} (1+\lambda)\left\|x^k - x^*\right\|^2 \\
&\quad + 2\beta(1-2\beta)\left(-\frac{\mu\gamma_1}{2}\left\|x^k - x^*\right\|^2 + \frac{4\gamma_1^2\sigma^2}{G} + 4\gamma_1^2\rho^2\mathbb{1}_{k-\frac{1}{2}}\right) \\
&\quad + \frac{16\gamma_2^2\sigma^2}{G} + 8\gamma_2^2\rho^2\mathbb{1}_{k-\frac{1}{2}} + \gamma_2^2\rho^2\left(2 + \frac{1}{\lambda}\right)\mathbb{1}_k \\
&\leq \left(1 + \lambda - 2\beta(1-2\beta)\frac{\mu\gamma_1}{2}\right)\left\|x^k - x^*\right\|^2 \\
&\quad + \frac{\gamma_1^2\sigma^2}{G} + \gamma_1^2\rho^2\mathbb{1}_{k-\frac{1}{2}} + \frac{16\gamma_2^2\sigma^2}{G} + 8\gamma_2^2\rho^2\mathbb{1}_{k-\frac{1}{2}} + \gamma_2^2\rho^2\left(2 + \frac{1}{\lambda}\right)\mathbb{1}_k \\
&\overset{0\leq\beta\leq 1/4}{\leq} \left(1 + \lambda - \frac{\mu\gamma_2}{2}\right)\left\|x^k - x^*\right\|^2 \\
&\quad + \frac{\sigma^2}{G}(\gamma_1^2 + 16\gamma_2^2) + \gamma_1^2\rho^2\mathbb{1}_{k-\frac{1}{2}} + 8\gamma_2^2\rho^2\mathbb{1}_{k-\frac{1}{2}} + \gamma_2^2\rho^2\left(2 + \frac{1}{\lambda}\right)\mathbb{1}_k \\
&\overset{\lambda=\mu\gamma_2/4}{\leq} \left(1 - \frac{\mu\gamma_2}{4}\right)\left\|x^k - x^*\right\|^2 \\
&\quad + \frac{\sigma^2}{G}(\gamma_1^2 + 16\gamma_2^2) + \gamma_1^2\rho^2\mathbb{1}_{k-\frac{1}{2}} + 8\gamma_2^2\rho^2\mathbb{1}_{k-\frac{1}{2}} + \gamma_2^2\rho^2\left(2 + \frac{4}{\mu\gamma_2}\right)\mathbb{1}_k.
\end{aligned}
$$

Next, taking the full expectation from the both sides and obtain

$$\mathbb{E}\left[\left\|x^{k+1}-x^*\right\|^2\right] \leq \left(1-\frac{\mu\gamma_2}{4}\right)\mathbb{E}\left[\left\|x^k-x^*\right\|^2\right]$$
$$+\frac{\sigma^2}{G}\left(\gamma_1^2+16\gamma_2^2\right)+\rho^2\left(\gamma_1^2+8\gamma_2^2\right)\mathbb{E}\mathbb{1}_{k-\frac{1}{2}}+2\rho^2\left(\gamma_2^2+\frac{2\gamma_2}{\mu}\right)\mathbb{E}\mathbb{1}_k.$$

Unrolling the recurrence, we derive the rest of the result:

$$\mathbb{E}\left\|x^{K+1}-x^*\right\|^2 \leq \left(1-\frac{\mu\gamma_2}{4}\right)^{K+1}\left\|x^0-x^*\right\|^2+\frac{4\sigma^2\left(\gamma_1^2+16\gamma_2^2\right)}{\gamma_2\mu(n-2B-m)}$$
$$+\rho^2\left(\gamma_1^2+8\gamma_2^2\right)\sum_i^K\left(1-\frac{\mu\gamma_2}{4}\right)^{K-i}\mathbb{E}\mathbb{1}_{i-\frac{1}{2}}$$
$$+2\rho^2\left(\gamma_2^2+\frac{2\gamma_2}{\mu}\right)\sum_i^K\left(1-\frac{\mu\gamma_2}{4}\right)^{K-i}\mathbb{E}\mathbb{1}_i.$$

Since $\gamma_2 \leq \frac{4}{\mu}$ and that implies

$$\mathbb{E}\left[\sum_i^T\mathbb{1}_i\left(1-\frac{\gamma\mu}{2}\right)^{T-i}\right] \leq \mathbb{E}\left[\sum_i^T\mathbb{1}_i\right] \leq \frac{nB}{m}. \tag{75}$$

using the expected number of at least one peer violations (57) we derive

$$\mathbb{E}\left\|x^{K+1}-x^*\right\|^2 \leq \left(1-\frac{\mu\gamma_2}{4}\right)^{K+1}\left\|x^0-x^*\right\|^2+\frac{4\sigma^2\left(\gamma_1^2+16\gamma_2^2\right)}{\gamma_2\mu^2(n-2B-m)}$$
$$+\rho^2\left(\gamma_1^2+8\gamma_2^2\right)\frac{nB}{m}+2\rho^2\left(\gamma_2^2+\frac{2\gamma_2}{\mu}\right)\frac{nB}{m}$$
$$\leq \left(1-\frac{\mu\gamma_2}{4}\right)^{K+1}\left\|x^0-x^*\right\|^2+\frac{4\sigma^2\left(\gamma_1^2+16\gamma_2^2\right)}{\gamma_2\mu(n-2B-m)}$$
$$+\rho^2\left(\gamma_1^2+10\gamma_2^2\right)\frac{nB}{m}+\frac{4\rho^2\gamma_2}{\mu}\frac{nB}{m},$$

that together with $\rho^2 = q\sigma^2$ with $q = 2C^2+12+\frac{12}{n-2B-m}$ and $C = \mathcal{O}(1)$ by Lemma E.1 give result of the theorem. $\qquad\square$

**Corollary 9.** *Let assumptions of Theorem 6 hold. Then $\mathbb{E}\left\|x^T-x^*\right\|^2 \leq \varepsilon$ holds after*

$$T = \widetilde{\mathcal{O}}\left(\frac{L}{\mu}+\frac{1}{\beta}+\frac{\sigma^2}{\beta^2\mu^2(n-2B-m)\varepsilon}+\frac{q\sigma^2Bn}{\beta\mu^2m\varepsilon}+\frac{q\sigma^2Bn}{\beta^2\mu^2m\sqrt{\varepsilon}}\right)$$

*iterations of* SEG-CC *with*

$$\gamma = \min\left\{\frac{1}{2L+2\mu},\frac{4\ln\left(\max\left\{2,\min\left\{\frac{m(n-2B-m)\beta^2\mu^2R^2K}{32m\sigma^2+4q\sigma^2\beta^2nB(n-2B-m)},\frac{m\mu^2\beta^2R^2K^2}{32qnB\sigma^2}\right\}\right\}\right)}{\mu\beta(K+1)}\right\}.$$

*Proof.* Next, we plug $\gamma_2 = \beta\gamma_1 \leq \gamma_1/4$ into the result of Theorem 6 and obtain

$$\mathbb{E}\left[\left\|x^{k+1}-x^*\right\|^2\right] \leq \left(1-\frac{\mu\beta\gamma_1}{4}\right)\left\|x^0-x^*\right\|^2+\frac{8\sigma^2\gamma_1}{\beta\mu(n-2B-m)}+2\rho^2\gamma_1^2\frac{nB}{m}+\frac{4\rho^2\beta\gamma_1}{\mu}\frac{nB}{m}. \tag{76}$$

Using the definition of $\rho$ ($\rho^2 = q\sigma^2 = \mathcal{O}(\sigma^2)$) from Lemma E.1 and if $B \leq \frac{n}{4}$, $m << n$ the result of Theorem 6 can be simplified as

$$\mathbb{E}\left\|x^T-x^*\right\|^2 \leq \left(1-\frac{\mu\beta\gamma_1}{4}\right)^T\left\|x^0-x^*\right\|^2+\frac{8\sigma^2\gamma_1}{\beta\mu(n-2B-m)}+2q\sigma^2\gamma_1^2\frac{nB}{m}+\frac{4q\sigma^2\beta\gamma_1}{\mu}\frac{nB}{m}.$$

Applying Lemma C.4 to the last bound we get the result of the corollary. $\qquad\square$

### E.3.4 Lipschitz Monotone Case

**Theorem 12.** *Suppose the assumptions of Theorem 11 and Assumption 5 hold. Then after $K$ iterations of* SEG-CC *(Algorithm 7)*

$$\mathbb{E}\Big[\textit{Gap}_{B_R(x^*)}\big(\overline{\boldsymbol{x}}^K\big)\Big] \leq \frac{3R^2}{2\gamma_2 K}, \tag{77}$$

*where* $\textit{Gap}_{B_R(x^*)}\big(\overline{\boldsymbol{x}}^K\big) = \max\limits_{u \in B_R(x^*)} \big\langle F(\boldsymbol{u}), \overline{\boldsymbol{x}}^K - \boldsymbol{u}\big\rangle$, $\overline{\boldsymbol{x}}^K = \frac{1}{K}\sum\limits_{k=0}^{K-1}\widetilde{\boldsymbol{x}}^k$ *and* $R \leq \|\boldsymbol{x}^0 - \boldsymbol{x}^*\|$.

*Proof.* We start the proof with the result of Lemma E.2

$$
\begin{aligned}
2\gamma_2\big\langle \overline{\boldsymbol{g}}_{\boldsymbol{\eta}^k}(\widetilde{\boldsymbol{x}}^k), \widetilde{\boldsymbol{x}}^k - \boldsymbol{u}\big\rangle \;\leq\;& \big\|\boldsymbol{x}^k - \boldsymbol{u}\big\|^2 - \big\|\boldsymbol{x}^{k+1} - \boldsymbol{u}\big\|^2 - 2\gamma_2\big\langle \widehat{\boldsymbol{g}}_{\boldsymbol{\eta}^k}(\widetilde{\boldsymbol{x}}^k) - \overline{\boldsymbol{g}}_{\boldsymbol{\eta}^k}(\widetilde{\boldsymbol{x}}^k), \boldsymbol{x}^k - \boldsymbol{u}\big\rangle \\
&+ 2\gamma_2^2\big\|\widehat{\boldsymbol{g}}_{\boldsymbol{\eta}^k}(\widetilde{\boldsymbol{x}}^k) - \overline{\boldsymbol{g}}_{\boldsymbol{\eta}^k}(\widetilde{\boldsymbol{x}}^k)\big\|^2 + 4\gamma_1\gamma_2\big\|\overline{\boldsymbol{g}}_{\boldsymbol{\eta}^k}(\widetilde{\boldsymbol{x}}^k) - F(\widetilde{\boldsymbol{x}}^k)\big\|^2 \\
&+ 4\gamma_1\gamma_2\big\|F(\boldsymbol{x}^k) - \overline{\boldsymbol{g}}_{\boldsymbol{\xi}^k}(\boldsymbol{x}^k)\big\|^2 + 4\gamma_1\gamma_2\big\|\overline{\boldsymbol{g}}_{\boldsymbol{\xi}^k}(\boldsymbol{x}^k) - \widehat{\boldsymbol{g}}_{\boldsymbol{\xi}^k}(\boldsymbol{x}^k)\big\|^2,
\end{aligned}
$$

that leads to the following inequality

$$
\begin{aligned}
&2\gamma_2\big\langle F(\widetilde{\boldsymbol{x}}^k), \widetilde{\boldsymbol{x}}^k - \boldsymbol{u}\big\rangle \\
&\leq\; \big\|\boldsymbol{x}^k - \boldsymbol{u}\big\|^2 - \big\|\boldsymbol{x}^{k+1} - \boldsymbol{u}\big\|^2 + 2\gamma_2^2\big\|\widehat{\boldsymbol{g}}_{\boldsymbol{\eta}^k}(\widetilde{\boldsymbol{x}}^k) - \overline{\boldsymbol{g}}_{\boldsymbol{\eta}^k}(\widetilde{\boldsymbol{x}}^k)\big\|^2 \\
&\quad + 2\gamma_2\big\langle F(\widetilde{\boldsymbol{x}}^k) - \overline{\boldsymbol{g}}_{\boldsymbol{\eta}^k}(\widetilde{\boldsymbol{x}}^k), \widetilde{\boldsymbol{x}}^k - \boldsymbol{u}\big\rangle - 2\gamma_2\big\langle \widehat{\boldsymbol{g}}_{\boldsymbol{\eta}^k}(\widetilde{\boldsymbol{x}}^k) - \overline{\boldsymbol{g}}_{\boldsymbol{\eta}^k}(\widetilde{\boldsymbol{x}}^k), \boldsymbol{x}^k - \boldsymbol{u}\big\rangle \\
&\quad + 4\gamma_1\gamma_2\Big(\big\|\overline{\boldsymbol{g}}_{\boldsymbol{\eta}^k}(\widetilde{\boldsymbol{x}}^k) - F(\widetilde{\boldsymbol{x}}^k)\big\|^2 + \big\|F(\boldsymbol{x}^k) - \overline{\boldsymbol{g}}_{\boldsymbol{\xi}^k}(\boldsymbol{x}^k)\big\|^2 + \big\|\overline{\boldsymbol{g}}_{\boldsymbol{\xi}^k}(\boldsymbol{x}^k) - \widehat{\boldsymbol{g}}_{\boldsymbol{\xi}^k}(\boldsymbol{x}^k)\big\|^2\Big).
\end{aligned}
$$

Assumption 5 implies that

$$\big\langle F(\boldsymbol{u}), \widetilde{\boldsymbol{x}}^k - \boldsymbol{u}\big\rangle \leq \big\langle F(\widetilde{\boldsymbol{x}}^k), \widetilde{\boldsymbol{x}}^k - \boldsymbol{u}\big\rangle \tag{78}$$

and consequently by Jensen inequality

$$
\begin{aligned}
&2\gamma_2 K\big\langle F(\boldsymbol{u}), \overline{\boldsymbol{x}}^K - \boldsymbol{u}\big\rangle \\
&\leq\; \big\|\boldsymbol{x}^0 - \boldsymbol{u}\big\|^2 + 2\gamma_2^2\sum_{k=0}^{K-1}\big\|\widehat{\boldsymbol{g}}_{\boldsymbol{\eta}^k}(\widetilde{\boldsymbol{x}}^k) - \overline{\boldsymbol{g}}_{\boldsymbol{\eta}^k}(\widetilde{\boldsymbol{x}}^k)\big\|^2 \\
&\quad + 2\gamma_2\sum_{k=0}^{K-1}\Big(\big\langle F(\widetilde{\boldsymbol{x}}^k) - \overline{\boldsymbol{g}}_{\boldsymbol{\eta}^k}(\widetilde{\boldsymbol{x}}^k), \widetilde{\boldsymbol{x}}^k - \boldsymbol{u}\big\rangle - \big\langle \widehat{\boldsymbol{g}}_{\boldsymbol{\eta}^k}(\widetilde{\boldsymbol{x}}^k) - \overline{\boldsymbol{g}}_{\boldsymbol{\eta}^k}(\widetilde{\boldsymbol{x}}^k), \boldsymbol{x}^k - \boldsymbol{u}\big\rangle\Big) \\
&\quad + 4\gamma_1\gamma_2\sum_{k=0}^{K-1}\Big(\big\|\overline{\boldsymbol{g}}_{\boldsymbol{\eta}^k}(\widetilde{\boldsymbol{x}}^k) - F(\widetilde{\boldsymbol{x}}^k)\big\|^2 + \big\|F(\boldsymbol{x}^k) - \overline{\boldsymbol{g}}_{\boldsymbol{\xi}^k}(\boldsymbol{x}^k)\big\|^2 + \big\|\overline{\boldsymbol{g}}_{\boldsymbol{\xi}^k}(\boldsymbol{x}^k) - \widehat{\boldsymbol{g}}_{\boldsymbol{\xi}^k}(\boldsymbol{x}^k)\big\|^2\Big),
\end{aligned}
$$

where $\overline{\boldsymbol{x}}^K = \frac{1}{K}\sum\limits_{k=0}^{K-1}\widetilde{\boldsymbol{x}}^k$.

Then maximization in $\boldsymbol{u}$ gives

$$
\begin{aligned}
&2\gamma_2 K\,\mathtt{Gap}_{B_R(x^*)}\big(\overline{\boldsymbol{x}}^K\big) \\
&\leq\; \max_{u \in B_R(x^*)}\big\|\boldsymbol{x}^0 - \boldsymbol{u}\big\|^2 + 2\gamma_2^2\sum_{k=0}^{K-1}\big\|\widehat{\boldsymbol{g}}_{\boldsymbol{\eta}^k}(\widetilde{\boldsymbol{x}}^k) - \overline{\boldsymbol{g}}_{\boldsymbol{\eta}^k}(\widetilde{\boldsymbol{x}}^k)\big\|^2 \\
&\quad + 4\gamma_1\gamma_2\sum_{k=0}^{K-1}\Big(\big\|\overline{\boldsymbol{g}}_{\boldsymbol{\eta}^k}(\widetilde{\boldsymbol{x}}^k) - F(\widetilde{\boldsymbol{x}}^k)\big\|^2 + \big\|F(\boldsymbol{x}^k) - \overline{\boldsymbol{g}}_{\boldsymbol{\xi}^k}(\boldsymbol{x}^k)\big\|^2 + \big\|\overline{\boldsymbol{g}}_{\boldsymbol{\xi}^k}(\boldsymbol{x}^k) - \widehat{\boldsymbol{g}}_{\boldsymbol{\xi}^k}(\boldsymbol{x}^k)\big\|^2\Big) \\
&\quad + 2\gamma_2\max_{u \in B_R(x^*)}\sum_{k=0}^{K-1}\big\langle F(\widetilde{\boldsymbol{x}}^k) - \overline{\boldsymbol{g}}_{\boldsymbol{\eta}^k}(\widetilde{\boldsymbol{x}}^k), \widetilde{\boldsymbol{x}}^k - \boldsymbol{u}\big\rangle \\
&\quad + 2\gamma_2\max_{u \in B_R(x^*)}\sum_{k=0}^{K-1}\big\langle \overline{\boldsymbol{g}}_{\boldsymbol{\eta}^k}(\widetilde{\boldsymbol{x}}^k) - \widehat{\boldsymbol{g}}_{\boldsymbol{\eta}^k}(\widetilde{\boldsymbol{x}}^k), \boldsymbol{x}^k - \boldsymbol{u}\big\rangle.
\end{aligned}
$$

By Lemma E.1

$$2\gamma_2^2\mathbb{E}\left(\sum_{k=0}^{K-1}\left\|\widehat{\boldsymbol{g}}_{\boldsymbol{\eta}^k}(\widetilde{\boldsymbol{x}}^k) - \overline{\boldsymbol{g}}_{\boldsymbol{\eta}^k}(\widetilde{\boldsymbol{x}}^k)\right\|^2\right) \le 2\gamma_2^2\rho^2\sum_{k=0}^{K-1}\mathbb{E}[\mathbb{1}_k] \le 2\gamma_2^2\rho^2\frac{nB}{m} \overset{(60)}{\le} \frac{R^2}{32}. \tag{79}$$

and

$$4\gamma_1\gamma_2\mathbb{E}\left(\sum_{k=0}^{K-1}\left\|\overline{\boldsymbol{g}}_{\boldsymbol{\xi}^k}(\boldsymbol{x}^k) - \widehat{\boldsymbol{g}}_{\boldsymbol{\xi}^k}(\boldsymbol{x}^k)\right\|^2\right) \le 4\gamma_1\gamma_2\rho^2\sum_{k=0}^{K-1}\mathbb{E}\left[\mathbb{1}_{k-\frac{1}{2}}\right] \le 4\gamma_1\gamma_2\rho^2\frac{nB}{m} \overset{(59),(60)}{\le} \frac{R^2}{5} \tag{80}$$

By Corollary 2

$$4\gamma_1\gamma_2\mathbb{E}\left(\sum_{k=0}^{K-1}\left(\left\|\overline{\boldsymbol{g}}_{\boldsymbol{\eta}^k}(\widetilde{\boldsymbol{x}}^k) - F(\widetilde{\boldsymbol{x}}^k)\right\|^2 + \left\|F(\boldsymbol{x}^k) - \overline{\boldsymbol{g}}_{\boldsymbol{\xi}^k}(\boldsymbol{x}^k)\right\|^2\right)\right) \le \frac{8\gamma_1\gamma_2K}{n-2B-m} \overset{(59),(60)}{\le} \frac{R^2}{4}. \tag{81}$$

$$
\begin{aligned}
2\gamma_2 &\max_{u\in B_R(x^*)}\sum_{k=0}^{K-1}\left\langle\overline{\boldsymbol{g}}_{\boldsymbol{\eta}^k}(\widetilde{\boldsymbol{x}}^k) - \widehat{\boldsymbol{g}}_{\boldsymbol{\eta}^k}(\widetilde{\boldsymbol{x}}^k), \boldsymbol{x}^k - \boldsymbol{u}\right\rangle \\
\le\quad & 2\gamma_2\max_{u\in B_R(x^*)}\sum_{k=0}^{K-1}\left\langle\overline{\boldsymbol{g}}_{\boldsymbol{\eta}^k}(\widetilde{\boldsymbol{x}}^k) - \widehat{\boldsymbol{g}}_{\boldsymbol{\eta}^k}(\widetilde{\boldsymbol{x}}^k), \boldsymbol{x}^k - \boldsymbol{x}^*\right\rangle \\
& +2\gamma_2\max_{u\in B_R(x^*)}\sum_{k=0}^{K-1}\left\langle\overline{\boldsymbol{g}}_{\boldsymbol{\eta}^k}(\widetilde{\boldsymbol{x}}^k) - \widehat{\boldsymbol{g}}_{\boldsymbol{\eta}^k}(\widetilde{\boldsymbol{x}}^k), \boldsymbol{x}^* - \boldsymbol{u}\right\rangle \\
\le\quad & 2\gamma_2\sum_{k=0}^{K-1}\left\|\boldsymbol{x}^k - \boldsymbol{x}^*\right\|\left\|\widehat{\boldsymbol{g}}_{\boldsymbol{\eta}^k}(\widetilde{\boldsymbol{x}}^k) - \overline{\boldsymbol{g}}_{\boldsymbol{\eta}^k}(\widetilde{\boldsymbol{x}}^k)\right\| \\
& +2\gamma_2\max_{u\in B_R(x^*)}\sum_{k=0}^{K-1}\left\|\boldsymbol{x}^* - \boldsymbol{u}\right\|\left\|\widehat{\boldsymbol{g}}_{\boldsymbol{\eta}^k}(\widetilde{\boldsymbol{x}}^k) - \overline{\boldsymbol{g}}_{\boldsymbol{\eta}^k}(\widetilde{\boldsymbol{x}}^k)\right\| \\
\overset{\substack{Lemma\ E.1}}{\le}\quad & 6\gamma_2\rho R_0\mathbb{1}_k.
\end{aligned}
$$

Taking the full expectation of both sides of the result of the previous chain we derive

$$2\gamma_2\mathbb{E}\max_{u\in B_R(x^*)}\sum_{k=0}^{K-1}\left\langle\overline{\boldsymbol{g}}_{\boldsymbol{\eta}^k}(\widetilde{\boldsymbol{x}}^k) - \widehat{\boldsymbol{g}}_{\boldsymbol{\eta}^k}(\widetilde{\boldsymbol{x}}^k), \boldsymbol{x}^k - \boldsymbol{u}\right\rangle \le 6\gamma_2\rho R_0\mathbb{E}\mathbb{1}_k \le 6\gamma_2\rho R_0\frac{nB}{m} \overset{(60)}{\le} \frac{3}{4}R^2.$$

Now the last term

$$2\gamma_2\max_{u\in B_R(x^*)}\sum_{k=0}^{K-1}\left\langle F(\widetilde{\boldsymbol{x}}^k) - \overline{\boldsymbol{g}}_{\boldsymbol{\eta}^k}(\widetilde{\boldsymbol{x}}^k), \widetilde{\boldsymbol{x}}^k - \boldsymbol{u}\right\rangle \tag{82}$$

Following Beznosikov et al. [2023] one can derive the bound for the next term:

$$
\begin{aligned}
\mathbb{E}\left[\sum_{k=0}^{K-1}\left\langle F(\widetilde{\boldsymbol{x}}^k) - \overline{\boldsymbol{g}}_{\boldsymbol{\eta}^k}(\widetilde{\boldsymbol{x}}^k), \widetilde{\boldsymbol{x}}^k\right\rangle\right] &= \mathbb{E}\left[\sum_{k=0}^{K-1}\left\langle\mathbb{E}[F(\widetilde{\boldsymbol{x}}^k) - \overline{\boldsymbol{g}}_{\boldsymbol{\eta}^k}(\widetilde{\boldsymbol{x}}^k)\mid\widetilde{\boldsymbol{x}}^k], \widetilde{\boldsymbol{x}}^k\right\rangle\right] = 0, \\
\mathbb{E}\left[\sum_{k=0}^{K-1}\left\langle F(\widetilde{\boldsymbol{x}}^k) - \overline{\boldsymbol{g}}_{\boldsymbol{\eta}^k}(\widetilde{\boldsymbol{x}}^k), \boldsymbol{x}^0\right\rangle\right] &= \sum_{k=0}^{K-1}\left\langle\mathbb{E}[F(\widetilde{\boldsymbol{x}}^k) - \overline{\boldsymbol{g}}_{\boldsymbol{\eta}^k}(\widetilde{\boldsymbol{x}}^k)], \boldsymbol{x}^0\right\rangle = 0,
\end{aligned}
$$

we have

$$2\gamma_2 \mathbb{E}\left[\max_{\boldsymbol{u}\in B_R(x^*)} \sum_{k=0}^{K-1} \langle F(\widetilde{\boldsymbol{x}}^k) - \overline{\boldsymbol{g}}_{\boldsymbol{\eta}^k}(\widetilde{\boldsymbol{x}}^k), \widetilde{\boldsymbol{x}}^k - \boldsymbol{u}\rangle\right]$$

$$= 2\gamma_2 \mathbb{E}\left[\sum_{k=0}^{K-1} \langle F(\widetilde{\boldsymbol{x}}^k) - \overline{\boldsymbol{g}}_{\boldsymbol{\eta}^k}(\widetilde{\boldsymbol{x}}^k), \widetilde{\boldsymbol{x}}^k\rangle\right]$$

$$+ 2\gamma_2 \mathbb{E}\left[\max_{\boldsymbol{u}\in B_R(x^*)} \sum_{k=0}^{K-1} \langle F(\widetilde{\boldsymbol{x}}^k) - \overline{\boldsymbol{g}}_{\boldsymbol{\eta}^k}(\widetilde{\boldsymbol{x}}^k), -\boldsymbol{u}\rangle\right]$$

$$= 2\gamma_2 \mathbb{E}\left[\max_{\boldsymbol{u}\in B_R(x^*)} \sum_{k=0}^{K-1} \langle F(\widetilde{\boldsymbol{x}}^k) - \overline{\boldsymbol{g}}_{\boldsymbol{\eta}^k}(\widetilde{\boldsymbol{x}}^k), -\boldsymbol{u}\rangle\right]$$

$$= 2\gamma_2 \mathbb{E}\left[\sum_{k=0}^{K-1} \langle F(\widetilde{\boldsymbol{x}}^k) - \overline{\boldsymbol{g}}_{\boldsymbol{\eta}^k}(\widetilde{\boldsymbol{x}}^k), \boldsymbol{x}^0\rangle\right]$$

$$+ 2\gamma_2 \mathbb{E}\left[\max_{\boldsymbol{u}\in B_R(x^*)} \sum_{k=0}^{K-1} \langle F(\widetilde{\boldsymbol{x}}^k) - \overline{\boldsymbol{g}}_{\boldsymbol{\eta}^k}(\widetilde{\boldsymbol{x}}^k), -\boldsymbol{u}\rangle\right]$$

$$= 2\gamma_2 K \mathbb{E}\left[\max_{\boldsymbol{u}\in B_R(x^*)} \left\langle \frac{1}{K}\sum_{k=0}^{K-1}(F(\widetilde{\boldsymbol{x}}^k) - \overline{\boldsymbol{g}}_{\boldsymbol{\eta}^k}(\widetilde{\boldsymbol{x}}^k)), \boldsymbol{x}^0 - \boldsymbol{u}\right\rangle\right]$$

$$\overset{(11)}{\leq} 2\gamma_2 K \mathbb{E}\left[\max_{\boldsymbol{u}\in B_R(x^*)} \left\{\frac{\gamma_2}{2}\left\|\frac{1}{K}\sum_{k=0}^{K-1}(F(\widetilde{\boldsymbol{x}}^k) - \overline{\boldsymbol{g}}_{\boldsymbol{\eta}^k}(\widetilde{\boldsymbol{x}}^k))\right\|^2 + \frac{1}{2\gamma_2}\|\boldsymbol{x}^0 - \boldsymbol{u}\|^2\right\}\right]$$

$$= \gamma_2^2 \mathbb{E}\left[\left\|\sum_{k=0}^{K-1}(F(\widetilde{\boldsymbol{x}}^k) - \overline{\boldsymbol{g}}_{\boldsymbol{\eta}^k}(\widetilde{\boldsymbol{x}}^k))\right\|^2\right] + \max_{\boldsymbol{u}\in B_R(x^*)}\|\boldsymbol{x}^0 - \boldsymbol{u}\|^2.$$

We notice that $\mathbb{E}[F(\widetilde{\boldsymbol{x}}^k) - \overline{\boldsymbol{g}}_{\boldsymbol{\eta}^k}(\widetilde{\boldsymbol{x}}^k) \mid F(\widetilde{\boldsymbol{x}}^0) - \overline{\boldsymbol{g}}_{\boldsymbol{\eta}^0}(\widetilde{\boldsymbol{x}}^0), \ldots, F(\widetilde{\boldsymbol{x}}^{k-1}) - \overline{\boldsymbol{g}}_{\boldsymbol{\eta}^{k-1}}(\widetilde{\boldsymbol{x}}^{k-1})] = 0$ for all $k \geq 1$, i.e., conditions of Lemma C.2 are satisfied. Therefore, applying Lemma C.2, we get

$$2\gamma_2 \mathbb{E}\left[\max_{\boldsymbol{u}\in B_R(x^*)} \sum_{k=0}^{K-1} \langle F(\widetilde{\boldsymbol{x}}^k) - \overline{\boldsymbol{g}}_{\boldsymbol{\eta}^k}(\widetilde{\boldsymbol{x}}^k), \widetilde{\boldsymbol{x}}^k - \boldsymbol{u}\rangle\right] \tag{83}$$

$$\leq \gamma_2^2 \sum_{k=0}^{K-1} \mathbb{E}[\|F(\widetilde{\boldsymbol{x}}^k) - \overline{\boldsymbol{g}}_{\boldsymbol{\eta}^k}(\widetilde{\boldsymbol{x}}^k)\|^2] \tag{84}$$

$$+ \max_{\boldsymbol{u}\in B_R(x^*)}\|\boldsymbol{x}^0 - \boldsymbol{u}\|^2 \tag{85}$$

$$\leq \frac{\gamma_2^2 K\sigma^2}{n - 2B - m} + \max_{\boldsymbol{u}\in B_R(x^*)}\|\boldsymbol{x}^0 - \boldsymbol{u}\|^2 \tag{86}$$

$$\overset{(59),(60)}{\leq} \frac{9}{8}R^2. \tag{87}$$

Assembling the above results together gives

$$2\gamma_2 K \mathbb{E}\text{Gap}_{B_R(x^*)}(\overline{\boldsymbol{x}}^K) \leq \frac{R^2}{32} + \frac{R^2}{5} + \frac{R^2}{4} + \frac{3}{4}R^2 + \frac{9}{8}R^2 \leq 3R^2. \tag{88}$$

$\square$

**Corollary 10.** *Let assumptions of Theorem 12 hold. Then* $\mathbb{E}\left[\text{Gap}_{B_R(x^*)}(\overline{\boldsymbol{x}}^K)\right] \leq \varepsilon$ *holds after*

$$K = \mathcal{O}\left(\frac{LR^2}{\varepsilon} + \frac{\sigma^2 R^2}{n\varepsilon^2} + \frac{\sigma n^2 R}{m\varepsilon}\right)$$

*iterations of* SEG-CC.

*Proof.*

$$\mathbb{E}\Big[\mathtt{Gap}_{B_R(x^*)}\big(\overline{\boldsymbol{x}}^K\big)\Big] \leq \frac{3R^2}{2\gamma_2 K} \quad \leq \quad \frac{3R^2}{2K}\left(4L + \sqrt{\frac{64\sigma^2 K}{(n-2B-m)R^2}} + \sqrt{\frac{64\rho^2 B^2 n^2}{m^2 R^2}}\right)$$

$$\leq \quad \frac{6R^2}{K} + \sqrt{\frac{144\sigma^2 R^2}{(n-2B-m)K}} + \frac{12\rho B n R}{mK}$$

Let us chose $K$ such that each of the last three terms less or equal $\varepsilon/3$, then

$$K = \max\left(\frac{18LR^2}{\varepsilon}, \frac{144 \cdot 9\sigma^2 R^2}{(n-2B-m)\varepsilon^2}, \frac{36\rho B n R}{m\varepsilon}\right),$$

where $\rho^2 = q\sigma^2$ with $q = 2C^2 + 12 + \frac{12}{n-2B-m}$ and $C = \mathcal{O}(1)$ by Lemma E.1. The latter implies that

$$\mathbb{E}\Big[\mathtt{Gap}_{B_R(x^*)}\big(\overline{\boldsymbol{x}}^K\big)\Big] \leq \varepsilon.$$

Using the definition of $\rho$ from Lemma E.1 and if $B \leq \frac{n}{4}$, $m << n$ the bound for $K$ can be easily derived. □

### E.4 Proofs for R-SEG-CC

#### E.4.1 Quasi Strongly Monotone Case

---
**Algorithm 8** R-SEG-CC
---
**Input:** $\boldsymbol{x}^0$ – starting point, $r$ – number of restarts, $\{\gamma_t\}_{t=1}^r$ – stepsizes for SEG-CC (Alg. 7), $\{K_t\}_{t=1}^r$ – number of iterations for SEG-CC (Alg. 7),
1: $\widehat{\boldsymbol{x}}^0 = \boldsymbol{x}^0$
2: **for** $t = 1, 2, \ldots, r$ **do**
3:     Run SEG-CC (Alg. 7) for $K_t$ iterations with stepsize $\gamma_t$, starting point $\widehat{\boldsymbol{x}}^{t-1}$,
4:     Define $\widehat{\boldsymbol{x}}^t$ as $\widehat{\boldsymbol{x}}^t = \frac{1}{K_t} \sum_{k=0}^{K_t} \boldsymbol{x}^{k,t}$, where $\boldsymbol{x}^{0,t}, \boldsymbol{x}^{1,t}, \ldots, \boldsymbol{x}^{K_t,t}$ are the iterates produced by SEG-CC .
5: **end for**
**Output:** $\widehat{\boldsymbol{x}}^r$

---

**Theorem** (Theorem 7 duplicate)**.** *Let Assumptions 1, 3, 4 hold. Then, after* $r = \left\lceil \log_2 \frac{R^2}{\varepsilon} \right\rceil - 1$ *restarts* R-SEG-CC *(Algotithm 8) with* $\gamma_{1_t} = \min\left\{\frac{1}{2L}, \sqrt{\frac{(G-B-m)R^2}{16\sigma^2 2^t K_t}}, \sqrt{\frac{mR^2}{8q\sigma^2 2^t Bn}}\right\}$, $\gamma_{2_t} = \min\left\{\frac{1}{4L}, \sqrt{\frac{m^2 R^2}{64q\sigma^2 2^t B^2 n^2}}, \sqrt{\frac{(G-B-m)R^2}{64\sigma^2 K_t}}\right\}$ *and* $K_t = \left\lceil \max\left\{\frac{8L}{\mu}, \frac{16n\sigma B\sqrt{q2^t}}{m\mu R}, \frac{256\sigma^2 2^t}{(G-B-m)\mu^2 R^2}\right\}\right\rceil$, *where* $R \geq \|\boldsymbol{x}^0 - \boldsymbol{x}^*\|$ *outputs* $\widehat{\boldsymbol{x}}^r$ *such that* $\mathbb{E}\|\widehat{\boldsymbol{x}}^r - \boldsymbol{x}^*\|^2 \leq \varepsilon$. *Moreover, the total number of executed iterations of* SEG-CC *is*

$$\sum_{t=1}^r K_t = \mathcal{O}\left(\frac{\ell}{\mu} \log \frac{\mu R_0^2}{\varepsilon} + \frac{\sigma^2}{(n-2B-m)\mu\varepsilon} + \frac{nB\sigma}{m\sqrt{\mu\varepsilon}}\right). \tag{89}$$

*Proof of Theorem 7.* $\overline{\boldsymbol{x}}^K = \frac{1}{K} \sum_{k=0}^{K-1} \widetilde{\boldsymbol{x}}^k$

$$\mu\mathbb{E}\Big[\|\overline{\boldsymbol{x}}^K - \boldsymbol{x}^*\|^2\Big] = \mu\mathbb{E}\left[\left\|\frac{1}{K}\sum_{k=0}^{K-1}(\widetilde{\boldsymbol{x}}^k - \boldsymbol{x}^*)\right\|^2\right] \leq \mu\mathbb{E}\left[\frac{1}{K}\sum_{k=0}^{K-1}\|\widetilde{\boldsymbol{x}}^k - \boldsymbol{x}^*\|^2\right]$$

$$= \frac{\mu}{K}\sum_{k=0}^{K-1}\mathbb{E}\Big[\|\widetilde{\boldsymbol{x}}^k - \boldsymbol{x}^*\|^2\Big] \stackrel{\text{(QSM)}}{\leq} \frac{1}{K}\sum_{k=0}^{K-1}\mathbb{E}\big[\langle F(\widetilde{\boldsymbol{x}}^k), \widetilde{\boldsymbol{x}}^k - \boldsymbol{x}^*\rangle\big].$$

Theorem 11 implies that SEG-CC with

$$\gamma_1 = \min\left\{\frac{1}{2L}, \sqrt{\frac{(n-2B-m)R^2}{16\sigma^2 K}}, \sqrt{\frac{mR^2}{8\rho^2 Bn}}\right\},$$

$$\gamma_2 = \min\left\{\frac{1}{4L}, \sqrt{\frac{m^2 R^2}{64\rho^2 B^2 n^2}}, \sqrt{\frac{(n-2B-m)R^2}{64\sigma^2 K}}\right\},$$

guarantees

$$\mu\mathbb{E}\left[\left\|\overline{\boldsymbol{x}}^K - \boldsymbol{x}^*\right\|^2\right] \leq \frac{R_0^2}{\gamma_2 K}$$

after $K$ iterations.

After the first restart we have

$$\mathbb{E}\left[\left\|\widehat{\boldsymbol{x}}^1 - \boldsymbol{x}^*\right\|^2\right] \leq \frac{R_0^2}{\mu\gamma_{2_1} K_1} \leq \frac{R_0^2}{2},$$

since $\mu\gamma_{2_1} K_1 \geq 2$.

Next, assume that we have $\mathbb{E}[\|\widehat{\boldsymbol{x}}^t - \boldsymbol{x}^*\|^2] \leq \frac{R_0^2}{2^t}$ for some $t \leq r-1$. Then, Theorem 11 implies that

$$\mathbb{E}\left[\left\|\widehat{\boldsymbol{x}}^{t+1} - \boldsymbol{x}^*\right\|^2 \mid \widehat{\boldsymbol{x}}^t\right] \leq \frac{\|\widehat{\boldsymbol{x}}^t - \boldsymbol{x}^*\|^2}{\mu\gamma_{2_t} K_t}.$$

Taking the full expectation from the both sides of previous inequality we get

$$\mathbb{E}\left[\left\|\widehat{\boldsymbol{x}}^{t+1} - \boldsymbol{x}^*\right\|^2\right] \leq \frac{\mathbb{E}[\|\widehat{\boldsymbol{x}}^t - \boldsymbol{x}^*\|^2]}{\mu\gamma_{2_t} K_t} \leq \frac{R_0^2}{2^t \mu\gamma_{2_t} K_t} \leq \frac{R_0^2}{2^{t+1}}.$$

Therefore, by mathematical induction we have that for all $t = 1, \ldots, r$

$$\mathbb{E}\left[\|\widehat{\boldsymbol{x}}^t - \boldsymbol{x}^*\|^2\right] \leq \frac{R_0^2}{2^t}.$$

Then, after $r = \left\lceil \log_2 \frac{R_0^2}{\varepsilon} \right\rceil - 1$ restarts of SEG-CC we have $\mathbb{E}\left[\|\widehat{\boldsymbol{x}}^r - \boldsymbol{x}^*\|^2\right] \leq \varepsilon$. The total number of iterations executed by SEG-CC is

$$\sum_{t=1}^r K_t = \mathcal{O}\left(\sum_{t=1}^r \max\left\{\frac{L}{\mu}, \frac{\sigma^2 2^t}{(n-2B-m)\mu^2 R_0^2}, \frac{nB\rho 2^{\frac{t}{2}}}{m\mu R_0}\right\}\right)$$

$$= \mathcal{O}\left(\frac{L}{\mu} r + \frac{\sigma^2 2^r}{(n-2B-m)\mu^2 R_0^2} + \frac{nB\rho 2^{\frac{r}{2}}}{m\mu R_0}\right)$$

$$= \mathcal{O}\left(\frac{L}{\mu} \log \frac{\mu R_0^2}{\varepsilon} + \frac{\sigma^2}{(n-2B-m)\mu^2 R_0^2} \cdot \frac{\mu R_0^2}{\varepsilon} + \frac{nB\rho}{m\mu R_0} \cdot \sqrt{\frac{\mu R_0^2}{\varepsilon}}\right)$$

$$= \mathcal{O}\left(\frac{L}{\mu} \log \frac{\mu R_0^2}{\varepsilon} + \frac{\sigma^2}{(n-2B-m)\mu\varepsilon} + \frac{nB\rho}{m\sqrt{\mu\varepsilon}}\right),$$

that together with $\rho^2 = q\sigma^2$ with $q = 2C^2 + 12 + \frac{12}{n-2B-m}$ and $C = \mathcal{O}(1)$ given by Lemma E.1 implies the result of the theorem. $\square$

**Corollary 11.** *Let assumptions of 7 hold. Then* $\mathbb{E}\left[\|\widehat{\boldsymbol{x}}^r - \boldsymbol{x}^*\|^2\right] \leq \varepsilon$ *holds after*

$$\sum_{t=1}^r K_t = \mathcal{O}\left(\frac{L}{\mu} \log \frac{\mu R^2}{\varepsilon} + \frac{\sigma^2}{n\mu\varepsilon} + \frac{n^2\sigma}{m\sqrt{\mu\varepsilon}}\right) \tag{90}$$

*iterations of* SEG-CC.

*Proof.* Using the definition of $\rho$ from Lemma E.1 and if $B \leq \frac{n}{4}$, $m \ll n$ the bound for $\sum_{t=1}^r K_t$ can be easily derived. $\square$

# F   Extra Experiments and Experimental details

## F.1   Quadratic games

Firstly, let us clarify notations made in (7):

$$\boldsymbol{x} = \begin{pmatrix} y \\ z \end{pmatrix}, \quad \mathbf{A}_i = \begin{pmatrix} \mathbf{A}_{1,i} & \mathbf{A}_{2,i} \\ -\mathbf{A}_{2,i} & \mathbf{A}_{3,i} \end{pmatrix}, \quad b_i = \begin{pmatrix} b_{1,i} \\ b_{2,i} \end{pmatrix},$$

with symmetric matrices $\mathbf{A}_{j,i}$ s.t. $\mu \mathbf{I} \preccurlyeq \mathbf{A}_{j,i} \preccurlyeq \ell \mathbf{I}$, $\mathbf{A}_i \in \mathbb{R}^{d \times d}$ and $b_i \in \mathbb{R}^d$.

**Data generation.**   For each $j$ we sample real random matrix $\mathbf{B}_i$ with elements sampled from a normal distribution. Then we compute the eigendecomposition and obtain the following $\mathbf{B}_i = \mathbf{U}_i^T \mathbf{D}_i \mathbf{U}_i$ with diagonal $\mathbf{D}_i$. Next, we scale $\mathbf{D}_i$ and obtain $\widehat{\mathbf{D}}_i$ with elements lying in $[\mu, \ell]$. Finally we compose $\mathbf{A}_{j,i}$ as $\mathbf{A}_{j,i} = \mathbf{U}_i^T \widehat{\mathbf{D}}_i \mathbf{U}_i$. This process ensures that eigenvalues of $\mathbf{A}_{j,i}$ all lie between $\mu$ and $\ell$, and thus $F(\boldsymbol{x})$ is strongly monotone and cocoercive. Vectors $b_i \in \mathbb{R}^d$ are sampled from a normal distribution with variance $10/d$.

**Experimental setup.**   For all the experiments we choose $\ell = 100$, $\mu = 0.1$, $s = 1000$ and $d = 50$.

For the experiments presented in the Appendix we simulate $n = 20$ nodes on a single machine and $B = 4$. For methods with checks of computations the only one peer attacks per iteration.

We used RFA with 5 buckets bucketing as an aggregator since it showed the best performance. For approximating the median we used Weiszfeld's method with 10 iterations and parameter $\nu = 0.1$ Pillutla et al. [2022].

RDEG [Adibi et al., 2022] provably works only if $n \geq 100$ we manually selected parameter $\epsilon = 0.5$ using a grid-search and picking the best performing value.

We present experiments with different attack (bit flipping (BF), random noise (RN), inner product manipulation (IPM) Xie et al. [2019] and "a little is enough" (ALIE) Baruch et al. [2019]) and different batchsizes (bs) 1, 10 and 100. If an attack has a parameter it is specified in the brackets on each plot.

**No checks.**   Firstly we provide a detailed comparison between methods that do not check computation with fixed learning rate value $\gamma = 3.3e - 5$. Code for quadratic games is available at https://github.com/nazya/sgda-ra[7].

---

[7]Code is based on https://github.com/hugobb/sgda

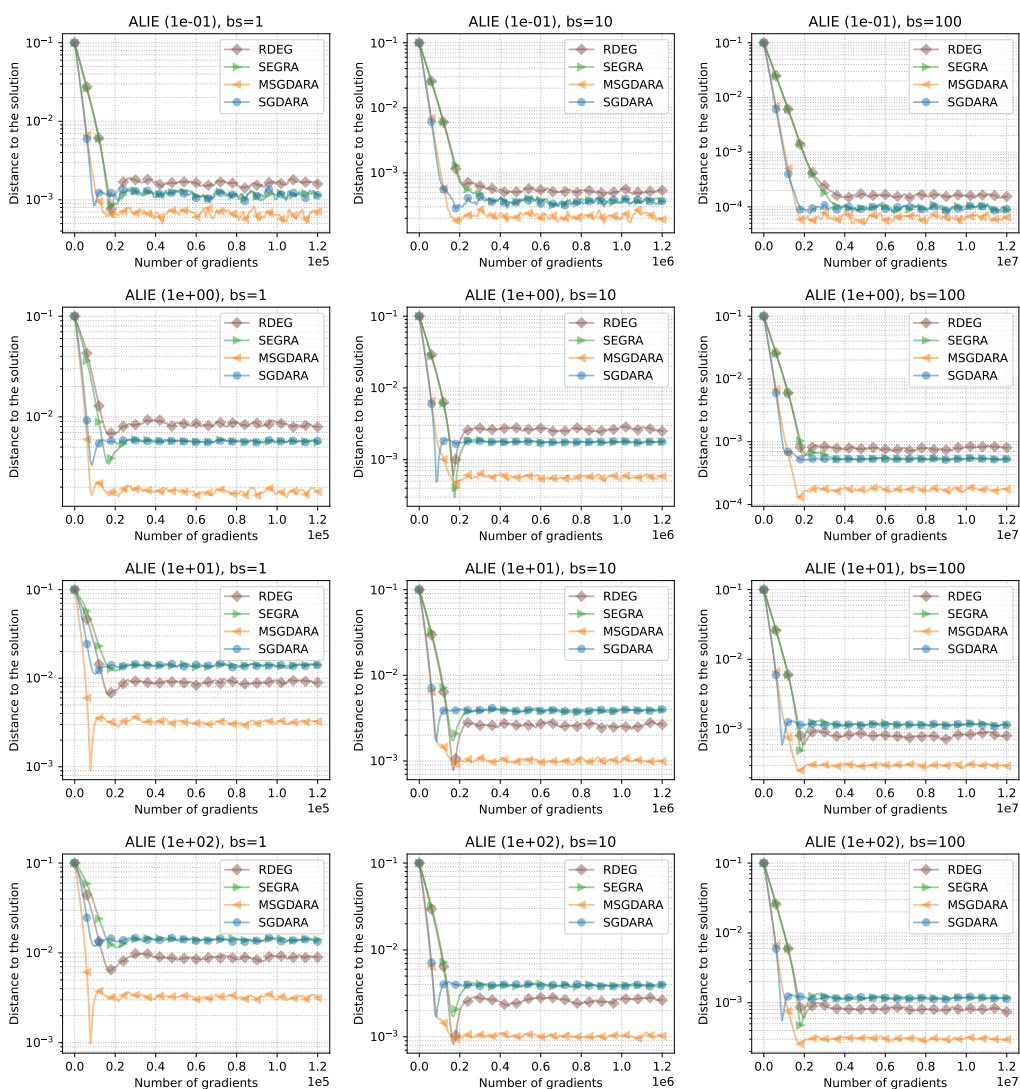

Figure 3: Distance to the solution ALIE attack with various of parameter values and batchsizes.

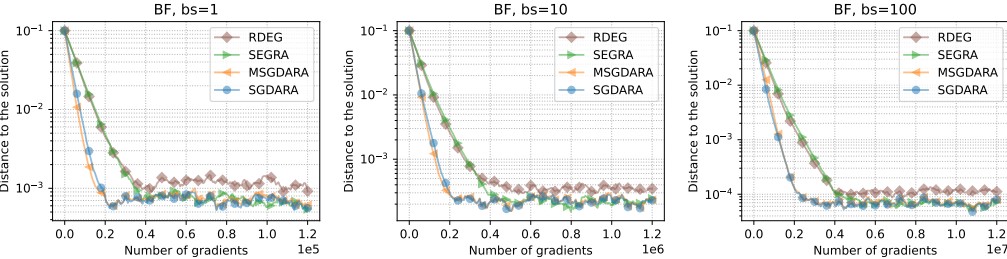

Figure 4: Distance to the solution BF attack with various batchsizes.

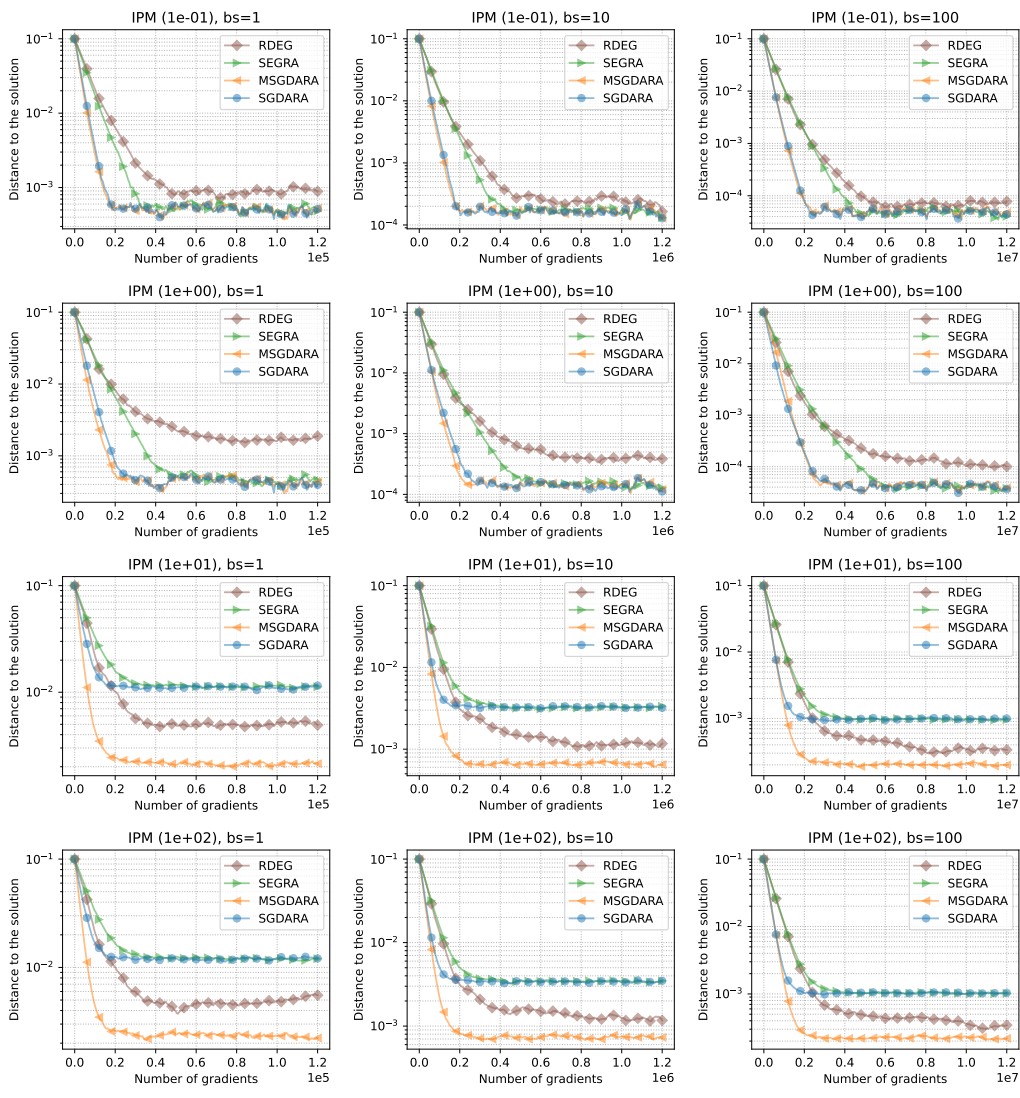

Figure 5: Distance to the solution under IPM attack with various parameter values and batchsizes.

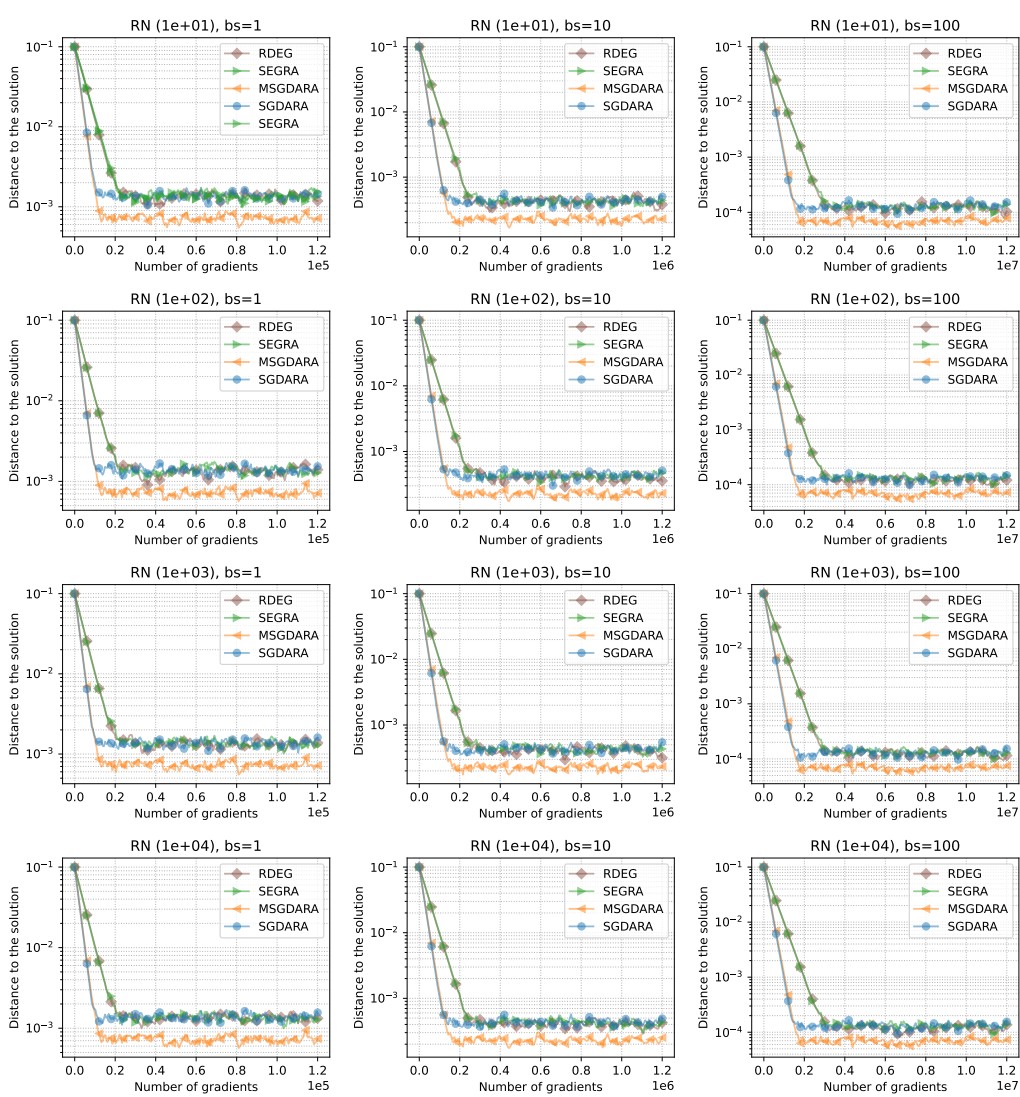

Figure 6: Distance to the solution under RN attack with various parameter values and batchsizes.

**Checks of Computations.** Next, using the same setup, we compare M-SGDA-RA, which showed the best performance in the above experiments, with methods that check computations. With checks of computation the best strategy for attackers is that at each iteration only one peer attacks, since it maximizes the expected number of rounds with the presence of actively malicious peers. So the comparison in this paragraph is performed in this setup.

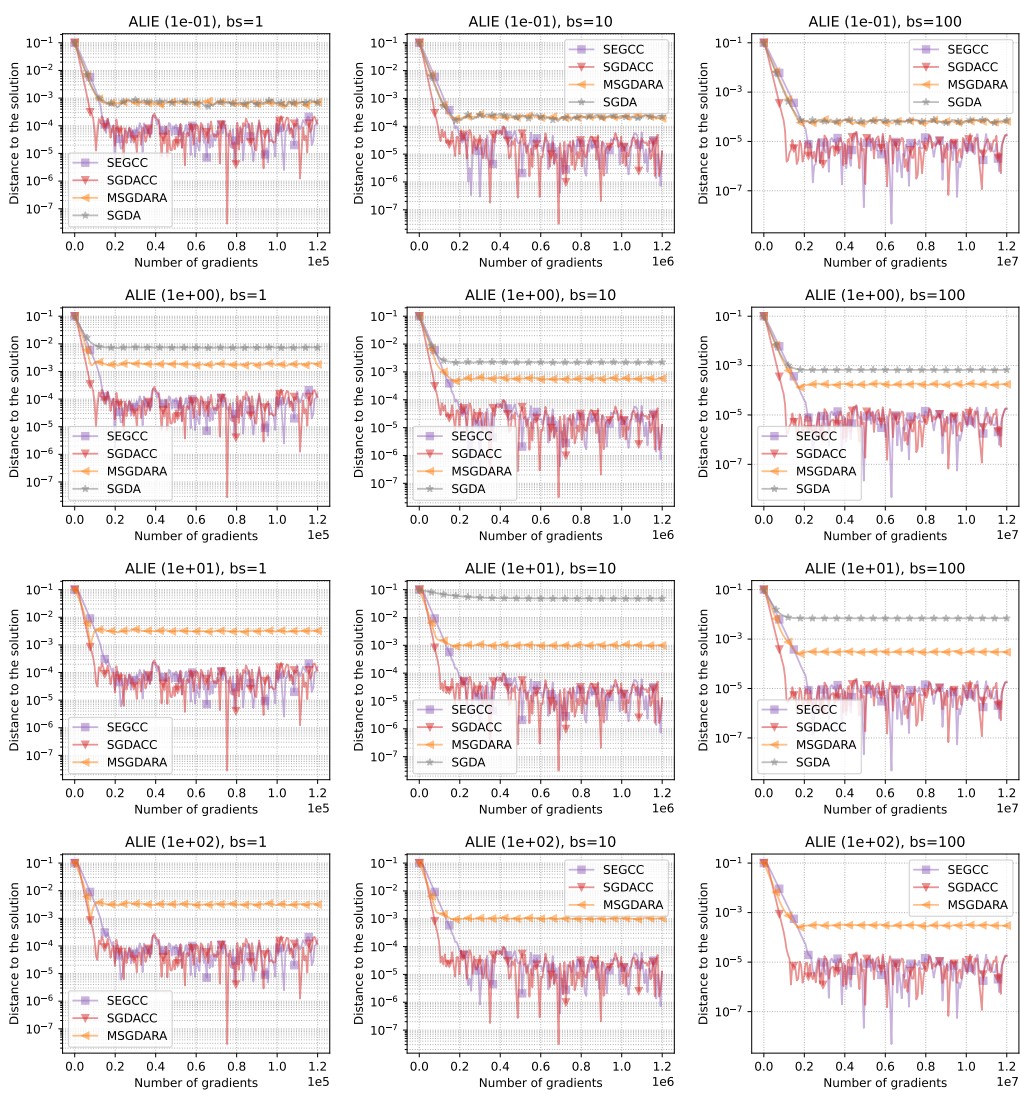

Figure 7: Distance to the solution under ALIE attack with various parameter values and batchsizes.

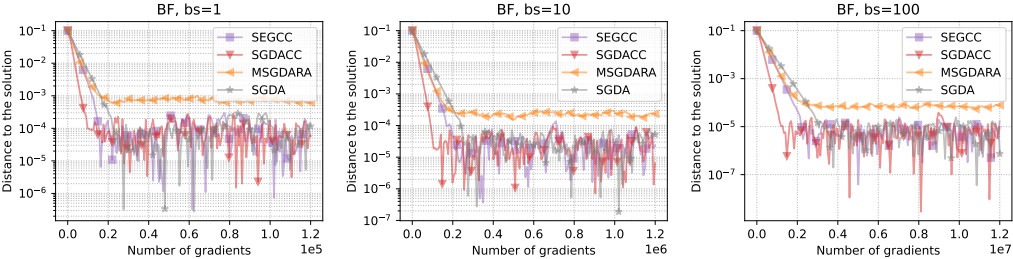

Figure 8: Distance to the solution under BF attack with various batchsizes.

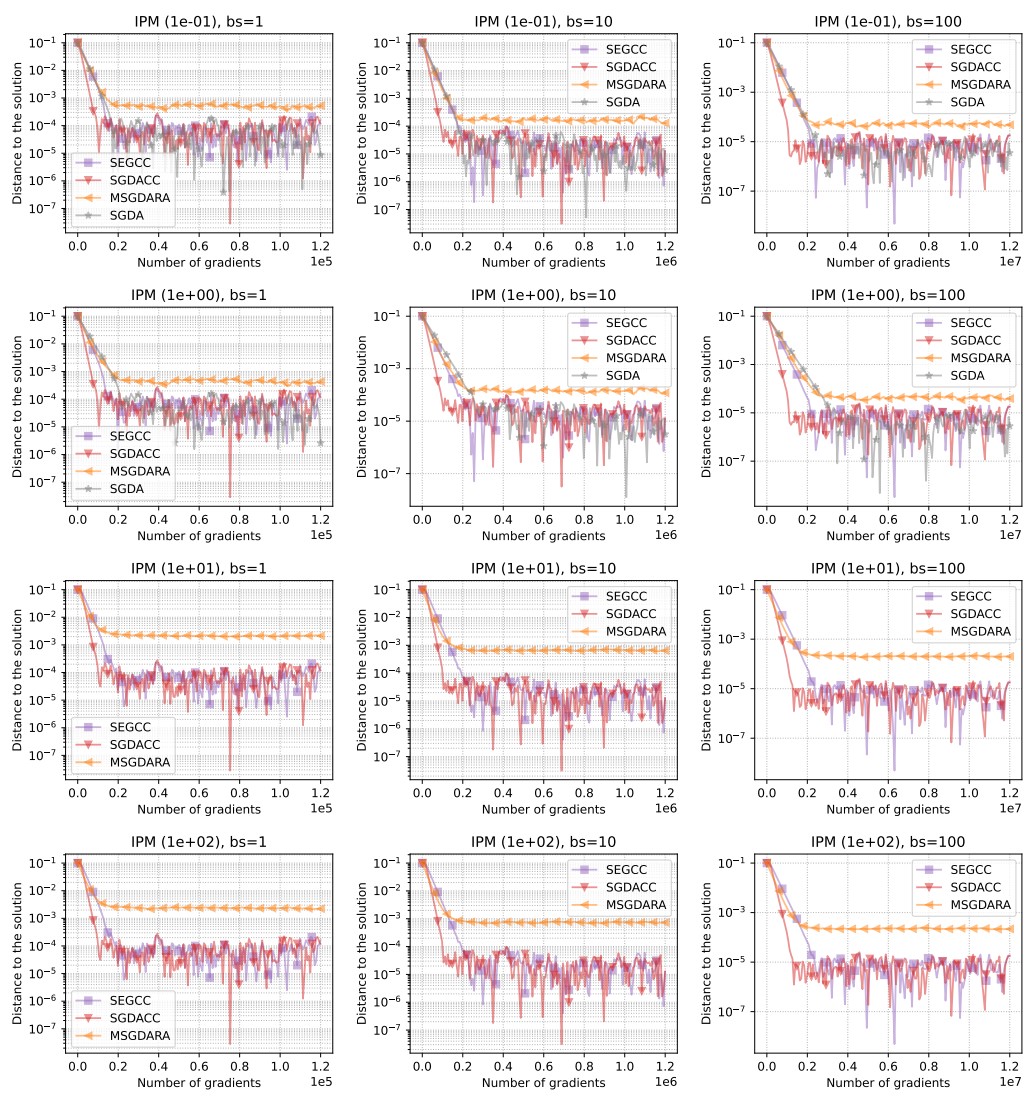

Figure 9: Distance to the solution under IPM attack with various parameter values and batchsizes.

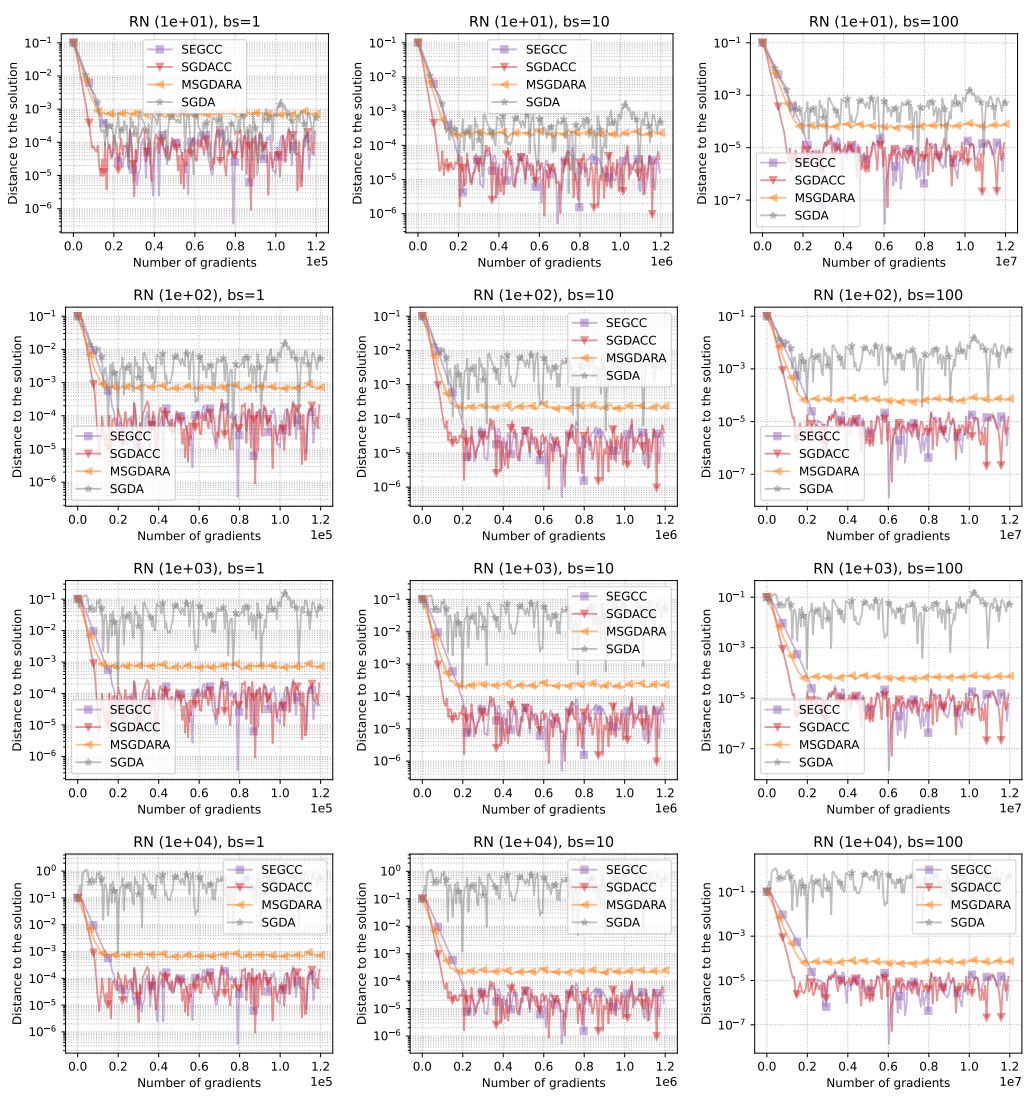

Figure 10: Distance to the solution under RN attack with various parameter values and batchsizes.

**Learning rates.** We conducted extra experiments to show the dependence on different learning rate values $1e-5, 2e-5, 5e-6$.

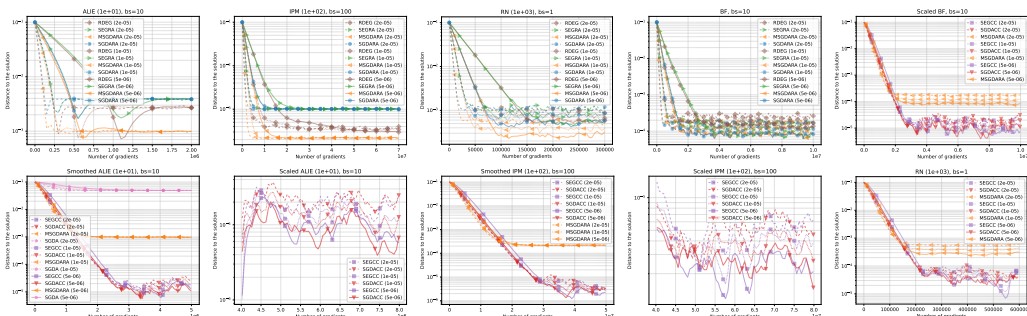

Figure 11: Distance to the solution under various attacks, batchsizes (bs) and learning rates (lr).

### F.2  Generative Adversarial Networks

One of the most well-known frameworks in which the objective function is a variational inequality is generative adversarial networks (GAN) Goodfellow et al. [2014]. In the simplest case of this setting, we have a generator $G : \mathbb{R}^z \to \mathbb{R}^d$ and a discriminator $D : \mathbb{R}^d \to \mathbb{R}$, where $z$ denotes the dimension of the latent space. The objective function can be written as

$$\min_G \max_D \quad \mathbb{E}_x \log(D(x)) + \mathbb{E}_z \log(1 - D(G(z))). \tag{91}$$

Here, it is understood that $D$ and $G$ are modeled as neural nets and can be optimized in the distributed setting with gradient descent ascent algorithms. However, due to the complexity of the GANs framework, tricks and adjustments are being employed to ensure good results, such as the Wasserstein GAN formulation [Gulrajani et al., 2017] with Lipschitz constraint on $D$ and the spectral normalization [Miyato et al., 2018] trick to ensure the Lipschitzness of $D$. The discriminator can thus benefit in practice from multiple gradient ascent steps per gradient descent step on the generator. In addition, Adam [Kingma and Ba, 2014] is often used for GANs as they can be very slow to converge and not perform as well with vanilla SGD.

Therefore, in our implementation of GANs in the distributed setting, we employ all of these techniques and show improvements when we add checks of gradient computations to the server. As for the gradients in our implementation, we can think of the accumulated Adam steps within the clients as "generalized gradients" and aggregate them in the server with checks of computations (by rewinding model and optimizer state). We tried aggregation after each descent or ascent step, after full descent-ascent step, and after multiple descent-ascent steps. For the first case, we found that GANs converge much more slowly. For the third case, the performance is better but checks of computations take more time. Thus, we choose to report the performance for the second case: aggregations of a full descent-ascent step. Though, we note that experiments for the other cases suggest similar improvements.

The dataset we chose for this experiment is CIFAR-10 [Krizhevsky et al., 2009] because it is more realistic than MNIST yet is still tractable to simulate in the distributed setting. We let $n = 10$, $B = 2$, and choose a learning rate of 0.001, $\beta_1 = 0.5$, and $\beta_2 = 0.9$ with a batch size of 64. We run the algorithms for 4600 epochs. We could not average across runs as the simulation is very compute intensive and the benefits are obvious. We compare SGDA-RA (RFA with bucket size 2) and SGDA-CC under the following byzantine attacks: i) no attack (NA), ii) label flipping (LF), iii) inner product manipulation (IPM) [Xie et al., 2019], and iv) a little is enough (ALIE) [Baruch et al., 2019]. The architecture of the GAN follows Miyato et al. [2018].

We show the results in Figure 12. The improvements are most significant for the ALIE attack. Even when no attacks are present, checks of computations only slightly affects convergence speed. This experiment should further justify our proposed algorithm and its real-world benefits even for a setting as complex as distributed GANs. Code for GANs is available at https://github.com/zeligism/vi-robust-agg.

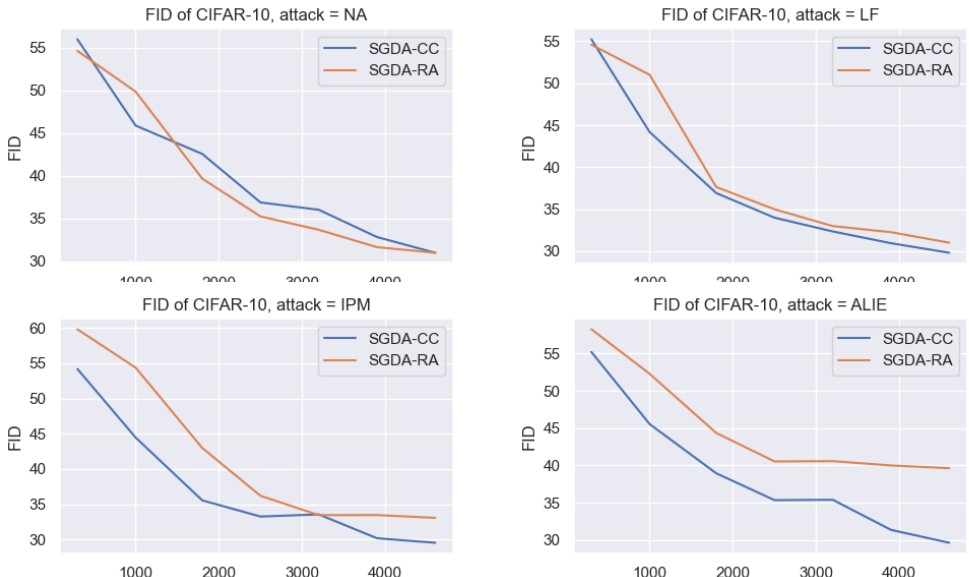

Figure 12: Comparison of FID to CIFAR-10 per epoch between SGDA-CC and SGDA-RA. The FID is calculated on 50,000 samples. (lower = better).

