# OpenReview forum: "Byzantine-Tolerant Methods for Distributed Variational Inequalities"
_NeurIPS.cc/2023/Conference — NeurIPS 2023 poster_

### Official Review · Reviewer_AxmQ · 2023-07-05

**Soundness:** 3 good
**Presentation:** 3 good
**Contribution:** 3 good
**Rating:** 7
**Confidence:** 4

**Summary:**

**Summary:** The authors propose robust optimization methods for solving distributed variational inequalities (VI). The authors provide theoretical guarantees for numerous popular algorithms for solving min-max (more generally VI) problems. The authors corroborate their theoretical findings with experiments on quadratic games and robust neural network training.

**Strengths:**

**Strengths:**

-	The authors provide guarantees for a set of algorithms for solving VIs with application to min-max optimization. Specifically, the authors propose a robust version of SGDA and SEG and provide theoretical convergence guarantees for the proposed algorithms. The authors show that if the batch size at each client is large enough the algorithms can converge to desired accuracy.
-	The authors propose momentum versions of the vanilla algorithms and establish improved convergence performance. The authors establish convergence to a neighborhood whose size can be reduced with increasing batch sizes.
-	The authors also propose random checks of computations versions of the methods for the homogeneous setting and provide a theoretical analysis of the proposed methods.
-	The authors evaluate the convergence performance of the proposed algorithms on a quadratic game and for robust neural network training.



**Weaknesses:**

**Weaknesses:**

-	A major weakness of the proposed results is the requirement of large batch sizes to ensure convergence to the desired accuracy. Is it feasible to circumvent this requirement?
-	Another drawback of the paper is that it is written in a very dense manner. The authors have attempted to include a host of results in the main paper which has left no space for stating the algorithms along with their discussions in the main paper. It is advisable to make the paper more accessible to the readers by including detailed algorithms in the main paper.
-	The results are presented in terms of iterate convergence but in the contribution section, the authors have mentioned that their result implies exact convergence for the non-convex case which requires convergence in terms of gradient norm. Please clarify why iterate convergence implies convergence for the non-convex case.
-	Some of the discussion from the contribution section is missing in the respective sections. For example, the discussion on the convergence of non-convex functions is not included in the section where momentum versions of the algorithms are presented.

---

Updated the score.



**Questions:**

Please see weaknesses section above.

---

> ### Author Rebuttal · Authors · 2023-08-08
>
> We thank the reviewer for the detailed feedback and a positive evaluation of our work. Below we address the reviewer’s questions and concerns.
>
> >**Large batch sizes requirement.**
>
> Part of our results (Theorems 1-3) indeed require large batches: for SGDA-RA and SEG-RA this limitation cannot be bypassed (see lines 172-173), and M-SGDA-RA it is unclear whether one can circumvent this issue since it is unknown even in the case of minimization problems (see lines 200-204). However, the proposed methods with random checks of computation **do not require large batches**: Theorems 4-7 are valid even if the batchsize equals $1$. Moreover, these results establish the convergence to any predefined optimization error.
>
> >**Paper is written in a dense manner.**
>
> We thank the reviewer for the suggestion. Indeed, we can move some of the results to the appendix as well as some details on the numerical experiments and use this space for providing more details about the algorithms and giving more discussion of the derived results. In the case of acceptance, there will also be an extra page for addressing the reviewers’ comments. We promise to add all necessary details requested by the reviewers in this case.
>
> >**Convergence in the non-convex case.**
>
> We do not claim that we have the results for non-convex problems: in lines 103-105, we emphasize that **the existing results** about the Byzantine robustness of the methods with client momentum are derived for non-convex minimization problems. In our work, we found out that for quasi-strongly monotone variational inequalities, it is not an easy task: according to our analysis, the method with momentum still requires large batches. Since monotonicity can be seen as a generalization of convexity to the case of operators, it is natural to ask a simpler question: can we circumvent this issue in the case of convex minimization problems? This remains an open question as we explain in lines 201-203: the current analysis of Momentum-SGD in the convex case relies on the unbiasedness of the stochastic gradient. Since robust aggregation breaks the unbiasedness, it is unclear for now how to fix this issue.
>
> We will make our statements clearer to avoid the readers’ confusion.

---

> > ### Comment · Reviewer_AxmQ · 2023-08-16
> > **Thank you for the response**
> >
> > I thank the authors for their responses. I believe the paper is worthy of publication, however, needs some minor changes in writing to make it more accessible to the readers.
> >
> > I have updated my score accordingly.

---

> > > ### Author Response · Authors · 2023-08-16
> > > **Thank you for the feedback**
> > >
> > > We thank the reviewer for the reply and for the increase in the score. Following the reviewer's suggestions and requests, we will apply all necessary changes to the final version.

---

### Official Review · Reviewer_vExH · 2023-07-05

**Soundness:** 3 good
**Presentation:** 2 fair
**Contribution:** 3 good
**Rating:** 6
**Confidence:** 4

**Summary:**

This paper studies convergence guarantees of Byzantine tolerant methods in the context of distributed variational inequalities.

**Strengths:**

The paper seems to contain a lot of work.

**Weaknesses:**

In my opinion, the main weakness of the paper is its writing. I acknowledge that I have not read the papers most related to this one, [Karimireddy et al., 2021], [Gorbunov et al., 2022], and [Adibi et al., 2022]. Therefore, even though I work on related problems, I struggled to understand the theoretical results. For example,
- Permutation invariance seems like a crucial concept but is not explained anywhere in the main paper.
- Theorem 1: all I could infer from the bound is that if $\delta=0$, the last two terms disappear, and the bound makes sense (since it generalizes smooth, strongly convex minimization). But, intuitively, why does the non-vanishing last term appear?
- Similar comments about Theorem 2 and 3. Adding momentum seems to require using an averaged iterate rather than using the last iterate. Can you comment on if (and why) is this necessary?
- Theorems 4-7 have little to no accompanying discussion or intuition for the stated bounds.
- More questions below.

I do not mean to belittle the authors' contribution. Given my judgment from reading the draft of the main paper, and apart from some questions I have below, I think this is a reasonable contribution. And I admit that many ML conference papers follow this style. But, I think this is not very helpful for the majority of the readers, except for the ones who are already too familiar with the field. Given the page limit in conference papers, maybe a journal would be a better venue for this paper.


**Questions:**

- Assumption 1/2 are for uniformly bounded variance/heterogeneity. But this can be unrealistic in the following case: looking at the bound in Theorem 1, with $\mu$ in the denominator in the last two terms, we can potentially kill these terms by scaling the loss function be a large constant, consequently increasing $\mu$. So, can the current results be modified to adapt to the case when the variance/heterogeneity is affine increasing with gradient norm squared?
- Aren't you also using assumption 2 in Theorem 1-3?
- Why are SGDA-RA and SEG-RA analyzed for separate assumptions? Doesn't SGDA-RA hold under assumptions of Theorem 2?
- Why are SEG-CC and R-SEG-CC stated under different assumptions?

Major confusion about random checks of computation:
- if the server knows random seeds, why does it need the additional $m$ clients to verify computations? Why can't it do that itself? This would be more privacy friendly also, since the clients can still trust the server, but not other clients.
- And how can a client check other clients, without having access to their datasets?
- Won't clients need to communicate their datasets for this to happen? Isn't that a major privacy violation?

---

> ### Author Rebuttal · Authors · 2023-08-08
>
> We thank the reviewer for the detailed feedback. Below we address the reviewer’s questions and concerns.
>
> >**The main weakness of the paper is its writing.**
>
> Thank you for your multiple suggestions and comments. We will apply all necessary changes to the final version of the paper to make it easier to read for a broader audience. In the case of acceptance, we will be allowed to have one more page in the main part that we will use to provide these remarks. Alternatively, to get more space, we can move some of the theorems to the appendix and shorten the part on the numerical experiments.
>
> >**Permutation invariance.**
>
> The definition is given in [1] (Definition B). We will add it to the final version.
>
> >**On Theorems 1-3.**
>
> These results recover SOTA results for SGDA and SEG when $\delta = 0$. When $\delta > 0$, there are non-reducible terms proportional to $\delta$. In Theorems 1 and 2, the terms proportional to $\delta \sigma^2$ appear because of the permutation invariance of SGDA-RA and SEG-RA. Intuitively speaking, permutation invariant algorithms do not track the information from the previous rounds. Therefore, Byzantine workers can hide small perturbations in the noise of the correct stochastic gradients in any single round that can accumulate over time, e.g., this happens with attacks from [2,3]. Next, the term proportional to $\delta\zeta^2$ is unavoidable in general for any algorithm [4]: when the problem is heterogeneous, Byzantines can jointly pretend to be good clients with non-representative data and shift the solution proportionally to the gradient dissimilarity of the good workers.
>
> Theorem 3: although M-SGDA-RA is non-permutation invariant, similar terms appear. As we explain in lines 201-204, this is related to the difficulties of the analysis of Momentum-SGD with biased stochastic gradient for **convex problems**. We bring the attention of the community to this non-trivial question via our results.
>
> >**Averaged-iterate analysis for the momentum-version.**
>
> We analyze the momentum version in terms of the averaged iterate to make the analysis simpler. We believe that one can extend the analysis to the last iterate as well, but we do not do it since we expect that the same problem (the need for large batches) will remain in the last-iterate analysis.
>
> >**Discussion of Theorem 4-7.**
>
> The main property of Theorems 4-7 is that they do not require large batches to converge to any predefined optimization error in the homogeneous case ($\zeta = 0$). They are the first results of their type for Byzantine-robust variational inequalities. Let us discuss here some noticeable details. In each of these theorems, the first two terms recover (up to numerical factors) the SOTA results (in the case of no Byzantines) for distributed SGDA or SEG, respectively. The last terms appear because of the presence of Byzantines. In Theorems 4 and 6, the last terms do not have a linear speed-up in $n$ and are proportional to $\gamma_1$ as well as the second term (there is a typo in Theorem 6: we should have $\gamma_1$ in the last term instead of $\gamma_1^2$). In Theorems 5 and 7, the last terms have better dependence on $\varepsilon$ than the second terms and, therefore, for small $\varepsilon$ the complexities coincide with the SOTA ones for SGDA and SEG (that are obtained for the case of no Byzantines).
>
> >**Assumptions 1 and 2.**
>
> We use bounded variance and heterogeneity assumptions for the sake of simplicity and to make the comparison with the related work easier (these assumptions were used in this area before). Our analysis can be extended to the case of affine increasing variance/heterogeneity, i.e., when $\sigma^2$ is replaced with $A\|\| F_i(x) \|\|^2 + \sigma^2$ in (2) and $\zeta^2$ is replaced with $B\|\|F(x)\|\|^2 + \zeta^2$ in (3), similarly to [4].
>
> >**Assumption 2 in Theorems 1-3.**
>
> Yes, we use it. Thank you for spotting the typo.
>
> >**Different assumptions for SGDA-RA and SEG-RA.**
>
> Thank you for the good question. In fact, (SC) follows from (Lip) and (QSM) with constant $\frac{L^2}{\mu}$: $\|\|F(x)\|\|^2 \leq L^2\|\|x-x^\ast\|\|^2 \leq \frac{L^2}{\mu}\langle F(x), x-x^\ast\rangle$. However, the star-cocoercivity $\ell$ can be significantly smaller than $\frac{L^2}{\mu}$. Next, when $\mu = 0$, SGDA is not guaranteed to converge without cocoercivity [5], while SEG converges. We have the results for SEG-type methods when $\mu = 0$.
>
> >**Different assumptions for SEG-CC and R-SEG-CC.**
>
> This is a typo: in Theorem 7 we should have Assumption 4 instead of 5. Thank you for spotting this.
>
> >**Checks of computations.**
>
> Thank you for the questions. As we mention in “Our Contributions” section, methods with the checks of computations are designed for the homogeneous case only ($\zeta = 0$). This case is also very important for Byzantine-robust applications, as we explain in lines 70-71. The key idea is to use the computing power of workers and not overload the server with extra computations of the mini-batches (though it is possible for a powerful server). Since the data is homogeneous, i.e., the whole data is available to everybody, there are no privacy issues.
>
> Achieving convergence to any predefined in the general heterogeneous setup is impossible under Assumption 2 [4]. However, this is an interesting direction for future research requiring rethinking the whole concept of Byzantine workers for the applications like federated learning. This goes far beyond the scope of this work.
>
> [1] Karimireddy et al. Learning from history for byzantine robust optimization. ICML 2021.
>
> [2] Baruch et al. A little is enough: Circumventing defenses for distributed learning. NeurIPS 2019.
>
> [3] Xie et al.  Fall of Empires: Breaking Byzantine-tolerant SGD by Inner Product Manipulation. UAI 2020.
>
> [4] Karimireddy et al. Byzantine-robust learning on heterogeneous datasets via bucketing. ICLR 2022.
>
> [5] Gidel et al. A Variational Inequality Perspective on Generative Adversarial Networks. ICLR 2019.

---

> > ### Comment · Reviewer_vExH · 2023-08-18
> > **Thanks for the response**
> >
> > Sorry about the late reply. Based on the authors' rebuttal to all the reviewers' comments, I will increase my score to 6. I urge the authors to incorporate the reviewers' comments to improve the readability.

---

> > > ### Author Response · Authors · 2023-08-21
> > > **Thank you for the feedback**
> > >
> > > We thank the reviewer for the reply and for the increase in the score. We promise to apply all necessary changes to the final version.

---

### Official Review · Reviewer_RWvG · 2023-07-08

**Soundness:** 4 excellent
**Presentation:** 3 good
**Contribution:** 3 good
**Rating:** 7
**Confidence:** 3

**Summary:**

This paper studies the problem of Byzantine-robust distributed learning with variation inequalities, e.g., max-min form. Three different robust methods for this problem are developed from the concepts of robust aggregation, client momentum, and checks of computations, respectively. For each of the proposed methods, this paper proves its convergence via strict theoretical analysis. Simulation experiments on quadratic game as well as robust NN training further demonstrate the superiority of the proposed methods on the predictive performance over the previous method.

**Strengths:**

1. The studied problem of Byzantine-robust distributed learning with variational inequalities is important. I think this paper do a good job clarifying different scenarios (or conditions) of this problem and develop an effective method for each of the scenario. The proposed various methods and the corresponding analysis improve the frontier of the relevant research areas.

2. This paper is technically sound with solid theoretical results. Comparison with other relevant work is clear.

3. This paper is well written and easy to follow.

**Weaknesses:**

1. The motivation of tackling variation inequalities problems is not so sufficient. Provide more examples on VIs besides the max-min form may help better understand the significance of this work.

**Questions:**

Minor issue:

The font size in Figure 1 is too small, which makes the legends almost unreadable.

**Limitations:**

Limitations have been properly discussed.

---

> ### Author Rebuttal · Authors · 2023-08-08
>
> We thank the reviewer for the feedback and a positive evaluation of our work. Below we address the reviewer’s questions and concerns.
>
> >**Motivation of tackling variational inequalities.**
>
> In our work, we are mostly motivated by min-max problem formulations that form a quite broad class of problems, as we illustrate in lines 27-33. Variational inequality problem is a convenient generalization of min-max problem. Therefore, we consider everything in the generality of variational inequalities. Regarding the additional motivating examples: variational inequalities arise in game theory, control theory, and differential equations [1].
>
> [1] Facchinei, F., & Pang, J. S. (2007). Finite-Dimensional Variational Inequalities and Complementarity Problems. Springer Science & Business Media.
>
> >**Font size in Figure 1.**
>
> We thank the reviewer for the comment. We will increase the fonts in Figures for the final version of the paper.

---

### Official Review · Reviewer_zd6L · 2023-07-09

**Soundness:** 3 good
**Presentation:** 3 good
**Contribution:** 3 good
**Rating:** 5
**Confidence:** 3

**Summary:**

In this paper, the authors provide robust variants of some popular optimization algorithms in the presence of Byzantine workers. They also use client momentum and random checks of computations to enhance the performance further. They provide theoretical guarantees of converges and empirically test their methods against Robust Distributed Extragradient (RDEG) by [Adibi et al., 2022].

**Strengths:**

The paper is well-written and ideas are presented clearly. The problem description is interesting and the proposed approach combines existing methods with theoretical guarantees. At first glance, proofs of convergence rates seem correct. As their main contribution, the dependency on large batch size is reduced to a lesser extent.

**Weaknesses:**

The novelty of this paper lies in combining existing ideas from the field of distributed optimization and applying them to the problem at hand. In my opinion, these are not original ideas but rather an extension of ideas to a different setting. This is definitely not a trivial task but I believe it's not entirely original either.

**Questions:**

It would strengthen the paper if authors could list down novel ideas/challenges of their paper which are unique to this problem - either in the problem formulation or in proof techniques.

**Limitations:**

Limitations are adequately addressed.

---

> ### Author Rebuttal · Authors · 2023-08-08
>
> We thank the reviewer for the feedback. Below we address the reviewer’s questions and concerns.
>
> >**Concerns about the novelty.**
>
> Our paper indeed relies on the existing methods like SGDA and SEG for variational inequalities and existing techniques for achieving Byzantine robustness studied for minimization problems. However, despite the popularity and importance of variational inequality problems and the significant differences with minimization problems in terms of various aspects [1], such combinations were not studied in the literature. In lines 36-45, we explain what gaps remained in the field before our work. The results obtained in our paper are not known before and, thus, are novel.
>
> Next, we believe that it is not obvious beforehand what combination should be considered and what convergence guarantees one can achieve for the new methods. For example, one could expect that client momentum resolves the issue of SGDA-RA and allows convergence without large batchsize. However, this turned out to be a highly non-trivial task. As we explain in lines 200-204, it is unclear whether one can prove the convergence without large batches even in the special case of convex minimization problems (note that monotonicity can be seen as a generalization of convexity to the case of operators). We are not aware of papers that discuss this issue of Momentum-SGD in the convex case. So, our work brings this problem to the attention of the community. However, this issue was not that evident beforehand since nobody reported it, and the existing analysis from [2] bypasses this issue by considering non-convex problems where the analysis is not that sensitive to the bias in the stochastic gradient.
>
> Therefore, we believe that our paper makes an important step towards a better understanding of Byzantine robustness in the context of variational inequalities.
>
> ---
> References:
>
> [1] Gidel et al. A Variational Inequality Perspective on Generative Adversarial Networks. ICLR 2019.
>
> [2] Karimireddy et al. Learning from history for byzantine robust optimization. ICML 2021.

---

### Author Rebuttal · Authors · 2023-08-08

We thank the reviewers for their feedback, time, and positive evaluation of our work. We appreciate that the reviewers acknowledged several strengths of our work:

- The paper is well-written (Reviewers zd6L and RWvG)

- The ideas are presented clearly (Reviewer zd6L)

- The problem is interesting (Reviewer zd6L)

- Dependency on the large batch size is reduced (Reviewer zd6L)

- The problem is important (Reviewer RWvG)

- The proposed various methods and the corresponding analysis improve the frontier of the relevant research areas (Reviewer RWvG)

- This paper is technically sound with solid theoretical results (Reviewer RWvG)

- Comparison with other relevant work is clear (Reviewer RWvG)

- The paper contains a lot of work (Reviewer vExH)

- The paper gives guarantees in different setups and evaluates the convergence performance of the proposed methods in the numerical experiments (Reviewer AxmQ)

The reviewers also raised several questions and concerns that we addressed in detail in our responses to each reviewer.

We also will be happy to participate in the discussion in the case if reviewers will still have questions/comments/concerns after reading our responses.

---

### Decision · Program_Chairs · 2023-09-21

**Decision:**

Accept (poster)

**Comment:**

This paper studies the problem of Byzantine-robust distributed learning with variational inequalities. The authors provide theoretical guarantees for numerous popular algorithms for solving min-max (more generally VI) problems. The authors corroborate their theoretical findings with experiments on quadratic games and robust neural network training. The overall feedback is positive, the AC agrees and recommends acceptance.